# Flavones enrich rhizosphere *Pseudomonas* to enhance nitrogen utilization and secondary root growth in *Populus*

Jiadong Wu[1,2,3,4,5], Sijia Liu[1,2,3,4,5], Haoyu Zhang[1,2,3,4], Sisi Chen[1,2,3,4], Jingna Si[1,2,3,4], Lin Liu[1,2,3,4], Yue Wang[1,2,3,4], Shuxian Tan[1,2,3,4], Yuxin Du[1,2,3,4], Zhelun Jin[1,2,3,4], Jianbo Xie [1,2,3,4] ✉ & Deqiang Zhang [1,2,3,4] ✉

Plant growth behavior is a function of genetic network architecture. The importance of root microbiome variation driving plant functional traits is increasingly recognized, but the genetic mechanisms governing this variation are less studied. Here, we collect roots and rhizosphere soils from nine *Populus* species belonging to four sections (*Leuce*, *Aigeiros*, *Tacamahaca*, and *Turanga*), generate metabolite and transcription data for roots and microbiota data for rhizospheres, and conduct comprehensive multi-omics analyses. We demonstrate that the roots of vigorous *Leuce* poplar enrich more *Pseudomonas*, compared with the poorly performing poplar. Moreover, we confirm that *Pseudomonas* is strongly associated with tricin and apigenin biosynthesis and identify that gene *GLABRA3* (*GL3*) is critical for tricin secretion. The elevated tricin secretion via constitutive transcription of *PopGL3* and *Chalcone synthase* (*PopCHS4*) can drive *Pseudomonas* colonization in the rhizosphere and further enhance poplar growth, nitrogen acquisition, and secondary root development in nitrogen-poor soil. This study reveals that plant-metabolite-microbe regulation patterns contribute to the poplar fitness and thoroughly decodes the key regulatory mechanisms of tricin, and provides insights into the interactions of the plant's key metabolites with its transcriptome and rhizosphere microbes.

Rhizosphere microbial community structure is highly dynamic in part due to the changes in root exudation over the course of plant development[1–3]. Although some chemical signals released by plants facilitate specific interactions, many have been recognized by previous studies. For example, flavonoids (luteolin, apigenin, etc.) could interact with rhizobial nodulation (NodD) proteins activating the transcription of nodulation genes responsible for the deformation of plant root hairs and assisting rhizobial entry via infection threads[4,5]; coumarins selectively affect the assembly of rhizosphere microbial communities, inducing the colonization of *Pseudomonas simiae* WCS417, thereby improving the niche establishment of microbial partners[6]. Under the infection of pathogens, plants employ a "cry for help" strategy, and signaling chemicals (L-malic acid, salicylic acid, etc.) activated by the immune response change the composition of the

[1]State Key Laboratory of Tree Genetics and Breeding, College of Biological Sciences and Technology, Beijing Forestry University, Beijing, PR China. [2]National Engineering Research Center of Tree Breeding and Ecological Restoration, College of Biological Sciences and Technology, Beijing Forestry University, Beijing, PR China. [3]Key Laboratory of Genetics and Breeding in Forest Trees and Ornamental Plants, Ministry of Education, College of Biological Sciences and Technology, Beijing Forestry University, Beijing, PR China. [4]The Tree and Ornamental Plant Breeding and Biotechnology Laboratory of National Forestry and Grassland Administration, Beijing Forestry University, Beijing, PR China. [5]These authors contributed equally: Jiadong Wu, Sijia Liu. ✉e-mail: jbxie@bjfu.edu.cn; DeqiangZhang@bjfu.edu.cn

rhizosphere microbiome, recruiting beneficial microorganisms to help them resist these stresses[7,8]. Therefore, it is crucial and challenging to elucidate the causal relationship between plant metabolites and beneficial microbes. Flavonoids are one of the most studied classes of such metabolites, regulating both plant development and the interaction with commensal microbes[2]. Root secretion of flavonoids occurs frequently under biotic stress and is involved in promoting microbial colonization during stress generation[9,10]. Infection of part of the tomato (*Solanum lycopersicum*) root system with *Ralstonia solanacearum* changes numerous root exudates and involves disease suppression via the recruitment of disease-suppressing *Streptomyces* for colonization, which was associated with increased exudation of 3-hydroxyflavone[9]. Under abiotic stress, flavonoid production is often elevated in plants[11]. However, our knowledge of how flavonoid-mediated plant-microbe interactions may improve plant resistance to abiotic stresses remains elusive.

The process of microbial recruitment by plants is essentially a "top-down" regulatory process, in which functional genes alter rhizosphere microbial community composition based on the regulation of metabolites or other signaling molecules[6,12,13]. Previous studies have effectively integrated plant transcriptomics or genomics with microbiome community data using methods such as Weighted Gene Co-Expression Network Analysis (WGCNA)[14] and Microbiome-Wide Association Studies (MWAS) analyses[15,16], demonstrating the significance of host gene expression (CYP72A154 and Nucleotide-Binding-Leucine-Rich-Repeat) in shaping the composition of microbial communities. However, analyzing the correlation between host genes and the microbial community cannot elucidate the role of metabolites in plant-regulated microbial structure. The establishment of a comprehensive network involving genes, metabolites, and rhizosphere microbes becomes crucial for a thorough understanding of plant-microorganism interactions.

Poplar (*Populus* L.), a globally cultivated fast-growing and high-yielding timber tree species, comprises five sections: *Leuce, Aigeiros, Tacamahaca, Turanga*, and *Leucoides*[17]. Distinct poplar genotypes exhibit various growth characteristics[18,19], and these differences profoundly influence the productivity and adaptability of the poplars[20,21], because the enhancement of certain growth traits may be closely linked to a plant's resistance to environmental stressors[22,23]. The fast-growing *P. euramericana* Dode manifests a more developed root system than the slow-growing *P. simonii* Carr, concurrently displaying heightened capabilities for nitrogen uptake and assimilation, promoting growth in nitrogen-deficient environments[20]. Additionally, diverse poplar genotypes (or sexes) shape rhizosphere communities by recruiting specific microbial taxa[24-26], under which microbes may alter host performance and fitness directly or via ecosystem services such as nutrient accessibility[27-29]. However, the genetic mechanism by which poplar genotypes regulate the overall assembly and functional changes of microbial communities in nitrogen-poor conditions remains largely unclear, as do the effects of host genotype-selected rhizosphere microbiomes on poplar growth and fitness.

In this study, we hypothesize that (1) the comprehensive gene-metabolite-microbe co-expression network provides an effective research tool to elucidate the genetic mechanisms by which poplar regulates metabolite-mediated specific microbial recruitment; (2) poplar regulates the secretion of flavonoids by functional genes to reshape the rhizosphere microbial community, and attract specific microbes that, through the "cry for help", improve plant adaptability to low-nitrogen stress. We combined transcriptome, metabolome, and microbiome datasets across various genotypes of poplar roots to establish a comprehensive gene-metabolite-microbe network. By using this multidimensional dataset, the mechanisms of poplar recruits target beneficial microbes and how these microbes affect host fitness were revealed. The function of the key genes was further investigated by using molecular experiments. We highlight how host genes, root exudates, and the rhizobiome have mutual interactions and how these explicit processes affect plant fitness.

## Results

### Growth promotion of poplar by the soil microbial community

To assess the importance of the root microbiome in plant fitness, we performed a pot experiment on nine representative poplar species derived from four sections (*Leuce, Aigeiros, Tacamahaca*, and *Turanga*; grown in low nitrogen, unsterilized natural mixed soil; total nitrogen: 0.089%). After three months of growth, eleven phenotypes (plant height, ground diameter, shoot biomass, root biomass, root length, leaf length, leaf width, leaf area, leaf number, chlorophyll content, and leaf nitrogen content) of poplar were detected (Supplementary Fig. 1; Supplementary Data 1). Results indicated that seedlings from *Leuce* demonstrated the most superior biomass, followed by *Aigeiros* and *Tacamahaca*, while *Turanga* exhibited the lowest (ANOVA, $P$ values < 0.01; Supplementary Fig. 1C, D). The growth parameters varied among species. For example, mean root biomass ranged from 0.56 to 15.76 g, plant height ranged from 27.02 to 129.91 cm, and leaf area ranged from 1.06 to 76.30 cm$^2$ (Supplementary Data 1). Notably, the shoot biomass of the fast-growing *P. tomentosa* Lumao 50 (LM50) was 14.02 times greater than the slowest-growing *P. euphratica* H (Peu-H).

To investigate whether genotype-mediated soil microbiota was involved in shaping disparities in poplar growth, a follow-up soil transplant experiment was conducted on the vigorous LM50 and the poorly performing Peu-H. When Peu-H was transplanted into the soil in which LM50 had previously been grown (LM50-grown soil), Peu-H significantly increased shoot biomass compared with soil in which Peu-H had previously been grown (Peu-H-grown soil; an increase of 27.22%; ANOVA, $P$ values < 0.01; Fig. 1A, B). In contrast, LM50 showed significant growth inhibition when transplanted into Peu-H-grown soil compared with LM50-grown soil (19.58% decrease in shoot biomass; ANOVA, $P$ values < 0.01; Fig. 1A, B). It is noticeable that LM50 produced 7.37 g more shoot biomass than Peu-H in sterilized soil, nevertheless, this discrepancy widened to 8.29 g (Peu-H-grown soil) and 10.26 g (LM50-grown soil), respectively (Fig. 1A, B). A similar trend was also observed in *P. alba × P. glandulosa* 84K (84K; $P$ values < 0.05; Supplementary Fig. 2A–C). Notably, there were no significant differences in nutrients in soils (bulk soils or rhizosphere soils) previously grown with LM50 and Peu-H (ANOVA; Supplementary Table 1), and root exudates from all genotypes had no significant effect on poplar (84K and Peu-H) biomass in sterilized soil (Supplementary Fig. 2D–I). These results demonstrate that plant-associated microbiota positively influences poplar growth, but the extent of this effect varies depending on plant genotype. Specifically, the soil microbial community shaped by the vigorous genotype was more conducive to plant growth, whereas the promoting effect of the soil microbial community recruited by the less robust genotype was weaker.

### Taxonomic features of the rhizosphere microbial composition between poplar genotypes

To evaluate the impact of the different poplar genotypes on the microbiome composition and functional potential, samples of bulk soil and rhizosphere soil were collected. Bacterial community composition across the nine poplar species was investigated for each sample type (bulk soil and rhizosphere soil) using Illumina MiSeq sequencing of the V3–V4 region of the 16S rRNA gene. The annotated taxonomic levels were (Domain, Phylum, Class, Order, Family, and Genus)[30]. Across sections, *Turanga* showed the highest Shannon diversity in rhizosphere microbiota, followed by *Aigeiros, Tacamahaca*, and *Leuce* (ANOVA, $P$ values < 0.05; Supplementary Fig. 3A). By contrast, there was no significant difference among bulk soil samples.

The compositional variation of the rhizosphere microbiome among the four sections is driven by significant shifts in the

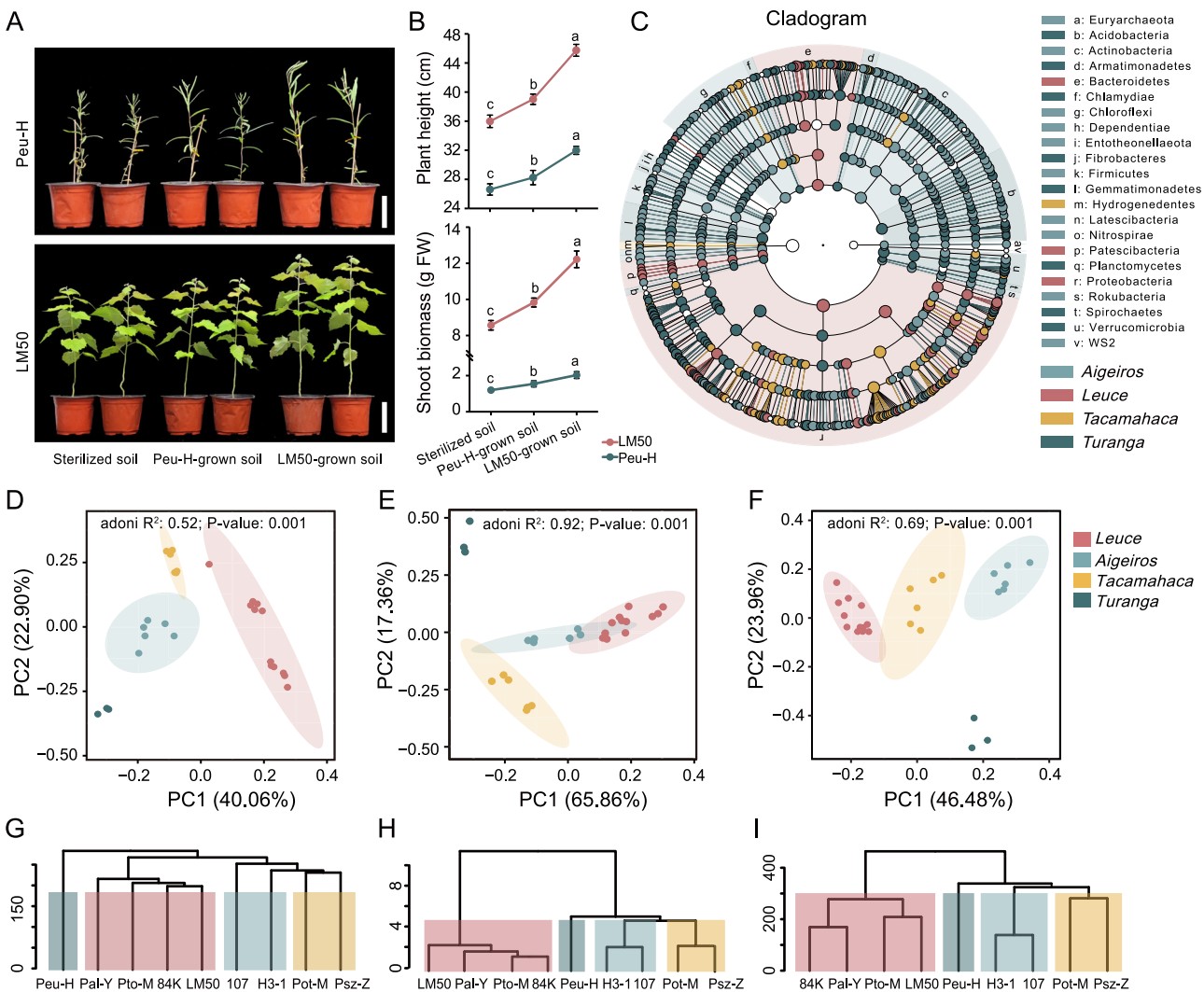

**Fig. 1 | The specific microbial taxa recruited by the poplar rhizosphere may be associated with plant performance. A** Morphological differences of LM50 and Peu-H transplants in different soils (LM50-grown soil or Peu-H-grown soil). **B** Plant height and fresh shoot biomass of LM50 and Peu-H transplants in different soils (LM50-grown soil or Peu-H-grown soil). $n = 3$ biologically independent samples. Each bar represents the mean ± SEM. **C** Linear discriminant analysis effect size (LEfSe) was performed to identify the rhizosphere bacteria that are differentially represented between the different poplar sections. From the inside to the outside, the sequence is boundary–phylum–class–order–family–genus. Each node represents a species, and the larger the node, the higher the relative abundance.

The letters represent different phyla, and the colors indicate that the species is significantly different in the corresponding section (LDA score > 2, two-sided Kruskal–Wallis test, FDR adjusted $P$ values < 0.05). Principal Component Analysis (PCA; $P$ values were calculated by one-way PERMANOVA) and Hierarchical Clustering Analysis (HCA) of the microbiome (**D**, **G**; ASV > 2), phenotype (**E**, **H**), and transcriptome (**F**, **I**; TPM > 0) datasets from the nine poplar species. Different letters indicate significantly different groups (One-way ANOVA, $P$ values < 0.05; $P$ values are shown in the Source Data file). FW, fresh weight. Scale bars: (**A**) 10 cm. Source data are provided as a Source Data file.

relative abundance of 22 specific bacterial phyla (Linear discriminate analysis effect size "LEfSe", LDA score > 2, Kruskal–Wallis test, FDR adjusted $P$ values < 0.05; Fig. 1C, Supplementary Fig. 3B, and Supplementary Data 2). At the genera level, we identified 109 specific markers in *Turanga*, 90 in *Aigeiros*, 45 in *Tacamahaca*, and 37 in *Leuce* (LDA score > 2, Kruskal–Wallis test, FDR adjusted $P$ values < 0.05; Fig. 1C), respectively. We noticed that *Nitrosospira* (1.07%), *Actinomadura* (0.03%), and *Tumebacillus* (0.23%) were highly abundant in the *Turanga* (Supplementary Data 3). *Bacillus* (1.61%) and *Enterobacter* (2.80%) were found to be enriched in *Aigeiros*. Notably, the 37 marker genera detected in *Leuce* accounted for 41.15% of the relative abundance, with *Pseudomonas* having the highest abundance at 13.77%. Together, these results indicate that genotype properties establish root-inhabiting bacterial communities by selecting specific microbial taxa.

## Co-expression network of gene expression, flavonoid productions, and rhizosphere microbial community
To explore the correlation between poplar gene expression and rhizosphere microbial recruitment, we generated 73.7 Gb of root transcriptomic data across nine poplar species (grown in low nitrogen, unsterilized natural mixed soil). This identified 38,739 expressed genes (TPM > 0). The Principal Component Analysis (PCA) and Hierarchical Clustering Analysis (HCA) based on all microbial (amplicon sequence variant, ASV > 2; PERMANOVA, $R^2 = 0.52$, $P$ values < 0.01; Fig. 1D, G and Supplementary Data 4), phenotypic (PERMANOVA, $R^2 = 0.92$, $P$ values < 0.01; Fig. 1E, H and Supplementary Data 4), and transcriptomic (PERMANOVA, $R^2 = 0.69$, $P$ values < 0.01; Fig. 1F, I and Supplementary Data 4) data clearly classified the nine poplar species into four distinct subgroups, each associated with a specific section. This indicates that microbial composition, growth characteristics, and gene expression were

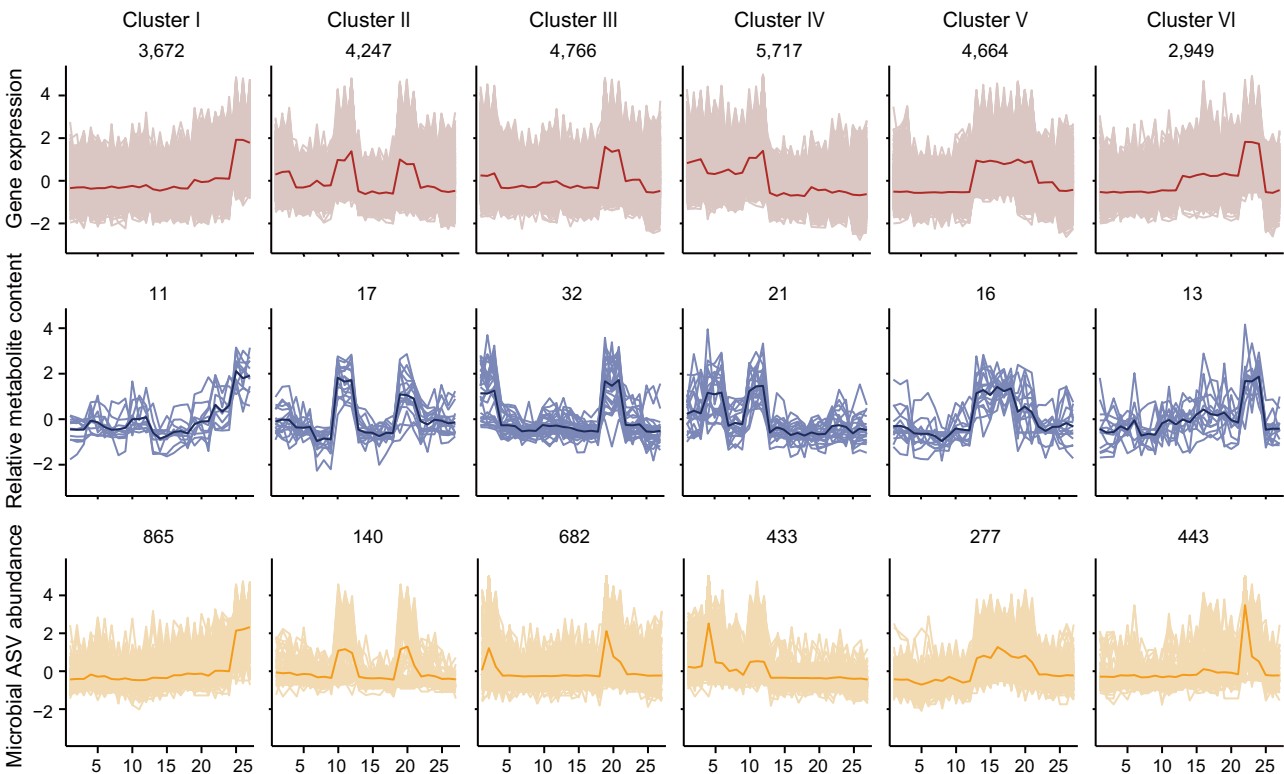

**Fig. 2 | Co-expression network of the transcriptome, metabolome, and microbiome.** The *k*-means clustering algorithm and Pearson's correlation analysis (two-sided; $r \geq 0.7$, *P* values < 0.01) divided poplar gene expression profiles (red), flavonoid metabolome expression profiles (blue), and microbiome (ASVs; orange) into six clusters. The *X*-axis depicts 27 samples from nine poplar species, and the *Y*-axis depicts the *Z*-scores standardized for each gene, flavonoid, and ASV. The bold line is the mean. The numbers shown in each box (for example, 3672 genes, 11 flavonoids, and 865 ASVs for Cluster I) come from the number of genes, flavonoids, and ASVs for all 27 samples in each cluster. The numbers on the *X*-axis represent the samples: *Leuce* (1–3, Pto-M; 4–6, 84K; 7–9, Pal-Y; 10–12, LM50); *Aigeiros* (13–15, H3-1; 16–18, 107); *Tacamahaca* (19–21, Pot-M; 22–24, Psz-Z); *Turanga* (25-27, Peu-H). The genes of Cluster I were enriched in *Turanga* (higher expression than at least one section; |log2FC| > 1, two-sided, FDR adjusted *P* values < 0.05) and the genes of Cluster IV were enriched in *Leuce* (higher expression than at least one section; |log2FC| > 1, two-sided, FDR adjusted *P* values < 0.05). Source data are provided as a Source Data file.

significantly affected by genotype. Moreover, functional enrichment analyses revealed that differentially expressed genes (DEGs; |log2FC| ≥ 1, FDR adjusted *P* values < 0.05; Supplementary Data 5) among the four sections were significantly enriched in functions related to flavonoid metabolism (*P* values < 0.05; Supplementary Fig. 4 and Supplementary Data 6). Flavonoids play vital roles in the assembly of plant root microbiome communities, such as the roots of Arabidopsis (*Arabidopsis thaliana* L.) and maize (*Zea mays* L.)[31,32]. Thus, we hypothesized that poplar functional genes regulate flavonoid synthesis to mediate changes in rhizosphere microbiome composition and diversity.

Next, we quantified 129 flavonoids from the root samples of nine poplar species, of which 110 (85.27%) were differentially accumulated across at least two species (fold change ≥ 3 or ≤ 0.333, *P* values < 0.05; Supplementary Data 7). To gain further insights into the gene-metabolites-microbiome regulatory network, differential flavonoids were initially classified into six clusters based on their accumulation patterns using the *k*-means clustering algorithm (Supplementary Data 8). Subsequently, a rigorous correction (Pearson; $r \geq 0.7$, *P* values < 0.01) was employed to screen for DEGs and ASVs that were significantly associated with the flavonoids in each cluster (Fig. 2, Supplementary Fig. 5, and Supplementary Data 8). Across all clusters, a total of 17,698 DEGs and 2579 ASVs were co-expressed with at least one flavonoid. The genes, flavonoids, and microbes within these clusters demonstrated a distinct abundance pattern related to specific sections, such as *Turanga* (Cluster I), *Leuce* (Cluster

IV), and *Aigeiros* (Cluster V). Altogether, these results indicate that the trends of gene expression, flavonoid accumulation, and rhizosphere microbial enrichment show significant section specificity.

To investigate whether the co-expression network could provide insights into the gene-flavonoid-microbial regulatory network in poplar, we identified 147 enzyme genes within the network that encode enzymes catalyzing the twelve enzymatic reaction steps of the flavonoid biosynthesis pathway (phenylalanine metabolism, phenylpropanoid biosynthesis, and flavonoid biosynthesis; Supplementary Data 9). Four out of thirteen *CHS* genes and all three *flavonoid 3′-hydroxylase* (*F3′H*) genes were present in Cluster IV, where *F3′H* plays a pivotal role in catalyzing the conversion of naringenin to eriodictyol and dihydrokaempferol to dihydroquercetin, crucial precursors in the biosynthesis of flavones and flavanols[33,34]. Correspondingly, eleven (11/21) flavones were observed in Cluster IV. Notably, our findings revealed the presence of 26 basic helix-loop-helix (bHLH) and 38 MYB transcription factors in Cluster IV. Members of the two gene families often synergistically regulate flavonoid biosynthesis[35,36]. Moreover, two *flavonol synthase* (*FLS*) genes and seven (7/16) flavonols were highly correlated in Cluster V. In summary, our network data contains a substantial number of genes related to flavonoid synthesis, indicating that the co-expression network facilitates elucidating the genetic mechanisms of microbial recruitment and identifying candidate genes.

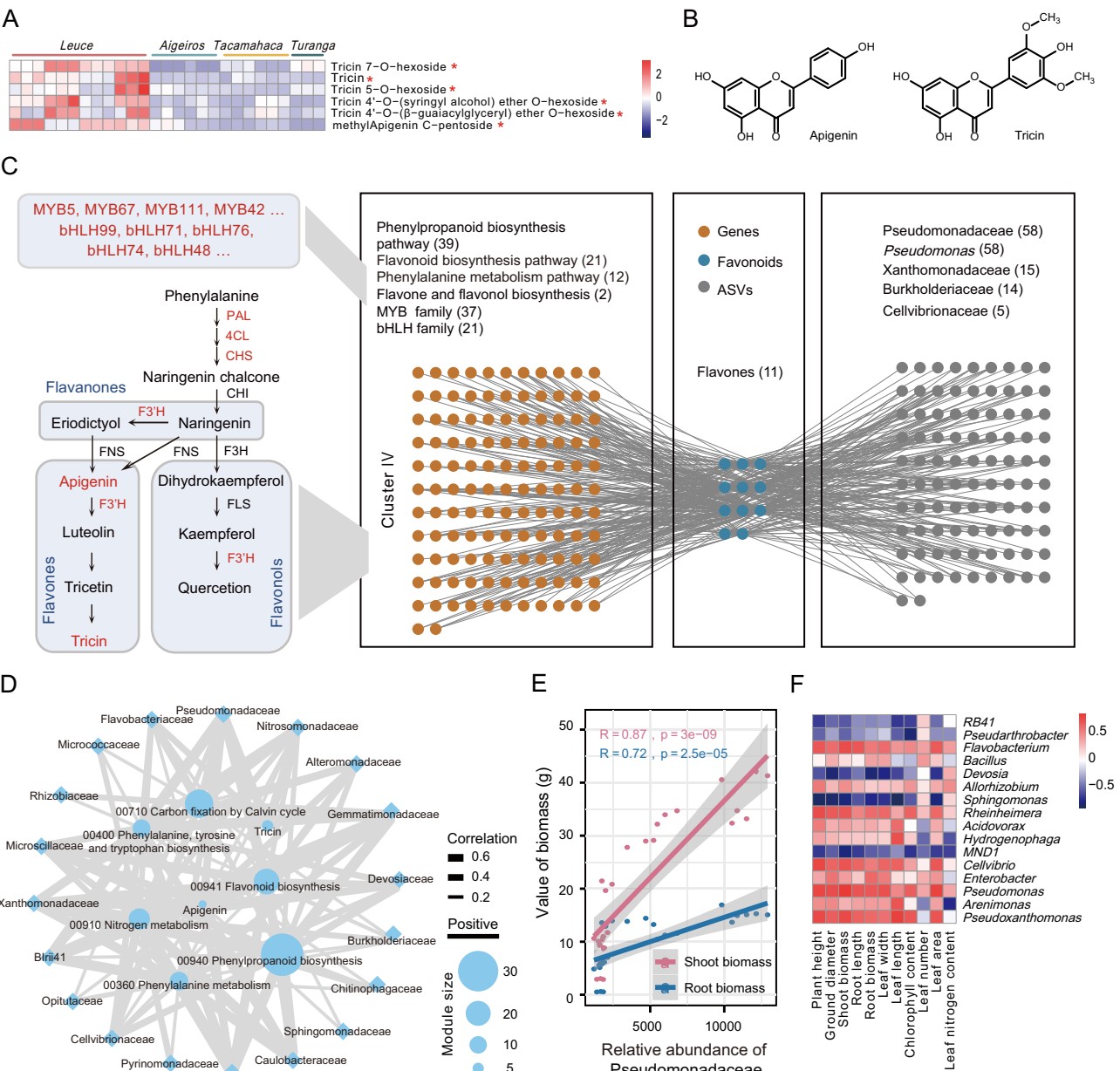

**Fig. 3 | Gene expression and flavonoid accumulation were related to rhizosphere microbial composition and poplar growth performance. A** Apigenin and tricin (including derivatives) heat map in the root systems of different poplar sections. 27 samples correspond to different colors (each color corresponds to one section). Asterisks denote the flavones that were enriched in *Leuce* (One-way ANOVA, *P* values < 0.05; *P* values are shown in the Source Data file). **B** Chemical structure formulas for apigenin and tricin. **C** Correlation network of flavonoid-related genes, flavones, and ASVs in Cluster IV. Red color indicates genes detected in the network. Highly correlated associations (two-sided; Pearson; $r \geq 0.7$, *P* values < 0.01) were present. **D** Correlation network of the top 20 microbial families with flavone modules and gene modules of Cluster IV (two-sided; Mantel test; *P* values < 0.05). The gene modules (00910 nitrogen metabolism, 00710 Carbon fixation by Calvin cycle, 00400 Phenylalanine, tyrosine, and tryptophan biosynthesis, 00360 phenylalanine metabolism, 00940 phenylpropanoid biosynthesis, and 00941 flavonoid biosynthesis) are modules of KEGG enrichment analysis in Cluster IV. The flavone modules are apigenin (with derivatives) and tricin (with derivatives) of Cluster IV. The node size represents the number of elements included (for example, the flavonoid biosynthesis module has 21 genes). Solid edges indicate positive relationships. Edge thickness denotes the strength of correlations. **E** Pearson's correlation between Pseudomonadaceae and the shoot and root biomass of poplars (two-sided). The gray shading around the line represents a confidence interval of 0.95. **F** Pearson's correlation (two-sided) between dominant genera and eleven characteristics of poplars. The color of the heat map represents the size of the correlation coefficient. Source data are provided as a Source Data file.

## Rhizosphere microbial composition was consistent with gene expression, flavonoid accumulation, and poplar growth performance

To further enrich putative regulating networks, we specifically focused on Cluster IV and Cluster I, which peaked at the *Leuce* and *Turanga*, respectively (Supplementary Fig. 5A–C), as the two sections demonstrated the most contrasting growth performances (Supplementary Fig. 1C, D). Functional enrichment analyses showed genes in Cluster I (enriched in *Turanga*; |log2FC| ≥ 1, *P* values < 0.05; Supplementary Data 5) were mainly associated with housekeeping functions such as genetic information processing, ribosome biogenesis, and mismatch repair (*P* values < 0.05; Supplementary Fig. 6A; Supplementary Data 6). By contrast, genes in Cluster IV (enriched in *Leuce*; |log2FC| ≥ 1, *P* values < 0.05; Supplementary Data 5) are involved in energy and

matter cycles (carbon fixation in photosynthetic organisms, nitrogen metabolism, and energy metabolism), as well as flavonoid metabolism (phenylalanine metabolism, phenylpropanoid biosynthesis, and flavonoid biosynthesis; *P* values < 0.05; Supplementary Fig. 6B; Supplementary Data 6). Therefore, we subsequently selected Cluster IV for further analysis.

We found that the ASVs in Cluster IV predominantly belong to Proteobacteria (234/433, 54.04%) and Bacteroidetes (81/433, 18.71%), with Pseudomonadaceae (62), Chitinophagaceae (22), Xanthomonadaceae (19), and Burkholderiaceae (16) being the most numerous, and these taxa were particularly enriched in the *Leuce* (LDA score > 2, FDR adjusted *P* values < 0.05; Supplementary Data 2). Within the metabolic cluster, the flavones, tricin, and apigenin (with their derivatives), were uncovered as the most enriched metabolites in *Leuce* (ANOVA, *P* values < 0.01; Fig. 3A). Apigenin could recruit beneficial bacteria such as *Rhizobium*, Oxalobacteraceae, and *Pseudomonas*, enhancing the plant's nitrogen uptake capacity[31,37,38]. Tricin is structurally similar to apigenin and shares the same KEGG pathway as apigenin (ko00944), suggesting that tricin may have a similar biological function to apigenin (Fig. 3B).

To determine whether poplar rhizosphere microbiome composition is linked with its transcriptome signature, flavonoid metabolism, and growth performance, we performed detailed analyses of the subnetwork of genes, flavones, microbes, and growth traits. Consistent with our expectations, the expression of genes in the flavonoid metabolism (phenylalanine metabolism, phenylpropanoid biosynthesis, and flavonoid biosynthesis) showed a significant positive correlation with the accumulation of flavones (*P* values < 0.01; Fig. 3C). The accumulation of flavones was associated with ASVs from Pseudomonadaceae, Burkholderiaceae, Cellvibrionaceae, and Xanthomonadaceae (*P* values < 0.01), among which Pseudomonadaceae had the highest number of ASVs (58; all ASVs belong to the *Pseudomonas*). Notably, Pseudomonadaceae was among the top families showing the highest correlation with flavone modules and flavonoid-related gene modules (*P* values < 0.01; Fig. 3D). Pseudomonadaceae, taxa enriched from 0.86% in the bulk soil to the highest abundance in the *Leuce* rhizosphere (13.78%), was specifically enriched in *Leuce* and correlated with poplar growth (ANOVA, *P* values < 0.01; Fig. 3E and Supplementary Fig. 7A, B). In particular, at the genus level, *Pseudomonas*, which had been demonstrated to have beneficial potentials in nitrogen fixation, phosphorus solubilization, secretion of growth hormones, and antimicrobial activities[39,40], was strongly correlated with plant growth characteristics (*P* values < 0.01; Fig. 3F and Supplementary Fig. 7C). Overall, these results showed that specific bacteria taxa become enriched as a consequence of Leuce-specific properties and are associated with gene expression, flavonoid accumulation, and plant growth.

### Tricin and apigenin-mediated pseudomonads enhancing nitrogen utilization and secondary root growth in poplar

We further isolated eleven *Pseudomonas* strains from rhizosphere soil samples of *Leuce* (Pto-M, 84K, Pal-Y, and LM50), and the 16S rRNA genes of ten isolates exhibited highly homologous (> 99%) to ASVs of Cluster IV (Supplementary Data 10). Further characterization showed that seven isolates possessed the capacity for nitrogen fixation and carried the *nitrogen fixation* (*nifH*) gene, and eight isolates demonstrated the secretion of indole-3-acetic acid (IAA; Supplementary Fig. 8A–D). Flagellated bacteria, such as pseudomonads, could achieve movement towards plant roots through swarming motility, with the success of this process determining the efficiency of root colonization[41]. Notably, Pto1, Pto5, and Pto10 enhanced swarming motility in the presence of 5 μM tricin and 100 μM apigenin (Fig. 4A). qRT-PCR analysis suggests that flagellar-related genes (*motA, fliG*, and *bifA*)[37] and biofilm formation-related gene *algU* were activated[42], which may be critical for successful bacterial root colonization (ANOVA, *P* values < 0.01; Fig. 4B).

To investigate the potential of *Pseudomonas* isolates on poplar fitness, we inoculated three individual strains and constructed synthetic communities (SynComs: Pto1, Pto5, and Pto10). Using $^{15}$N isotope labeling, we traced the nitrogen absorbed by poplar from the soil, while nitrogen fixed by microbes from the air remained unlabeled. Inoculated isolates significantly increased the shoot biomass (26.04%–48.03%), root biomass (57.51%–81.46%), and leaf nitrogen content (7.98%–10.15%) of poplar (84K; ANOVA, *P* values < 0.01; Fig. 4C, D and Supplementary Fig. 9A). Similar trends were also observed in other plant species (*P* values < 0.01; Supplementary Fig. 10). After inoculation, the $^{15}$N ratio of leaves of inoculated isolates decreased, indicating that pseudomonads promoted the nitrogen absorption of poplar through biological nitrogen fixation (BNF; Supplementary Table 2). Consistently, in sterile nitrogen-poor even medium, the number and length of poplar secondary roots (SRs) increased by 9.92 and 2.88 times after inoculation with Pto1, respectively (*P* values < 0.01; Fig. 4E, F). However, the promoting effects of Pto1 on poplar growth and SR induction were diminished in a medium with sufficient nitrogen supply (Fig. 4F and Supplementary Fig. 9B). These results suggest that the functions of pseudomonads may rely on the cross-talk between specific nitrogen starvation signaling and plant responses.

### Pto1 induces the *PLT3PLT5PLT7*-mediated LR pathway in Arabidopsis by secreting IAA

The intricate architecture of the root system in dicotyledons has multiple types of SRs and encompassing lateral roots (LRs), adventitious lateral roots (adLRs), which are regulated by distinct genetic pathways[43–46]. To elucidate the nature of the induced SRs following inoculation with Pto1, we conducted a structural analysis of the root systems of the *plethora 3,5,7* (*plt3plt5plt7*) Arabidopsis triple mutant, which exhibits compromised LR formation, and the *wuschel-related homeobox 11,12* (*wox11wox12*) Arabidopsis double mutant, displaying deficiencies in adLR and adventitious roots (AR) formation[47]. In the *plt3plt5plt7* mutants inoculated with Pto1, no visible SR was observed at seven days (Fig. 4H, I). Conversely, the *wox11wox12* mutants exhibited a significant increase in the number of SRs, comparable to the increase observed in wild-type roots following Pto1 inoculation. When IAA was added to the medium, the development pattern of Arabidopsis roots was similar to that resulting from Pto1 inoculation. However, the auxin inhibitor 2,3,5-triiodobenzoic acid (TIBA) hindered SR growth in all Arabidopsis lines, regardless of the presence of Pto1. These results suggest that Pto1-secreted IAA plays a critical role in inducing *PLT3PLT5PLT7*-mediated LR pathways in Arabidopsis.

### *PopGL3* regulates the synthesis of tricin to recruit *Pseudomonas*

We conducted an analysis of the co-expression network to identify the regulators associated with flavone biosynthesis, given its significance in microbial recruitment. In Cluster IV, which enriched apigenin and tricin, a member of the bHLH transcription factor family, *bHLH1* (i.e., *GL3*), was identified (enriched in *Leuce*; ANOVA, *P* values < 0.01; Supplementary Fig. 11A). It exhibited strong co-expression with flavones and genes related to flavonoid biosynthesis (Supplementary Fig. 11A). DAP-seq experiment revealed that *PopGL3* could regulate the transcription of *peroxidase 2* (*PopPA2*), *PopF3'H, Phospho-2-oxo-3-deoxyheptonate aldolase* (*PopDAHP*), *cinnamoyl coa reductase 1* (*PopCCR1*), and *tetrahydroberberine oxidase* (*PopTHB*), which are involved in flavonoid synthesis (Fig. 5A–C). The constitutive expression of *PopGL3* or *PopCHS4* could activate the transcription of *PopF3'H* and *flavone synthase* (*PopFNS*) and release more tricin in the rhizosphere of the *PopCHS4-OE* (chalcone synthase catalyzes the first committed step of the multi-branched flavonoid pathway) and *PopGL3-OE* lines (*P* values < 0.01; Fig. 5 E, F, Supplementary Fig. 12, and Supplementary Fig. 13).

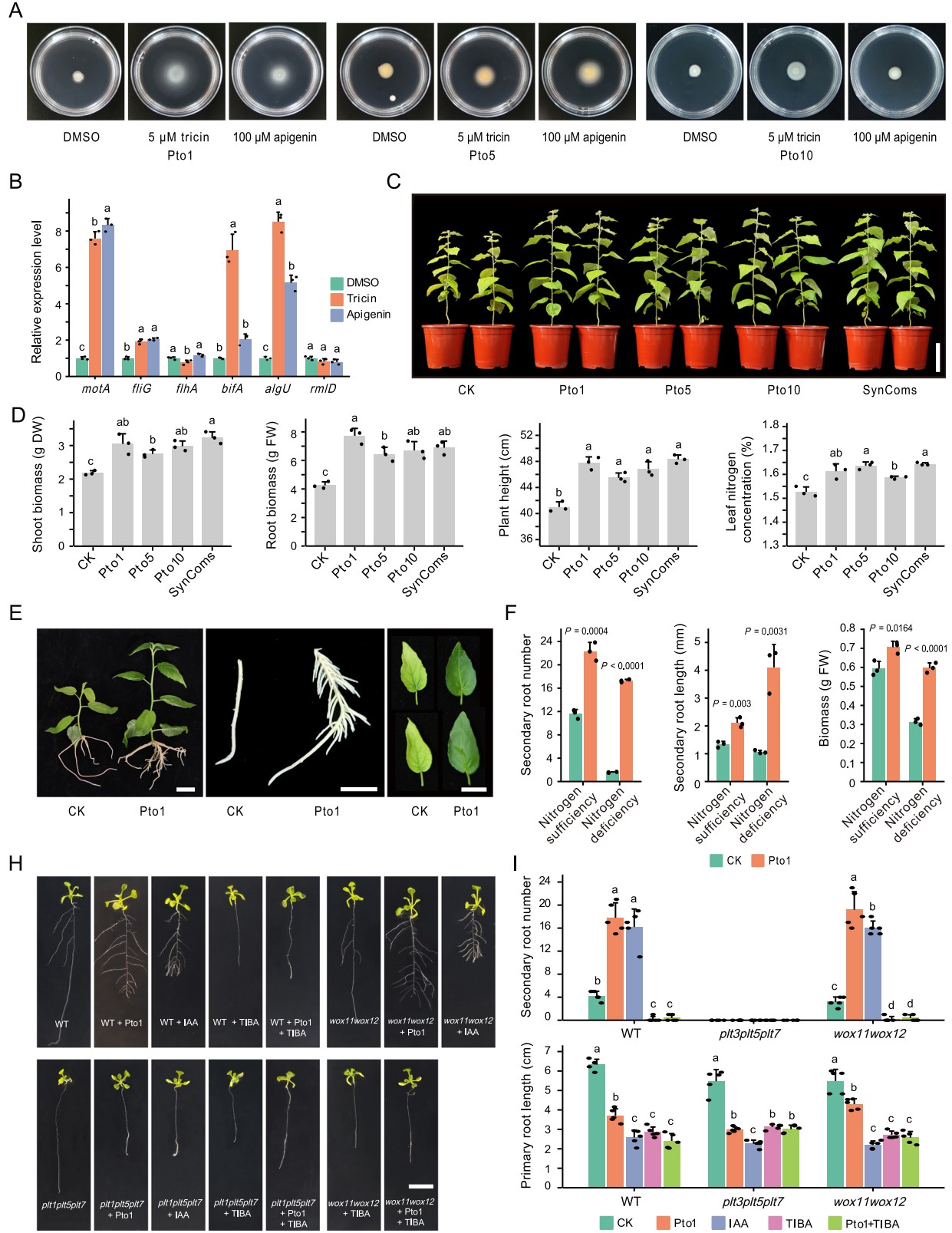

The contribution of BNF to nitrogen nutrition in different poplar genotypes was determined by the $^{15}$N isotope dilution method. Following two months of growth in an unsterilized natural soil mixture (low nitrogen; with a small amount of $^{15}$N-labeled ammonium nitrate), *PopGL3-OE* and *PopCHS4-OE* plants displayed increased biomass and leaf nitrogen accumulation (*P* values < 0.01; Fig. 5D, G and Supplementary Fig. 11B, C). Compared with the wild-type, the contribution of

BNF by transgenic plant root microorganisms increased (Supplementary Table 3). Conversely, all genotypes grew weakly in sterilized soil, with no difference in biomass production. Amplicon sequencing was used to elucidate the reasons for the differences in BNF among different poplar genotypes. The results indicated that transgenic (*PopGL3-OE* and *PopCHS4-OE*) plants reshaped the rhizosphere microbial composition and significantly enriched *Pseudomonas* (*P*

**Fig. 4 | Flavone-mediated pseudomonads promote nitrogen uptake and secondary root growth in poplar. A** Swarming motility of *Pseudomonas* strains Pto1, Pto5, and Pto10 on 0.3% agar medium in the presence of either 5 µM tricin or 100 µM apigenin. **B** qRT-PCR assays revealed that tricin and apigenin induce the expression of flagellar-related genes (*motA*, *fliG*, *flhA*, and *bifA*) and biofilm formation-related genes (*algU* and *rmlD*) in pseudomonad (Pto1). The DMSO-treated strain was used as a negative control. *n* = 3 biologically independent samples. **C** Pot experiment of inoculating poplar (84K) with pseudomonads in nitrogen-poor soil. **D** Dry shoot biomass, fresh root biomass, plant height, and leaf nitrogen concentration of poplar (84K) inoculated with pseudomonads in nitrogen-poor soil. *n* = 3 biologically independent samples. **E** Growth differences of poplars (84K) inoculated with Pto1 in sterile nitrogen-poor culture medium. Whole plant (left), root (middle), leaf (right). **F** The secondary root number, secondary root length,

and total fresh biomass of poplars (84K) inoculated with Pto1 in sterile nitrogen-poor medium and sterile nitrogen-rich medium. *n* = 3 biologically independent samples. 10 mM MgSO$_4$ solution was used as a negative control (two-sided Student's *t*-test). **H** Growth differences of wild-type (WT), *plt3plt5plt7*, and *wox11wox12* Arabidopsis seedlings growing on 1/2 MS agar plates with Pto1, IAA, TIBA, Pto1 + TIBA, or mock. **I** Quantification of secondary root (SR) number and primary root length in WT, *plt3plt5plt7*, and *wox11wox12* Arabidopsis seedlings under mock, IAA, TIBA, Pto1 + TIBA, and Pto1 inoculated conditions. *n* = 5 biologically independent samples. Different letters indicate significantly different groups (One-way ANOVA, *P* values < 0.05; *P* values are shown in the Source Data file). Each bar represents the mean ± SEM. DW, dry weight; FW, fresh weight. Scale bars: (**C**) 10 cm; (**E**) 1 cm; (**H**) 1 cm. Source data are provided as a Source Data file.

values < 0.01; Fig. 5H, I, Supplementary Fig. 11D, E, by STAMP; Supplementary Fig. 14, Supplementary Data 11, by ANCOM-BC2).

Evidence from our experiment suggests that the increased abundance of *Pseudomonas* in the transgenic plants is like due, in part, to the greater absolute depletion of most other bacterial lineages, but does not rule out the positive selection by the transgenic plants through the tricin pathway. To confirm the increased root colonization of *Pseudomonas* in the transgenic plants, we tagged Pto1 with the red fluorescent protein (RFP) gene and used confocal microscopy to image the colonization of root tissue across various genotypes. We observed significantly enhanced colonization and increased fluorescence density in the roots of *PopGL3-OE* and *PopCHS4-OE* plants using confocal microscopy (*P* values < 0.01; Fig. 6A, B). Additionally, the colony forming units (CFUs) statistics further confirm this conclusion (*P* values < 0.01; Fig. 6C). In conclusion, *PopGL3*, a regulator of flavone biosynthesis, recruits *Pseudomonas* by secreting tricin to promote the growth and nitrogen uptake of poplar. Taken together, these data suggest that in a controlled laboratory setting and in the absence of other microbes, the observed increase in *Pseudomonas* abundance in the *PopGL3-OE plants* is accompanied by increased colonization and that this increase is potentially beneficial to poplar fitness.

## Discussion

### Comprehensive gene expression, flavonoid metabolism, and rhizosphere microbial community co-expression network construction

Given the importance of the rhizomicrobiome in plant development, nutrition acquisition, and stress tolerance, deciphering the molecular regulatory network of plant-microbe interactions could substantially contribute to improving plant yield and quality. Current multi-omics studies of plant-microbial interactions have mostly relied on methods such as WGCNA and MWAS, which are confined to the analysis of these binary transcriptome-microbiome datasets, often failing to effectively find metabolites (or other signaling molecules) that directly shape the structure of plant-microbial communities[14–16]. In this study, a dataset comprising gene expression, metabolic profiling, and microbial community derived from four sections of poplar was generated, constructing a comprehensive gene-flavonoid-microbe co-expression network (Supplementary Fig. 15). The 110 differential flavonoids, 17,698 DEGs, and 2579 ASVs were classified into six co-expression clusters. Among them, DEGs and ASVs, which are closely related to flavonoids, accounted for 45.69% of the total expressed genes and 9.30% of the total rhizosphere ASVs, respectively.

Within this network, we identified 147 enzyme genes that encode enzymes catalyzing the twelve enzymatic reaction steps of the flavonoid biosynthesis pathway. The MBW ternary complexes containing R2R3-MYB and bHLH transcription factors along with WD-repeat proteins have been reported to regulate the biosynthesis of flavonoids[48,49]. A total of 97 MYB and 72 bHLH transcription factors were identified, including MYC2 (bHLH), which shifted root microbiota composition to enhance Arabidopsis growth and immunity under

shade[50]. Moreover, we've unveiled the pivotal role of flavonoids in shaping the composition of the poplar root-associated microbial community, particularly in their intimate associations with beneficial microbes like *Pseudomonas*, *Bacillus*, and Actinobacteriota, known to confer advantages to plant fitness[9,41,51]. The investigation within the network not only unveils intricate linkages between plant genetic regulation and metabolite synthesis but also elucidates the direct influence of these metabolites on the structure of microbial communities, offering valuable guidance for future experimental designs.

Although the clustering patterns of DEGs and ASVs strongly associated with flavonoids are consistent with global genes and microbes, some compounds, such as hormones and terpenoids, were not quantified in our samples due to the limited scope of detection in this study. Additionally, the root endosphere microbiome or fungi were not also tested. When we obtain this information, the number of genes and microbes co-expressed with metabolites is likely to increase further, providing a richer resource for in-depth investigations of the plant genetic networks that regulate the recruitment of microbes by metabolic pathways.

### Strategies of poplar "cry for help" rhizosphere microorganisms under low nitrogen conditions

Previous studies have shown that stressed plants recruit beneficial bacteria to colonize their roots by secreting metabolites, promoting the opposite effects on plant growth and health induced by stress, known as a "cry for help" strategy[9,41]. Flavonoids, a major category of specialized metabolites in plants, significantly influence plant growth and development and play a critical role in mediating several plant-microbe interactions[1,3]. For instance, maize FNSI2-mediated apigenin and luteolin have been shown to enhance the abundance of Oxalobacteraceae in the plant rhizosphere, improving host performance under nutrient-limiting conditions[31].

Using the gene-flavonoid-microbe co-expression network, we investigated Cluster IV, which peaked at the fastest-growing *Leuce*. We found that genes in Cluster IV were significantly enriched in flavonoid metabolism-related pathways, including phenylalanine metabolism, phenylpropanoid biosynthesis, and flavonoid biosynthesis. Notably, Pseudomonadaceae (62) in Cluster IV is the taxa with the most numerous ASVs. The Pseudomonadaceae have been demonstrated to enhance plant growth through the processes of BNF or phosphorus solubilization[39]. Within the metabolic cluster, flavones such as tricin and apigenin (with their derivatives) were the most abundant and were significantly enriched in *Leuce*. Correlation analysis revealed that genes related to flavonoid biosynthesis and flavones exhibited the strongest association with Pseudomonadaceae and *Pseudomonas*, while the increased abundance of Pseudomonadaceae and *Pseudomonas* was highly correlated with poplar's growth characteristics.

Experiments have shown that apigenin and tricin (in Cluster IV) enhance the swarming motility and biofilm synthesis of pseudomonad isolates, and this flavone-mediated mechanism significantly promotes the mobility of the pseudomonads at the soil/root interface, favoring

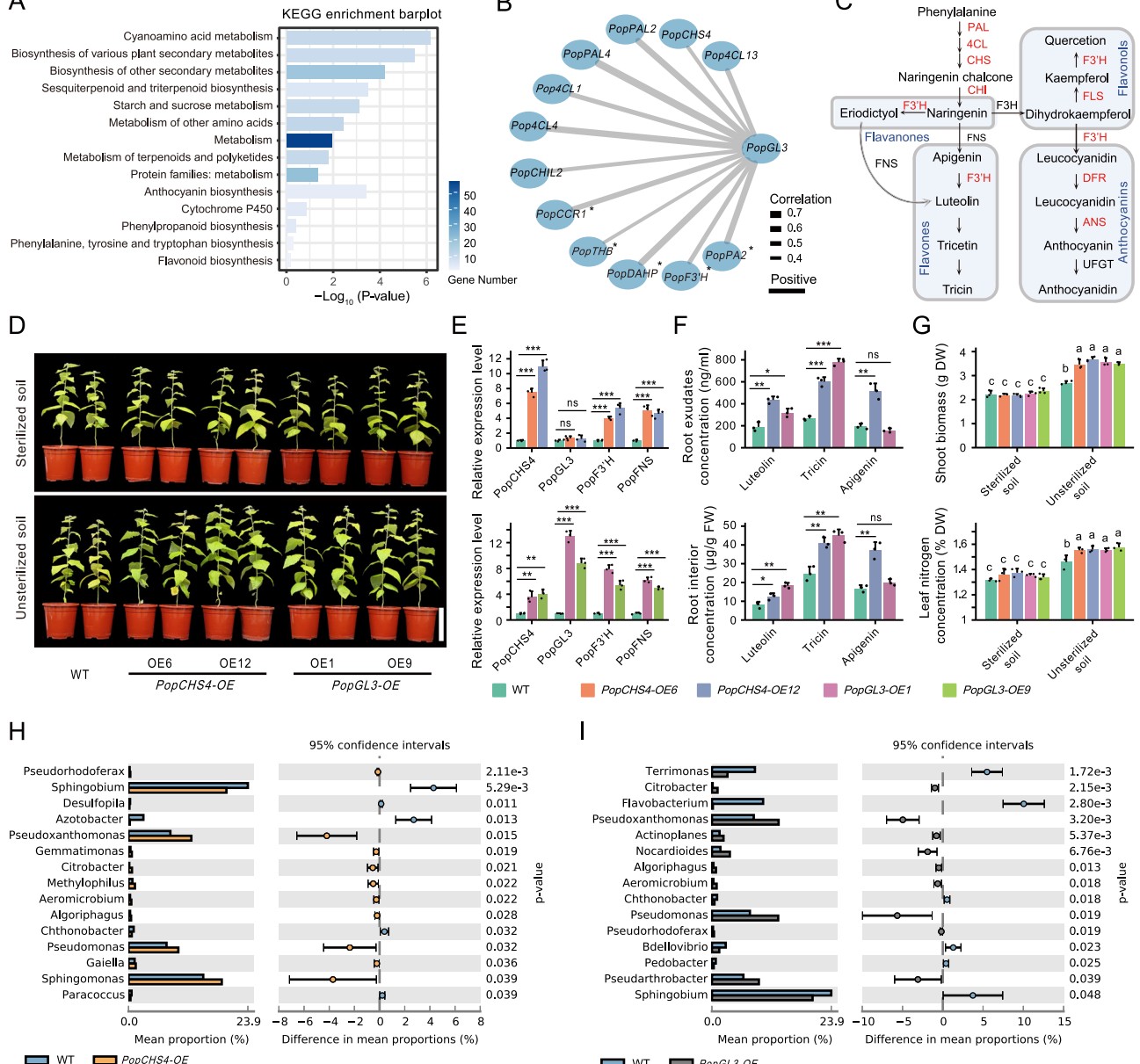

**Fig. 5 | *PopGL3* regulates tricin synthesis to recruit *Pseudomonas*. A** KEGG enrichment analyses of DAP experimental analysis results for *PopGL3* (One-sided Fisher exact-test, *P* values are not adjusted). **B** A flavonoid-related gene network was established based on the DAP assay results of *PopGL3*. Pearson correlation coefficient values were calculated for each pair of genes (two-sided). Solid edges indicate positive relationships. Edge thickness denotes the strength of correlations. Asterisks denote the genes that were the result of two repeats of the DAP experiment, and the other genes were the result of one repeat. **C** Schematic representation of flavonoid biosynthesis and regulation in poplar. The red font indicates genes regulated by *PopGL3* based on the DAP assay (results of at least one repeat). **D** Growth differences between WT, *PopCHS4-OE*, and *PopGL3-OE* poplar lines in sterilized or unsterilized nitrogen-poor soil. Gene relative expression level (**E**), root interior and root exudate flavone concentration (**F**) of WT, *PopCHS4-OE*, and

*PopGL3-OE* poplar lines in unsterilized nitrogen-poor soil. *n* = 3 biologically independent samples. **G** Dry shoot biomass and leaf nitrogen concentration of WT, *PopCHS4-OE*, and *PopGL3-OE* poplar lines in sterilized or unsterilized nitrogen-poor soil. *n* = 3 biologically independent samples. Abundance differences between *PopCHS4-OE* (**H**) and *PopGL3-OE* (**I**) poplar lines and WT rhizosphere microbiomes at the genus level (two-sided Welch's *t*-test, by STAMP). $n_{WT}$ = 3, $n_{PopCHS4-OE}$ = 6, $n_{PopGL3-OE}$ = 6; biologically independent samples. Asterisks indicate significant differences between different groups (two-sided Student's *t*-test, ***P* values < 0.001, ***P* values < 0.01, **P* values < 0.05, ns: not significant; *P* values are shown in Source Data file). Different letters indicate significantly different groups (One-way ANOVA, *P* values < 0.05; *P* values are shown in the Source Data file). Each bar represents the mean ± SEM. DW, dry weight; FW, fresh weight. Scale bar: (**D**) 15 cm. Source data are provided as a Source Data file.

the successful colonization of the plant root surface[37,41]. In Cluster IV, the transcription factor *GL3* had strong co-expression with flavones and genes related to flavonoid biosynthesis. *GL3* was reported to interact with MYB transcription factors and WD40 repeat proteins to form the MYB-bHLH-WD40 (MBW) transcriptional complex, regulating anthocyanin synthesis[48,52]. However, the potential roles of *GL3* in flavone synthesis and interactions with the rhizosphere microbiome

remain unclear. Rhizosphere microbiome analyses of *PopGL3-OE* and *PopCHS4-OE* plants, integrating the metabolite profiles of root extracts and secretions, demonstrate the causal role of *PopGL3* in tricin secretion and recruitment of *Pseudomonas*. A series of inoculation experiments with tricin-mediated isolates confirmed their beneficial effects on poplar growth, nitrogen accumulation, and SR growth. In summary, our findings suggest that the poplar *GL3* gene regulates tricin synthesis

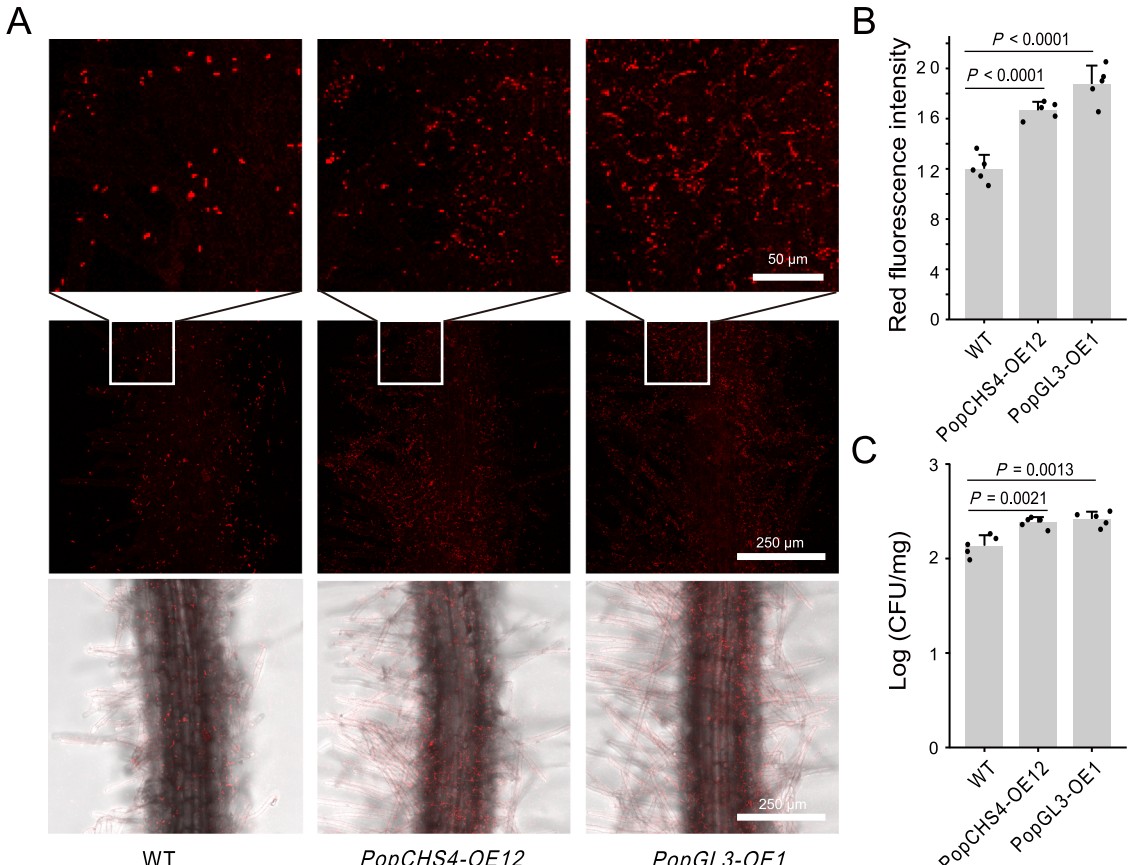

**Fig. 6 | *PopGL3* recruits pseudomonads to colonize poplar roots. A** Confocal fluorescence imaging of RFP-tagged Pto1 colonizing poplar roots of different genotypes. **B** The fluorescence intensity of RFP in poplar roots of different genotypes. The fluorescence intensity of the samples was measured by the ImageJ. $n = 5$ biologically independent samples (two-sided Student's $t$-test). **C** Colonization amount of different genotypes of poplar roots with Pto1. Colony forming units (CFUs) were quantified per fresh weight of roots. $n = 5$ biologically independent samples (two-sided Student's $t$-test). Each bar represents the mean ± SEM. Source data are provided as a Source Data file.

and secretion to call for pseudomonad colonization to help it grow and nitrogen absorption under nutrition-deficient conditions (Fig. 7).

## "Matthew effect" of interaction between host and microorganisms

We found that LM50 (*Leuce*) produced 7.37 g more stem biomass than Peu-H (*Turanga*) in sterile and nitrogen-poor soil. In the unsterilized, nitrogen-deficient soil, this difference widened to 8.29 g (Peu-H-grown soil) and 10.26 g (LM50-grown soil), respectively. Notably, there were no significant differences in nutrients between soils (bulk soils or rhizosphere soils) previously grown with LM50 and Peu-H; root exudates had no significant effect on poplar growth; and bulk soil microbial diversity showed no significant variation. These results indicate that genotype-specific microbiota exerts varying degrees of positive feedback on poplar growth. Each section recruits specific taxa to shape its own rhizosphere microbial community. For instance, the *Leuce* enriches *Pseudomonas* to aid in nitrogen uptake and SR growth. The abundance of *Pseudomonas* in the *Leuce*, *Aigeiros*, *Tacamahaca*, and *Turanga* is 13.77%, 3.47%, 2.78%, and 2.76%, respectively. Therefore, differences in the host's ability to "cry for help" to beneficial microorganisms lead to different degrees of feedback from recruited microorganisms on their own fitness. This disparity, possibly regulated by plant genes, signifies that vigorous *Leuce* elevated the tricin secretion via heightened *GL3* expression, driving pseudomonad colonization in the rhizosphere and enhancing growth, nitrogen acquisition, and SR development in nitrogen-poor soil (Fig. 7). Consistent with our finding, plant resistance genes *GsMYB10* transgenic soybean recruited *Bacillus* and *Aspergillus*, which further enhanced plant resistance to stresses under aluminum (Al) toxicity[53]. In social psychology, the "Matthew effect" describes the phenomenon that the strong become stronger and the weak become weaker[54]. We introduce the concept of the "Matthew effect" in plant-microbial interactions. That is, vigorous or resistant plant genotypes can recruit specific microbes to give them more growth advantages or better resistance. Parallelly, this effect may also be reflected in the interaction between microbes and roots. Root caps and root hairs serve as crucial determinants for the assembly process of the rhizosphere microbiome[55,56]. As the quantity and length of SR increase, the spatial distribution of plant-secreted nutrients and metabolites also expands, enhancing the plant's regulatory influence over the rhizosphere microbial community. Conversely, the rhizosphere's available area for microbial colonization expands with the development of SRs, prompting enhanced beneficial activities by microorganisms toward the plant. This is consistent with previous studies showing that the assembly process of plant rhizosphere microorganisms is closely related to plant root structure[57–59].

## Methods
### Plant and soil materials, growth conditions
Nine species of poplars from four sections (Supplementary Table 4), *Leuce* (*Populus tomentosa* × *P. bolleana* M "Pto-M", *P. alba* × *P. glandulosa* 84K "84K", *P. alba* × *P. glandulosa* Y "Pal-Y", *P. tomentosa* Lumao 50 "LM50"), *Aigeiros* (*P. euramericana* 74/76 "107", *P. euramericana* H3-

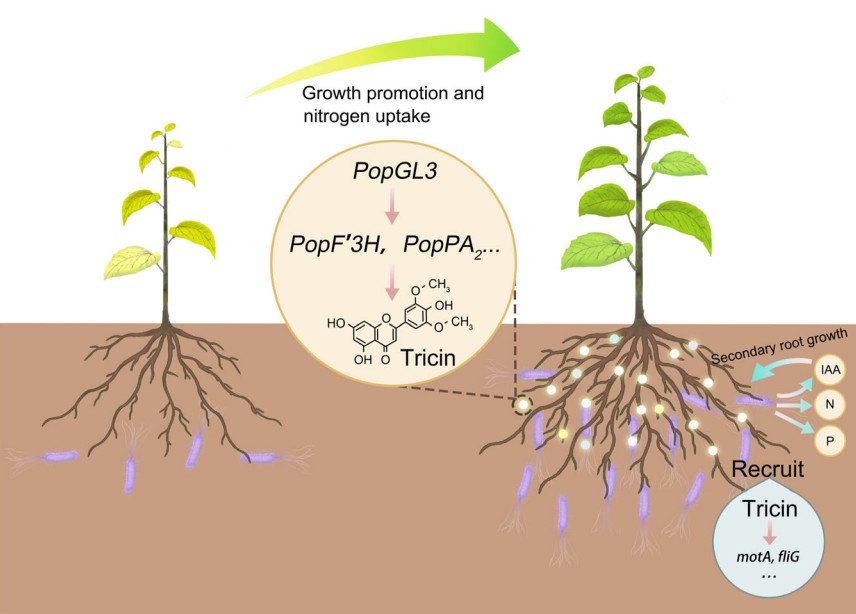

**Fig. 7 | Proposed model for flavone-dependent, microbiota-mediated secondary root formation and plant performance.** In nitrogen-poor soil, poplar roots secreted flavone and recruited *Pseudomonas* to colonize the rhizosphere, thus changing the composition of the rhizosphere microbial community. By secreting auxin IAA, *Pseudomonas* can induce secondary root formation to promote plant growth and nitrogen absorption indirectly and promote plant growth and nitrogen absorption directly through biological nitrogen fixation.

1 "H3-1"), *Tacamahaca* (*P. trichocarpa* M "Pot-M", *P. szechuanica* Z "Psz-Z"), and *Turanga* (*P. euphratica* H "Peu-H"), were examined in this study. Tissue culture plantlets of poplar clones (84K, Psz-Z, Pot-M, and Peu-H) were maintained in our laboratory, while the remaining five species were collected from the GuanXian state-owned *P. tomentosa* forest farm in Shandong Province, China (E: 115°22'8", N: 36°30'54") to acquire sterile monoclonal tissue culture seedlings.

The soils of *Leuce*, *Aigeiros*, and *Tacamahaca* were collected from the plantations of *P. tomentosa*, *P. euramericana*, and *P. simonii* in the GuanXian state-owned forest farm, respectively, while the soil of *Turanga* was collected from the natural forest of *P. euphratica* in Danglang tribe, Aksu, Xinjiang (E: 80°15'18", N: 40°45'39"; Supplementary Table 5). Notably, the poplars in these forests were more than 15 years old and had never been fertilized. At a distance of 2–3 m around the poplars, we collected 10–40 cm of soil after removing the surface 10 cm of soil. Each forest soil is taken from at least five poplars, and all soil samples collected from each site are mixed.

To provide more abundant microorganisms for different genotypes of poplar, four parts of soil were mixed in equal volumes and thoroughly stirred. Subsequently, tissue culture seedlings of the nine poplar species were simultaneously transplanted into the mixed soil for pot experiments, ensuring at least five biological replications per species while randomly situating all poplar samples. They were grown in the same environment (phytochamber conditions: 25 °C; 16 h day/8 h night light cycle) for 3 months, and water poured every 2 days. In order to ensure the normal growth of the seedlings, we used 1 g fertilizer (Huaduo 1, China; N: 20%, P: 20%, and K: 20%) in 1 L sterile water to fertilize after transplanting and watered 300 ml per pot of poplar, not again in the later period.

## Plant measurements and sample collection

On the day of destructive sampling, we examined eleven representative characteristics of nine poplar species, encompassing plant height, ground diameter, shoot biomass, root biomass, root length, leaf length, leaf width, leaf area, leaf number, chlorophyll content, and leaf nitrogen content (Supplementary Data 1). Since the root and

rhizosphere soil of one Peu-H plant were insufficient for subsequent sequencing experiments, the two individual plants were treated as one biological replicate, and the one biological replicate of all phenotypic data for Peu-H was the average of the two biological plants. For other poplar species, three plants of comparable growth were selected as biological replicates. Leaf length and width of the third, fourth, and fifth completely expanded leaves at the top were measured, and the leaf area was calculated by ImageJ (v.1.53q)[60] analysis. Leaves were defined as fully expanded if the leaf length was more than 4.0 cm. Chlorophyll content was determined as the average of 20 measurements with a chlorophyll meter (SPAD-502 Plus, Konica Minolta, Japan) in the middle third of the leaf in the longitudinal direction. The complete above-ground part of the plant was harvested, and fresh biomass was determined.

Each plant's rooting system was subsampled for the assessment of multiple response variables: root metabolomics for flavonoids metabolite analysis, root transcriptome analysis, and rhizosphere soils for 16S rRNA amplicon-based sequencing. In order to focus analyses on the most active roots[61], only fine roots (< 2 mm in diameter) were utilized for these analyses. For metabolomics and transcriptomics, roots were quickly rinsed in deionized water and frozen in liquid nitrogen immediately. For rhizosphere soil, the qualified roots were collected in a 50-ml centrifuge tube containing 30 ml of sterile Phosphate Buffer Saline (PBS) buffer (pH 7.0, per liter: 6.33 g $NaH_2PO_4·H_2O$, 16.5 g $Na_2HPO_4·7H_2O$, and 200 µl Silwet L-77) and stored on ice for further processing in the laboratory. For each plant, bulk soil samples were collected from around the root system and frozen in liquid nitrogen immediately. Rhizosphere samples were extracted from the corresponding root segments. Centrifuge tubes with samples were shaken for 30 min at 50 rpm in a constant-temperature shaker incubator, and the shaking step was repeated twice. Afterwards, rhizosphere samples were centrifuged for 10 min at 4,500 g at 4 °C. The supernatant was then removed, and sterile water was added to resuspend the soil. Finally, the samples were frozen in liquid nitrogen and stored at −80 °C. All samples were stored at −80 °C until processed.

## Soil transplantation and root secretion inoculation assays

To investigate the effect of poplar genotypes on the nutrient acquisition capacity of root microbiota, we examined the growth status of LM50 and Peu-H under nitrogen-poor conditions when subjected to reciprocal root microbiota inoculation. The soil transplantation experiment was carried out as previously described with minor modifications[31]. Firstly, sterile-cultured seedlings of LM50 and Peu-H were transplanted into mixed natural soil for pot cultivation, with a normal growth of 8 weeks. During transplantation, rhizosphere soils from poplar (LM50 and Peu-H) were collected according to the described method, and the rhizosphere soil suspension was reintroduced into the original pots from which the plants were uprooted. Subsequently, LM50, Peu-H, and 84K sterile-cultured seedlings were individually transplanted into pots containing soil (bulk soil and rhizosphere soil) corresponding to either the LM50 or Peu-H genotype, with three replicates for each treatment. As a control, sterile PBS buffer was used to reintroduce sterilized soil. For soil sterilization, a uniform mixture of soil that was used for Peu-H and LM50 cultivation was subjected to high-pressure sterilization at 121 °C for 60 min, followed by cooling at room temperature for at least 24 h. Plant growth was monitored after transplantation, and after a continuous growth period of 8 weeks, both the shoot and root biomass of the poplar were measured.

The basic nutrients in rhizosphere soils and bulk soils of LM50 and Peu-H were measured. The soil chemical traits, including total nitrogen, available phosphorus, and soil organic matter, were measured according to standard protocols[62]. Available potassium was analyzed by the flame photometer method[63].

To demonstrate how different types of poplar root exudates affect poplar growth, we collected LM50 and Peu-H exudates using sterile hydroponics and applied to soil pots. Initially, both LM50 and Peu-H were rooted in solid rooting media, growing roots of 1–2 cm after 10 days. The root systems were carefully removed from the solid medium and placed at the center of a custom-made sterile hydroponic setup (Supplementary Fig. 16) containing the same volume of liquid medium (1/2 MS, no sucrose), with 10 replicates for each genotype. After 20 days, poplar developed hydroponic root systems, root exudates were collected and passed through a 0.2-μm filter. All root exudates of each genotype were mixed separately and stored at −80 °C. Subsequently, sterile-cultured seedlings of 84K and LM50 were transplanted into sterilized soil for pot cultivation, and 20 ml of various types of root secretions were applied to the soil pots every 2 weeks. 1/2 MS sterile (no sucrose) solution of uncultured plants was used as a control. After growing continuously for 8 weeks (LM50 grows for 6 weeks), the shoot and root biomass of the poplar were measured.

## Establishment of the co-expression network

To unveil the intricate genetic regulatory network of poplar mediating microbial composition, the R package cluster (v.2.1.4) with the $k$-means method was used to analyze the co-expression/co-regulation of flavonoids in root samples of nine poplars. Rigorous Pearson's correlation analysis was performed to identify the DEGs and ASVs ($r \geq 0.7$, $P$ values < 0.01) that were significantly associated with each flavonoid using the WGCNA (v.1.71) package.

## Correlation analysis between modules and bacterial families

Correlation analysis of Mantel tests was performed using the vegan (v.2.6-2) package between modules and microbiological families. The networks were visualized using Cytoscape (v.3.9.1)[64].

## Pseudomonads swarming motility assay

The swarming motility experiment followed previously described with minor modifications[37]. Briefly, *Pseudomonas* strains were cultured in liquid King's B (KB) medium for 12 h until reaching a turbidity of 1.0 at 600 nm. Tricin and apigenin were separately prepared as 1 mM and 10 mM stock solutions in DMSO. Flavones at final concentrations of 1, 3, 5, 10, 20, 30, 50, and 100 μM were added to semi-solid Luria-Bertani (LB) medium containing 0.3% (w/v) agar in proportion to the volume. DMSO was added in equal volume to the negative control and then used after condensation. Data were collected at 12 h after inoculation. Each experiment was performed using three independent agar plates.

## Pseudomonads total RNA extraction, cDNA preparation, and quantitative real-time PCR assay

The cultured Pto1 cells were harvested by centrifugation at 4 °C. Total RNA was extracted with Trizol and chloroform. The first-strand cDNAs were synthesized using a GoScript™ Reverse Transcription System (Promega, USA). For real-time PCR analysis, gene-specific primers listed in Supplemental Data 12 were used. Each PCR reaction (20 μl) contained 10 μl of SYBR Premix ExTaq (TaKaRa, Japan), 1 μl of cDNA samples, and 200 nM primers. The reactions were performed on a 7500 Fast Real-Time PCR System (Applied Biosystems, USA). The thermocycling conditions were 95 °C for 30 s and 40 cycles of 95 °C for 5 s, 60 °C for 30 s. Amplification specificity was assessed using a melting curve analysis. The cDNA of the DMSO-treated strain was used as a template for the negative control. The DAPDH gene was used to normalize the data, and at least three biological replicates were performed.

## Growth promotion experiment with pseudomonad inoculation

To explore the potential of pseudomonads in enhancing plant growth and nitrogen uptake under nitrogen-deficient conditions, we inoculated *Pseudomonas* isolates into soil pots and sterile tissue culture plant roots, respectively. For soil potting, natural mixed soil was subjected to high-pressure sterilization at 121 °C for 60 min. The soil was left to cool at room temperature for at least 24 h before sterile-cultured poplar seedlings were transplanted into the sterilized soil for pot cultivation. Wheat (*Triticum aestivum* L.) and radish (*Raphanus sativus* L.) seeds were surface-sterilized with a 4% NaClO solution (v/v) for 15 min, washed in sterile water three times for 15 min, and seeded into the sterilized natural soil for germination. *Pseudomonas* strains (Pto1, Pto5, and Pto10) were cultured in KB liquid medium overnight at 28 °C, then centrifuged and resuspended in 1 × PBS buffer to an OD600 of 0.2. Each strain, or the simple SynComs (Pto1, Pto5, and Pto10), was inoculated onto three seedlings every 2 weeks, with an inactivated heavy suspension as a control. After 8 weeks of growth (wheat and radish were 6 weeks), the shoot and root biomass of plants were measured.

For aseptic tissue culture, the low-nitrogen medium consisted of 1/2 Murashige & Skoog (MS; 300 mg/L NH₄NO₃), while the normal nitrogen medium contained 1/2 MS (1650 mg/L NH₄NO₃). Bacterial cultures (Pto1) were centrifuged, washed, and resuspended in 10 mM MgSO₄, adjusted to OD600 = 1.0. Seedlings of 84K poplar were grown for 10 days until the root length reached 1–2 cm, and the resuspended bacterial solution was dropped on the root. 10 mM MgSO₄ solution was used as a negative control. Shoot and root biomass was determined after 20 days of growth, while SR density was calculated manually by the number of SRs of the main root.

*Arabidopsis thaliana* L. cv Columbia-0 (Col0) was employed as a wild-type (WT). The utilized Arabidopsis mutants included *plt3plt5plt7* and *wox11wox12*[47]. Arabidopsis seeds underwent surface sterilization using 3% (v/v) sodium hypochlorite and were planted on plates containing 1/2 MS medium supplemented with 5 g/L sucrose. After 1 week of growth, transfer to the new medium (CK or 10 μM 2,3,5-Triiodobenzoic acid "TIBA", or 0.1 mg/L indole-3-acetic acid "IAA") and place the above 10 mM MgSO₄ suspensions on the roots of Arabidopsis. The SR numbers were counted after 7–10 days of co-culture.

## Construction of poplar transgenic lines

Total RNA was extracted from the root tissues of 84K. The frozen roots were fully pulverized using liquid nitrogen, and the RNA molecules were then meticulously extracted via the utilization of the RNeasy Mini Kit (Qiagen, Germany) following the instructions provided by the manufacturer. The first-strand cDNA synthesis is described for bacterium samples. The full-length coding sequences (CDS; Supplementary Data 13) of the target genes were cloned from the 84K cDNA library template using the specific primer (Supplementary Data 12), and the CDSs were linked to the pBI121-eGFP overexpression vector by One-step cloning technology. Plasmids containing the correct insertion were introduced into Agrobacterium tumefaciens strain GV3101. The transformation of 84K was carried out following the previously described transformation method for calli[65]. After obtaining antibiotic-resistant seedlings, PCR identification was performed using two primers to determine whether the target genes were inserted into the 84K genome. DNA was extracted using the Plant Genomic DNA Kit (TIANGEN, China) following the instructions provided by the manufacturer. The first set of primers were the sequences at both ends of the polyclonal site inserted by the target gene. To eliminate GV3101 contamination, a second set of primers was designed based on the GV3101 virulence gene *VirD2* sequence (Supplementary Data 12). The first group of primers has DNA bands, and the second group of primers without bands are transgenic lines. At least 10 independent lines were obtained for each gene, and the optimal lines were selected for further investigation based on target gene expression levels and target traits.

## RNA extraction and qPCR from transgenic poplar lines

Fresh roots of transgenic lines and WT were harvested and rapidly ground into a fine powder using liquid nitrogen. Total RNA was extracted as described above. The RNA concentration and the 260/280 nm ratio were determined using NanoDrop 2000. The first-strand cDNA synthesis and real-time PCR analysis as described for bacterium samples. The 18S gene (Primer sequences in Supplemental Data 12) was used to normalize the data, and at least three biological replicates were performed.

## RFP-labeled *Pseudomonas* strain Pto1

The pre-cultured Pto1 was transferred to a new liquid LB medium at a volume ratio of 1:1000. The culture was incubated at 28 °C for 20–24 h to a turbidity of 1.8–2.0 at 600 nm. Then, 1 ml of bacterial solution was collected in a sterile centrifuge tube and centrifuged at 4 °C and 13,500 g for 1 min. After decanting the supernatant, 1 ml of pre-chilled sterile ddH$_2$O was added and gently suspended to wash the cells. Centrifuge for 1 min and then wash again. Centrifuge for 2 min and wash for the last time. Centrifuge for 3 min and pour out the supernatant. Add 100 µl of pre-cooled sterile ddH$_2$O and gently suspend the cells for future use.

Then, 5 µl of the plasmid (carried RFP tag) was added to the competent cells and gently mixed. The mixture was subsequently added to a pre-chilled electroporation cuvette (inner groove width 2 mm) and incubated on ice for 10 min. The MicroPulser electroporator (BIO-RAD, USA) was used for the conversion, with a voltage of 2.5 KV and a time of 5.0 ms. The mixture was rinsed with 500 µl of LB culture solution and transferred to a new centrifuge tube. Water bathed at 28 °C for 1 h, the mixture was evenly spread onto solid LB medium containing antibiotics and incubated at 28 °C. Positive monoclonal clones were identified using PCR and selected for subsequent experiments.

## Colonization of poplar roots by RFP-tagged isolate

To confirm the colonization behavior of Pto1 in the poplar rhizosphere, it was tagged with an RFP tag. Following the hydroponic cultivation method described above, both transgenic and wild-type poplars were grown in a sterile, equal-volume 1/2 MS solution (10 g/L sucrose), with five replicates for each genotype. After 20 days, 200 µl of Pto1-RFP suspensions (resuspended in 10 mM MgSO$_4$; OD600 = 0.5–0.6) was added for co-cultivation. After 2 days, the RFP fluorescence signal was observed using a laser-scanning confocal microscope (Leica TCS SP8, Germany). The fluorescence intensity of the samples was measured by the ImageJ (v.1.53q).

## Pseudomonads root colonization measurements

Following the hydroponic cultivation method described above, both transgenic and wild-type poplars were grown in a sterile, equal-volume 1/2 MS solution (10 g/L sucrose), with five replicates for each genotype. After 20 days, 200 µl of Pto1 suspensions (resuspended in 10 mM MgSO$_4$; OD600 = 0.5–0.6) were added for co-cultivation. After 2 days, the roots soaked in the culture solution were collected, and the surface moisture was drained with sterile paper. The roots were collected in pre-weighed tubes and fresh weight was recorded. Samples were machine-homogenized by TissueLyser (Retsch) at a frequency of 25 Hz. A continuous diluent was prepared with a PBS solution and spread on KB agar plates. After overnight culturing at 28 °C, bacteria colonies were counted and expressed as CFU/mg root fresh weight.

## Statistics and reproducibility

The observed significant differences in the variance of parameters were evaluated using two-sided Student's *t*-tests (for two groups) or one-way ANOVA (for more than two groups). We performed PCA for rhizosphere bacterial community composition, phenotype, and root transcriptome data using the prcomp package in R software (v.4.1.3), and used the hclust package for HCA. Statistical significance of differences between poplar genotypes was assessed by PERMANOVA using the vegan (v.2.6-2) package. No specific statistical methods were used to predetermine the sample size. No data were excluded from analysis in any of the experiments. In the same experiment, all plant materials were exposed to the same growth conditions, and plants of different genotypes or treatments were randomly placed. The investigators were blinded to allocation during all experiments and outcome assessments.

All other methods used in this study are described in the Supporting Information (Supplementary Methods 1–7).

## Reporting summary

Further information on research design is available in the Nature Portfolio Reporting Summary linked to this article.

## Data availability

The raw amplicon data of four poplar sections are publicly accessible in the Genome Sequence Archive of the Beijing Institute of Genomics BIG Data Center, Chinese Academy of Sciences, under CRA015093. The raw root transcriptome data of four poplar sections are publicly available under CRA015096. The raw amplicon data for transgenic poplars and wild types are publicly available under CRA015469. The raw DAP-seq data for the *PopGL3* gene are publicly available under CRA015475. The metabonomics raw data and other research data are available in the figshare database [https://doi.org/10.6084/m9.figshare.26426578]. Source data are provided with this paper.

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

## Acknowledgements

This work was supported by funding from the Fundamental Research Funds for the National Key R&D Program of China [2022YFD2201600 (J.X.), 2023YFD2200203 (J.X.), 2022YFD2200602 (J.X.)]; the Project of the National Natural Science Foundation of China [32371906 (J.X.), 32022057, 31972954 (J.X.), 32170370 (D.Z.)]; Central Universities [QNTD202305 (J.X.)]; Forestry and Grassland Science and Technology Innovation Youth Top Talent Project of China [2020132607 (J.X.)]; the 111 Project [No.B20050 (D.Z.)]. We thank Professors Ton Bisseling and Huchen Li for their careful guidance and constructive ideas on this study. We thank Professors Viola Willemsen and Huchen Li for sharing the *plt3plt5plt7* and *wox11wox12* mutants.

## Author contributions

D.Z. and J.X. designed the experiments. J.W., S.L., H.Z., J.S., and Y.W. collected and analyzed the data. J.W., S.L., H.Z., and Y.W. performed the experiments. J.W. and J.X. wrote the manuscript. H.Z., S.C., S.T., Y.D., L.L., and Z.J. revised the manuscript. D.Z. and J.X. obtained funding and are responsible for this article. All authors read and approved the manuscript. J.W. and S.L. contributed equally to this work.

## Competing interests

The authors declare no competing interests.
