## [Transparent Peer Review file · Nature Communications]

Flavones enrich rhizosphere *Pseudomonas* to enhance nitrogen utilization and secondary root growth in *Populus*

Corresponding Author: Professor Jianbo Xie

Version 0:

Reviewer comments:

Reviewer #2

(Remarks to the Author)

The study by Wu et al. provides some interesting insights into how rhizosphere microbiomes influence Poplar growth via various mechanisms. The authors have conducted an extensive series of experiments and analyses at both the microbial and molecular levels, utilizing datasets from transcriptomes, metabolomes, and microbiomes across various Poplar genotypes to establish a gene-metabolite-microbe network. This method is particularly compelling and underscores the significance of plant-microbe interactions in determining plant performance. However, there are several critical issues that could substantially compromise the quality and integrity of the manuscript.

Major concerns:

- 1) Bioinformatic analysis of amplicon sequencing: I am not familiar with Mothur. Based on my experience with QIIME 2 for processing raw sequencing data, the authors should use paired-end data for profiling bacterial communities due to the 500 bp base pair coverage in the quality control. Typically, forward-end reads yield better quality than reverse-end reads. I am concerned about the decision to not trim reads separately and would appreciate clarification on the quality score thresholds used to filter out low-quality reads. Specifically, how were sequences under 300 bp or over 500 bp treated? Additionally, I noticed that the authors used the QIIME 2 pipeline for taxonomic assignment. Why wasn't QIIME 2 used to process the raw sequencing data as well? Furthermore, I would like to know the identity threshold the authors set for taxonomic classification.
- 2) Enrichment analysis: I believe there may be a critical oversight concerning the genes related to flavonoid metabolism reported as overrepresented in this manuscript. According to the data presented in Supplementary Data 5, the adjusted p-value (even p value) for flavonoid metabolism in the enrichment analysis is over 0.05, with the enrichment factor of 0.62, which these genes related to flavonoid metabolism, I believe, are not overrepresented. This could represent a fatal flaw, as much of the subsequent experiments and analyses rely on this assumption.
- 3) Speculative conclusion: In some sections, the conclusions cannot be fully supported by the results. For examples, the conclusion (Overall, these results showed that flavonoid pathway genes in Leuce roots regulate the secretion of flavones to enrich specific beneficial bacteria, thereby promoting poplar nitrogen metabolism and growth.), is primarily based on correlation analyses. This appears overly definitive and speculative without stronger causal evidence.
- 4) Language and structure: The manuscript is challenging to read in places, likely due to the structure of placing the Results and Discussion sections before the Methods section. This leads to a lot of confusions in the Results and Methods sections. I recommend reorganizing and refining the text to improve readability and coherence. Besides, the discussions should be improved to closely link to results. I noticed that there are a lot of discussion in the Results section, which should be addressed to maintain a clear distinction between these sections. Otherwise, the authors can try to combine Results and Discussion sections.

Minor concerns:

- L. 31 What are these four sections?
- L. 34 What does "robust" mean here? Do you mean more relative abundance or number of taxa?
- L. 35-36 Suggest deleting "agreeing with the 'Matthew effect' on poplar-microbe interaction ". The concept of the 'Matthew effect' may not be familiar to many readers. It would be more appropriate to discuss this concept in the Discussion section, where you can provide a detailed explanation to help readers understand its relevance and implications in the context of your study.
- L. 38 Please spell out the full name of the gene the first time it is mentioned in the text.
- L. 41-44 Please ensure consistent tense.
- L. 85-87. Please clarify the reason why the correlation analysis between genes (are these microbial or plant genes?) and

microbial factors (are these at the community level or functional level?) is insufficient to elucidate regulatory pathways using meta-omics perspectives.

L.74 Overall assembly and functional changes of microbial communities

L.75 as do ...

L.92-94 This sentence is unclear to me.

L. 95 What are the molecular analysis?

L.96 explicit

L.97 It would benefit readers to understand the scope of this study if the authors could provide their hypotheses

L.99 This subtitle is unclear to me. It could benefit from some modifications to improve clarity.

L. 100-104 I understand the intent behind the authors' statements, but these sentences could benefit from further refinement for enhanced clarity and coherence. Besides, could you specify what these eleven phenotypes are?

L.108-109 Could you please indicate where these results are displayed?

Li 115-119 In the Methods section (L.448), it is stated that soil mixed from four parts was used for the pot study. However, this description is somewhat confusing as it suggests that different soil sources might have been used to investigate plant-microbial interactions.

L. 119 "growth inhibition" to what (plant roots? shoots? total biomass?). Please clarify it.

L.125-126 Did you apply root exudates to different genotypes? It appears that the root exudates were only applied to the 84K genotype. Could you please clarify this?

L.127-129 I don't believe this is due to the root microbial community. Rather, the 'rhizosphere microbiome' would presumably be more accurate.

L. 129-131 In addition to microbial communities, soil nutrients play a crucial role in influencing plant growth. I would suggest including a comparison of soil nutrient levels in this study as well.

L. 132 poplar genotypes?

L.138 Please specify the type of diversity. I think here is the alpha diversity.

L. 140-141 This sentence seem discussion.

L. 142 The term 'compositional volatility' used here is somewhat unusual. I assume the authors intended to refer to 'compositional variation' instead.

L. 145-148: What percentage of taxa were identified at the genus level? Is the 16S primer set sufficient for genus-level taxonomy identification (if so, provide the evidence by citing previous references in the method section and briefly mention here)?

L. 146: "exclusively found". In contrast to the taxa exclusively linked with each plant section, what percentage of microbial taxa are shared across these sections? If a significant portion of taxa are shared and their relative abundance remains consistent across plant sections, how can we ascertain that alterations in plant performance are attributed to the unique taxa rather than the shared ones, which might change their functions in response to varying plant genotypes (or soil sources, if not used mixed soil)?

L: 150: "highly abundant" clarify whether it is relative abundance or absolute abundance based here.

L.155-163 This part looks more like discussion rather than results.

L.166- 167 and 171 – 173: Similar to the comments above, it is not known if the changes in plant gene expressions respond to microbial mediated activities or soil chemistry or both. Clarify if the mixed soil was used for this section.

L.168-171 I strongly recommend conducting a statistical analysis, such as PERMANOVA, to determine whether genotypes significantly affect the rhizosphere bacterial community composition, phenotype, and root transcriptomics.

L. 174 Why haven't adjusted p-values been used here? Relying solely on p-values to filter over-represented genes could lead to false positives.

L. 177-180 and 203-204 Again, these are discussion.

L.183 Please explain the rationale behind setting this particular cutoff value of $|\log_2FC|$?

L.188 What do you mean by 'a unified and distinct abundance pattern'? How can these two conditions coexist simultaneously?

L. 208 the elucidation of ...

L.226: Briefly indicate here for what tools were used to identify/predict the function of these OTUs?

L. 254 Could you please display the magnitude of relative abundance in Figure 3E?

L. 264: Were these rhizosphere soil collected from the pots with the mixed soil samples cultivated with Leuce, or from the site where Leuce was initially grown?

L. 265 Do you mean sanger sequencing of each strain?

L.272 may be

L. 279 What is this poplar genotype?

L.291 Why did not you use poplar to directly test the effect of Pto1 on lateral root growth. Arabidopsis and poplar differ significantly in their physiology, ecology, and genetic makeup. What works in Arabidopsis might not directly translate to poplar due to these differences.

L. 306: This does not exclude the possibility that the effect on the IAA pathways could be indirect.

L.307 recruit

L. 313 to 324: Too many assumption here without the direct results to support these assumptions. Suggest only keeping the direct result in the result section and move the assumptions to the discussion section.

Line 325: what was the soil source?

Line 325-347: Briefly explain the exp design for L. 325 to 347 will help us to understand the scope of this section.

L.413-419 Again, concluding that the Pto1 inoculation induces the LR pathway mediated by PLT3PLT5PLT7 through the secretion of IAA based solely on results from Arabidopsis is a substantial extrapolation. This leap assumes a high degree of conservation in the interaction between the bacterial strain and these very different plant species.

L. 429-430: Kindly suggest moving this part to the end of Methods section.

L. 448 What do you mean by four parts of soil? Did you mix all soil samples that collected from each site?
L. 452 Please add the detailed plant incubation conditions and fertilization in the supplementary information and cite it here.
L. 458: No chlorophyll data were presented in the result section?
Figure 3F: what is the unit of X-axis?
Fig. 4 I am quite confused about the statistical analysis in Fig 4B and F. What do you mean by different groups? I also noticed the absence of asterisks above the bars in the figures. What does that statistically mean?

Reviewer #3

(Remarks to the Author)

Reviewer #4

(Remarks to the Author)

The authors presented a comprehensive study that intergrated multi-omics data to investigate the molecular mechanisms underlying root microbiome variation in *Populus* species. Extensive work has been conducted on both the plant and microbe aspects to validate hypotheses generated from correlation analyses. However, it remains unclear from the provided information how the authors selected and prioritized candidates such as flavonoids and *Pseudomonas* for validation. There are also some concerns regarding the methods used in the bioinformatic analysis and experimental procedures (see the detailed comments below). While the authors have performed a huge amount of work to investigate plant-microbe interactions, this reviewer has doubts about the novelty and significance of the main results summarized as the following.

1. This study provided new (?) insights into plant-microbe interactions by using several techniques including 16S amplicon sequencing, plant transcriptome analyses, targeted metabolomics and validating with selected compounds and bacteria.
2. Different genotypes of poplar trees harbour different microbiome compositions (this fact has been already widely described)
3. Different poplar species exude different compounds, of particular interest in this study are flavonoides, why?
4. *Pseudomonas*, among all rhizosphere microorganisms, is selected in this study for further experiments, why?
5. *Pseudomonas* enhanced the plant growth by promoting N uptake and lateral root growth. This fact has been also well-known in existing literatures (e.g., <https://doi.org/10.1111/nph.18199>).
6. In PopGL3-overexpressing poplar, the authors detected more recruitment of *Pseudomonas* based on fluorescence microscopy. However, the provided pictures do not clearly illustrate this increase.

Major comments

• Controls for the soil transplant experiment

(Supplementary Method 12) could you clarify what is meant by “sterilized soil”? Was this soil previously used for *Peu-H* or *LM50* cultivation, or was it the mixed natural soil without prior cultivation. Was the nutrient content in the plant-conditioned soil taken into account? Additionally, was such sterilized soil intended as a control for host-conditioned microbiome or microbiome in general? Furthermore, how were the root exudates processed and stored for the experiment to assess their impact on poplar growth and what was the CK group?

• Microbiome analysis

1. How was the 16S amplicon data normalized? Why OTU clustering was used? Is there a specific reason to choose this method over ASV-based approaches? The evidence is mounting that the latter has several advantages (e.g., <https://doi.org/10.1371/journal.pone.0227434>, <https://doi.org/10.1186/s12864-020-07126-4>).
2. The differential abundance analysis was performed using LEfSe and/or STAMP. It is unclear to this reviewer when and why these methods were applied. Additionally, it is not documented in the method whether any corrections were applied to the obtained P-values. Several tools for differential abundance analysis are available, such as ANCOMBC2.
3. The differential abundance analysis showed that *Pseudomonas*, *Cellvibrio* and *Actinobacteriota* were the most abundant taxa, but the authors only focus on *Pseudomonas*, what about the other taxa?
4. L149-158: The PGPRs were inferred at taxonomic level, but how close were those poplar-associated *Bacillus*, *Pseudomonas* and *Cellvibrio* related to the reported beneficial strains? Both plant growth-promoting and pathogenic bacteria could come from the same genus. More evidence is needed to show the functional potential of the enriched taxa, otherwise the authors should be careful with these statements.

• *Pseudomonas* colonization

Based on the pictures shown in Figure 6, it is very difficult to determine whether *Pseudomonas* colonize more in the overexpression lines. If and how the fluorescence density was normalized? Was the root area/biomass taken into account? Could the authors add extra proves? For example, colonization assays can be performed by counting CFUs on selective media.

• Transcriptome

The authors determined from the transcriptome results that flavonoid biosynthesis was highly expressed compared to what condition exactly? It's unclear which comparisons were made to reach this conclusion. Additionally, besides flavonoid pathways, other pathways were also overexpressed. Why did the authors choose flavonoids for further analysis and validation? Many other genes, such as the ones involved in terpenoids and phenylpropanoids biosynthesis, were also enriched (Figure S5). What is the significance and relevance of these compounds? Were they also exuded by different

poplar species at varying abundance levels?

- Metabolome

As mentioned earlier, it is unclear to the reviewer why the authors chose targeted metabolomics when many other pathways were enriched.

Minor comments

L142: volatility?

L196: which are these 159 candidate genes? Data S7 gave an overview of the co-response clusters, but flavonoid biosynthetic genes/pathways are not highlighted.

L292: why switch to Arabidopsis? Why the plt3plt5plt7 triple mutant and wox11wox12 double mutant were selected?

L321: (constitutive) expression level of PopGL3 in the PopCHS4-OE line was not shown.

L377: genotype-cried microbiota >> genotype-specific?

L388: PopGL3 and PopCHS transgenic plants >> PopGL3-OE and PopCHS4-OE plants?

L395: in nutrient-deficient conditions >> under nutrient-deficient conditions

L398: core microbiome was discussed here, but no related results were shown in this manuscript. Does this statement stand true also for poplar and is Pseudomonas core in this context?

L437: typo? Pse-Z >> Pse-Z, Ptr-M>>Ptr-M?

L445-L447: this sentence needs to be rephrased.

L452: which fertilizer? At what concentration/composition?

L477: why only fine roots were utilized and what is meant by "responses"?

L482: by the laboratory >> in the laboratory; in each plant >> for each plant

Co-reviewer statement: I co-reviewed this manuscript with one of the reviewers who provided the listed reports. This is part of the Nature Communications initiative to facilitate training in peer review and to provide appropriate recognition for Early Career Researchers who co-review manuscripts.

Reviewer #5

(Remarks to the Author)

The authors state that the overall goal of this manuscript is to generate a comprehensive gene-metabolite-microbe network. After reading this manuscript many times, it is unclear to me if that is the actual goal. I am impressed by the large dataset the authors are presenting and see the utility in such a dataset. There have been significant resources put into this manuscript in terms of both experiments and data collected. Unfortunately, as written, it is unclear what the major questions are within this project, what hypotheses the authors are testing, and how this dataset is being leveraged to answer them. Thorough revisions are needed before publication. I recommend the authors develop a conceptual framework or diagram that highlights the questions they are asking and demonstrates the utility of each individual experiment and dataset. It was very challenging to follow the flow and rationale of each individual experiment within the methods and results. A solid conceptual framework can be an organizing theme throughout the whole manuscript. Additionally, there are numerous grammatical errors throughout that need addressing. Detailed comments/suggestions/questions from each section are below.

Abstract:

Within the abstract, the authors mention the 'Matthew Effect' on poplar - microbe interactions. This concept is not picked up within the introduction and was confusing.

Introduction:

Line 90 - This paragraph is confusing and needs clarification.

I believe the authors mean 'In this study', then on line 92 the statement 'the regulating chains that how' is confusing. It is unclear what the authors mean by this. What are the key questions and hypotheses driving this research? The authors do not lay the groundwork to set up the story they are telling throughout the rest of the paper.

Methods - There are numerous areas of the methods that require clarification and restructuring. It is unclear how all the methods/analyses fit together and inform a coherent story. Looking through the supplementary material, it appears there are significantly more experiments/measurements than were included in the main document. It is unclear to me how the data chosen inform a coherent story. I suggest major restructuring throughout. A targeted question or set of questions should be established within the introduction and followed throughout the manuscript.

1. Within the methods, it is confusing how and why the soils were collected. I assume the authors wished to provide diverse natural inoculum for the plants to select from. Did the authors add inoculum to a standard mix to control for variation in nutrients as opposed to microbes?
2. It is also unclear from the methods how the experiment flowed. Reading the results, it appears numerous experiments were conducted sequentially.
3. A schematic highlighting the plants selected including the origin, and the soils collected would be helpful.
4. Was there a 'no inoculum' control? From the figure, it looks like there was a sterilized soil control, but there is no mention of that in the methods.
5. Why weren't all the biological reps used for the plant measurement data? Why was Peu-H treated differently and averaged?
6. Why the focus on pseudomonads and swarming?

Results -

1. 'robust species become more robust'. How do you define robustness? This section title is confusing.
2. It is not surprising that different poplar species exhibit different traits.
3. LM50 showed growth inhibition when transplanted into Peu-H grown soil, and Peu-H showed increased growth in LM50 soil. Is this driven by variation in soil nutrients? How were inoculations done? Was it whole soil or a subset?
4. Line 142, what is 'compositional volatility'?
5. Line 160 - the authors haven't shown specific taxa increase plant growth or performance. I would soften this statement.
6. Did anything other than flavonoids show up as significant in your results?
7. Functional profiles inferred from 16S data are not trustworthy. I do not recommend their use.

Discussion - As with the intro and methods, the discussion is quite scattered and unfocused. A good conceptual framework or model could really help streamline this manuscript.

1. The 'cry for help' theory is first introduced in the discussion. This theory could be introduced in the introduction and help unify the sections of the paper.
2. Line 377 - what is 'genotype-cried'?
3. The 'matthew effect' is introduced in the discussion with limited explanation or follow through.

Version 1:

Reviewer comments:

Reviewer #4

(Remarks to the Author)

Reviewer #5

(Remarks to the Author)

The authors have made significant improvements to this manuscript and have addressed the majority of my concerns. I still struggle a bit with how all the datasets fit together to tell a cohesive story and recommend a bit of restructuring and a useful conceptual diagram to solidify how everything flows together.

Reviewer #6

(Remarks to the Author)

I was invited to review the authors' responses to the comments from Reviewer #2. There are 4 major comments (comments 1-4) and 53 minor comments (comments 5-57). The authors have sufficiently responded to the minor comments, so I am going to focus on the major comments. The authors' responses to comments 2-4 are also sufficient but I have one additional concern regarding comment 1 as follows.

The authors' responses to Reviewer #2's major comment 1 are not sufficient. In major comment 1, the reviewer proposed a different option for bioinformatic analysis of amplicon sequencing. The authors responded that they have changed their bioinformatics analysis method of amplicon sequencing and that they have modified the Methods section. However, it is unclear how the change of the bioinformatics analysis affects the results. The authors should clarify whether the results have been affected and if so, how the results change.

Response to the editor and reviewers (NCOMMS-24-15351)

Many thanks to the editor and reviewers for their constructive and helpful comments. We have thoroughly revised the manuscript in response to these comments. We greatly appreciate the time that went into providing such detailed reviews and hope we have addressed the concerns sufficiently. We now present our point-by-point responses to the reviewers:

#####Reviewer #1:#####

Reviewer #1 (Response):

The authors presented a comprehensive study that intergrated multi-omics data to investigate the molecular mechanisms underlying root microbiome variation in Populus species. Extensive work has been conducted on both the plant and microbe aspects to validate hypotheses generated from correlation analyses. However, it remains unclear from the provided information how the authors selected and prioritized candiates such as flavonoids and Pseudomonas for validation. There are also some concerns regarding the methods used in the bioinformatic analysis and experimental procedures (see the detailed comments below). While the authors have performed a huge amount of work to investigate plant-microbe interactions, this reviewer has doubts about the novelty and significance of the main results summarized as the following.

Our response: We sincerely appreciate Reviewer 1 for the time and effort in reviewing the manuscript. Your insightful feedback has been instrumental in enhancing the quality and rigor of our work. We have made point-by-point responses to these comments and made corresponding revisions to the manuscript as below. We hope that these concerns have been adequately addressed.

1. Reviewer #1: This study provided new (?) insights into plant-microbe interations by using several techniques including 16S amplicon sequencing, plant transcriptome analyses, targeted metabolomics and validating with selcted compounds and bacteria.

Our response: We thank Reviewer 1 for this kind suggestion. Our laboratory has long focused on the genetic mechanisms underlying stress resistance and wood formation in the perennial model plant poplar. Building on these studies, we have discovered that the stress resistance and growth of poplar may be closely related to the rhizosphere microbiome. Therefore, the research direction of stress resistance in perennial poplar has been further extended to the interaction between host and microorganism.

The link of plant genes to the rhizomicrobiome is complicated, necessitating consideration of ‘genes-downstream, genes-metabolite, or other signalling substances-microbe’ pathways¹. Notably, the gene impact on the rhizomicrobiome is often accomplished through metabolites. Therefore, how to combine host-centric transcriptomics and metabolomics with microbe-centric microbiomics is crucial for unraveling the complex mechanisms of plant gene-microbe interactions^{2,3}. Microbiome-Wide Association Studies (MWAS) well attach phytomics to microbiomics and demonstrate that host genomics does influence the composition of the microbiome. MWAS has been used to reveal the genetic mechanisms of the assembled rhizosphere microbiome of plants such as Arabidopsis, sorghum, barley, and maize⁴⁻⁷. In addition, Weighted Gene Co-Expression Network Analysis (WGCNA) has also been used to jointly analyze the transcriptomics and microbiomics of *Glycyrrhiza uralensis* Fisch⁸. However, analysis of the correlation between the two omics of host genes and microbial factors cannot effectively reveal the role of metabolites or other signaling substances in the pathways of plant regulation of microbial structure. Among them, MWAS requires extensive genomic data from numerous samples, and the analysis based on limited samples limits more detailed regulatory pathway analysis. Notably, these studies rarely combine molecular and microbial experiments to further elucidate the genetic mechanisms of host microbial recruitment and the effects of specific microbes on host growth or stress resistance.

Through combined molecular techniques, such as CRISPR/Cas system (e.g., Cas9), and gene overexpression, and validation of selected compounds and bacteria, Song et al. (2021) showed that FERONIA restricts *Pseudomonas* in the Arabidopsis

rhizosphere microbiome by regulating reactive oxygen species⁹. Yang et al. (2023) showed that after tomato roots were infected by *Ralstonia solanacearum*, the RIN transcriptional factor would change the secretion of riboflavin and 3-hydroxyflavone, and the disease was suppressed by recruiting *Streptomyces*¹⁰. However, these studies have not been able to effectively identify plant genes that regulate the rhizosphere microbiota on a large scale.

Our study generated a multivariate dataset of poplar transcriptome, metabolome, microbiome, and growth traits under low nitrogen conditions, and based on the *k*-means clustering algorithm and Pearson's correlation analysis using metabolites as media, a comprehensive gene-metabolite-microbe co-expression network of different poplar genotypes was established for the first time. Subsequently, subnetwork analysis broke down the wall of the four-way relationship between plant gene-metabolite-microbial-phenotype, thus identifying the links between plant gene expression, metabolite accumulation, growth phenotype, and microbiome function. Additionally, using the co-expression network and refined validation of selected genes, compounds, and bacteria, we revealed for the first time that the *PopGL3* gene in poplar mediates the enrichment of *Pseudomonas* by regulating tricetin synthesis, thereby promoting lateral root growth and nitrogen uptake in poplar. These findings elucidate the genetic mechanism underlying the rapid growth of *Leuce* from a host-microbe interaction perspective, validating the network's effectiveness and practicality. Our network can serve as a valuable resource for other researchers, and our approach offers new insights for future studies. We hope that this concern has been adequately addressed.

References

1. Liu, Q., Cheng, L., Nian, H., Jin, J. & Lian, T. Linking plant functional genes to rhizosphere microbes: a review. *Plant Biotechnol. J.* **21**, 902-917 (2023).
2. Stringlis, I. A. et al. MYB72-dependent coumarin exudation shapes root microbiome assembly to promote plant health. *Proc. Natl. Acad. Sci. U. S. A.* **115**, (2018).

3. Danxia He, S. K. S., Li, P., Richa, K. J. I. V. & Rafael J. L. Morcillo, P. W. P. A. Flavonoid-attracted *Aeromonas* sp. from the Arabidopsis root microbiome enhances plant dehydration resistance. *Isme J.* **16**, 2633 (2022).
4. Jason G. Wallace, K. A. K. L. Quantitative Genetics of the Maize Leaf Microbiome. *Phytobiomes Journal.* **2**, 208-224 (2018).
5. Bergelson, J., Mittelstrass, J. & Horton, M. W. Characterizing both bacteria and fungi improves understanding of the Arabidopsis root microbiome. *Sci. Rep.* **9**, 24 (2019).
6. Deng, S. et al. Genome wide association study reveals plant loci controlling heritability of the rhizosphere microbiome. *Isme J.* **15**, 3181-3194 (2021).
7. Escudero-Martinez, C. et al. Identifying plant genes shaping microbiota composition in the barley rhizosphere. *Nat. Commun.* **13**, 3443 (2022).
8. Zhong, C. et al. Multi-omics profiling reveals comprehensive microbe-plant-metabolite regulation patterns for medicinal plant *Glycyrrhiza uralensis* Fisch. *Plant Biotechnol. J.* **20**, 1874-1887 (2022).
9. Song, Y. et al. FERONIA restricts *Pseudomonas* in the rhizosphere microbiome via regulation of reactive oxygen species. *Nat. Plants.* **7**, 644-654 (2021).
10. Yang, K. et al. RIN enhances plant disease resistance via root exudate-mediated assembly of disease-suppressive rhizosphere microbiota. *Mol. Plant.* **16**, 1379-1395 (2023).

2. Reviewer #1: Different genotypes of poplar trees harbour different microbiome compositions (this fact has been already widely described)

Our response: We thank Reviewer 1 for this kind suggestion. As the reviewer noted, numerous studies have demonstrated that different genotypes (and even different sexes) of poplars harbor different microbiome compositions. For instance, Yan et al. (2024) showed that the rhizosphere bacterial community of poplar was significantly dominated by host genetic effects and soil organic carbon and carbon to nitrogen ratio (C/N)¹. Zhao et al. (2023) reported that poplar sex influenced the response and recovery of fungal and bacterial communities to drought, and sexual dimorphism was

detected in the regulation of rhizospheric nutrient reactions². These studies have merely revealed compositional differences in the microbiomes of different poplar genotypes or explored the microbiome's functions. However, the biological pathways of host regulation of microbial structure are difficult to accurately elucidate through these studies. Our research reveals the rhizosphere microbial composition among different poplar sections, integrating transcriptomic, metabolomic, and phenotypic data to elucidate the gene-metabolite-microbe-phenotype regulatory mechanisms governing the composition of the rhizosphere microbiome (Fig. 2, 3). Through multi-omic and molecular genetics approaches, such as microbial diversity, gene function, and metabolite quantification analyses, we elucidated the genetic mechanism by which the *PopGL3* gene in poplar recruits *Pseudomonas* by regulating the synthesis of triclin. Furthermore, through strain isolation, nitrogen isotope labeling, and the inoculation of isolated strains, we verified the impact of specific microbes on host growth and nutrient uptake. **In conclusion, our results emphasize the genetic mechanisms by which poplar genotypes mediate microbial community responses to low nitrogen stress through flavonoid regulation.**

In addition, driven by the consideration of poplar industry development, we elucidate the rhizosphere microbial structures of different poplar sections and the impact of genotype-specific microbes on poplar growth and nutrient uptake. This can help facilitate an understanding of the combination of poplar breeding with microbe resource utilization and provide a theoretical basis for scientific advancement to support the development of the forestry industry. We hope that this concern has been adequately addressed.

References

1. Yan, K. et al. Rare and abundant bacterial communities in poplar rhizosphere soils respond differently to genetic effects. *Sci. Total Environ.* **908**, 168216 (2024).
2. Zhao, Y., Chen, L., Chen, Y., Yang, Q. & Liu, M. Plant sexual variation modulates

rhizospheric nutrient processes through the soil microbiome response to drought and rewetting in *Populus cathayana*. *Biol. Fertil. Soils*. **59**, 571-587 (2023).

3. Reviewer #1: Different poplar species exudate different compounds, of particular interest in this study are flavonoides, why?

Our response: We thank Reviewer 1 for this comment. As the reviewer stated, different genotypes of poplars exudate different compounds^{1,2}. In previous studies, flavonoids, terpenoids, strigolactones, and coumarins in root plant secretions have been shown to affect plant fitness and regulate the assembly of specific microbial taxa in the rhizosphere³⁻⁶. In the study, the differentially expressed genes of the four sections significantly enriched flavonoid synthesis and terpenoid synthesis pathways.

Flavonoids are one of the most studied classes of such metabolites, regulating both plant development and the interaction with commensal microbes³. In addition, the importance of flavonoids in the process of *Rhizobium* nodulation in plants belonging to the Fabaceae family is best known⁷, and the role of flavonoids in the recruitment of beneficial microorganisms under biological stress has been extensively studied^{8,9}. However, other mechanisms by which flavonoids determine interaction with microorganisms involved in plant abiotic stress responses have remained partially elusive. We have highlighted these descriptions in the Introduction section of the revised manuscript.

For instance:

Introduction

Flavonoids are one of the most studied classes of such metabolites, regulating both plant development and the interaction with commensal microbes. Root secretion of flavonoids occurs frequently under biotic stress and is involved in promoting microbial colonization during stress generation. Infection of part of the tomato (*Solanum lycopersicum*) root system with *Ralstonia solanacearum* changes numerous root exudates and involves disease suppression via the recruitment of disease-suppressing *Streptomyces* for colonization, which was associated with

increased exudation of 3-hydroxyflavone. Under abiotic stress, flavonoid production is often elevated in plants. However, our knowledge of how flavonoid-mediated plant-microbe interactions may improve plant resistance to abiotic stresses remains elusive.

As mentioned earlier, our laboratory has also carried out long-term research on the genetic mechanisms of poplar stress resistance and wood formation. The secretion of flavonoids occurs frequently under biotic or abiotic stress, which is the main defense substance of plants external factors such as light and temperature. Lu et al. (2021) conducted a multi-omics analysis of 300 *Populus tomentosa* individuals and revealed the genetic architecture of flavonoid metabolites in poplar leaves¹⁰. Lignin is one of the important components of xylem, and lignins and flavonoids are synthesized through the same metabolic pathway (ko00940: Phenylpropanoid biosynthesis) and interconnected with the synthesis of flavonoids. Other laboratory members have resolved the genetic mechanism of flavonoid synthesis in the poplar population's xylem (unpublished). Based on the research of the poplar population, we carried out research on the genetic mechanism of flavonoid synthesis in poplar roots, and further extended it to the influence of the interaction between flavonoids and microbes on poplar growth and stress resistance. In conclusion, based on the above reasons, flavonoids were chosen to perform further analyses. We hope that this concern has been adequately addressed.

References

1. Veach, A. M. et al. Rhizosphere microbiomes diverge among *Populus trichocarpa* plant-host genotypes and chemotypes, but it depends on soil origin. *Microbiome*. **7**, (2019).
2. Li, Z., Rubert-Nason, K. F., Jamieson, M. A., Raffa, K. F. & Lindroth, R. L. Root Secondary Metabolites in *Populus tremuloides*: Effects of Simulated Climate Warming, Defoliation, and Genotype. *J. Chem. Ecol.* **47**, 313-321 (2021).

3. Wang, L. et al. Multifaceted roles of flavonoids mediating plant-microbe interactions. *Microbiome*. **10**, (2022).
4. Stringlis, I. A. et al. MYB72-dependent coumarin exudation shapes root microbiome assembly to promote plant health. *Proc. Natl. Acad. Sci. U. S. A.* **115**, E5213-E5222 (2018).
5. Yu, P. et al. Plant flavones enrich rhizosphere Oxalobacteraceae to improve maize performance under nitrogen deprivation. *Nat. Plants*. **7**, 481-499 (2021).
6. Zhong, Y. et al. Root-secreted bitter triterpene modulates the rhizosphere microbiota to improve plant fitness. *Nat. Plants*. **8**, 887-896 (2022).
7. Mathesius, U. The role of the flavonoid pathway in *Medicago truncatula* in root nodule formation. A review. *The Model Legume Medicago truncatula*. 2019. pp. 434-438.
8. Yang, K. et al. RIN enhances plant disease resistance via root exudate-mediated assembly of disease-suppressive rhizosphere microbiota. *Mol. Plant*. **16**, 1379-1395 (2023).
9. Murata, K. et al. Natural variation in the expression and catalytic activity of a naringenin 7-O-methyltransferase influences antifungal defenses in diverse rice cultivars. *The Plant Journal*. **101**, 1103-1117 (2020).
10. Lu, W. et al. Multi-omics analysis provides insights into genetic architecture of flavonoid metabolites in *Populus*. *Ind. Crop. Prod.* **168**, 113612 (2021).

4. Reviewer #1: *Pseudomonas*, among all rhizosphere microorganisms, is selected in this study for further experiments, why?

Our response: We thank Reviewer 1 for this kind comment. In bulk soil, the relative abundances of Pseudomonadaceae and *Pseudomonas* in the *Leuce* were 0.86% and 0.86%, respectively, while the relative abundances of them were enriched to 13.78% and 13.78%, respectively, in the rhizosphere soil of the *Leuce*. Notably, **Pseudomonadaceae is the most abundant specific family in the *Leuce* rhizosphere, and *Pseudomonas* is the most abundant specific genus.** We have highlighted these descriptions in the revised manuscript and changed the figure (Supplementary Fig. 8).

For instance:

Pseudomonadaceae, taxa enriched from 0.86% in the bulk soil to the highest abundance in the *Leuce* rhizosphere (13.78%), was specifically enriched in *Leuce* and correlated with poplar growth (ANOVA, P -values < 0.01; Fig. 3E and Supplementary Fig. 8A, B).

Supplementary Fig. 8 Relative abundance of *Pseudomonadaceae* and *Pseudomonas*. (A) Relative abundance of the top 10 rhizosphere microbial families in bulk soil and rhizosphere soil. The numbers are the relative abundances of *Pseudomonadaceae*. The abundance of *Pseudomonadaceae* (B) and *Pseudomonas* (C) in the rhizosphere of four sections. Different letters indicate significantly different groups (One-way ANOVA, P -values < 0.05).

To further enrich putative regulating networks, we specifically focused on Cluster IV, which was abundant in the *Leuce*, as the sections demonstrated the best growth performances. In cluster IV, ***Pseudomonadaceae* showed the highest number of flavonoid-associated ASVs, totaling 58 (all ASVs belong to the *Pseudomonas*)**, while the second most numerous family, *Xanthomonadaceae*, had only 15 ASVs. We have added these new descriptions to the revised manuscript and provide a new figure (Fig. 3C).

For instance:

The accumulation of flavones was associated with ASVs from Pseudomonadaceae, Burkholderiaceae, Cellvibrionaceae, and Xanthomonadaceae, among which Pseudomonadaceae had the highest number of ASVs (58; all ASVs belong to the *Pseudomonas*).

Fig. 3 (C) Correlation network of flavonoid-related genes, flavones, and ASVs in Cluster IV. Red color indicates genes detected in the network. Highly correlated associations ($r \geq 0.7$, P -values < 0.01) were present.

Moreover, our established gene-metabolite-microbe co-expression network serves as an effective tool for identifying beneficial microbes. In addition to Pseudomonadaceae, we also found that among the ASVs associated with flavonoid accumulation, the most abundant families were Burkholderiaceae, Cellvibrionaceae, and Xanthomonadaceae. During bacterial isolation, we utilized both general and selective media to separate the target microbes. However, we were unable to isolate any Xanthomonadaceae or Cellvibrionaceae strains. Instead, eleven *Pseudomonas* strains and two Burkholderiaceae strains were isolated. **We failed to perform the genetic analysis of the two Burkholderiaceae strains based on the red fluorescent protein-based reporter systems.** Consequently, we could only select *Pseudomonas* strains for further experiments. We hope that this concern has been adequately addressed.

5. Reviewer #1: *Pseudomonas* enhanced the plant growth by promoting N uptake and lateral root growth. This fact has been also well-known in existing literatures (e.g., <https://doi.org/10.1111/nph.18199>).

Our response: We thank Reviewer 1 for this kind comment. As the reviewer stated, previous studies have shown that *Pseudomonas* has beneficial potentials in nitrogen fixation, phosphorus solubilization, secretion of growth hormones, and antimicrobial activities^{1,2}, and could promote the growth of plant lateral roots. *Pseudomonas mosselii* strain 923 in the rice rhizosphere has been shown to specifically inhibit the growth of plant bacterial pathogens *Xanthomonas* species and the fungal pathogen *Magnaporthe oryzae* by secreting Pseudoiodinine¹. *Pseudomonas* strain CM11 in the *Castanea mollissima* rhizosphere can specifically induce Arabidopsis lateral root growth³. Several stains in the family Pseudomonadaceae, including *Pseudomonas stutzeri* A1501, *Pseudomonas stutzeri* DSM4166, *Pseudomonas szotifigens* 6HT33bT, and *Pseudomonas* sp. strain K1 can fix nitrogen from the air⁴. Although the beneficial functions of *Pseudomonas* have been demonstrated, it has not yet been reported whether and how the perennial woody model plant poplar recruits *Pseudomonas* for root colonization. The genetic pathways involved in poplar's recruitment of *Pseudomonas* and how the *Pseudomonas* strains influence poplar growth and resistance are intriguing questions that warrant further investigation.

In this study, the primary objective is to elucidate the regulating mechanism mediating the composition of rhizosphere microbial communities through metabolites, and to assess the impact of specific microbes on poplar growth and adaptation to low-nitrogen environments. Using our established gene-metabolite-microbe co-expression network, we demonstrate that poplars drive *Pseudomonas* colonization by regulating triclin synthesis through *PopGL3*, thereby enhancing nitrogen uptake and lateral root growth under low-nitrogen conditions. We hope that this concern has been adequately addressed.

References

1. Yang, R. et al. The natural pyrazolotriazine pseudoiodinine from *Pseudomonas*

- mosselii* 923 inhibits plant bacterial and fungal pathogens. *Nat. Commun.* **14**, 734 (2023).
2. Pérez Rodriguez, M. M. et al. Halotolerant native bacteria *Enterobacter* 64S1 and *Pseudomonas* 42P4 alleviate saline stress in tomato plants. *Physiol. Plant.* **174**, e13742 (2022).
 3. Li, Q. et al. Plant growth-promoting rhizobacterium *Pseudomonas* sp. CM11 specifically induces lateral roots. *New Phytol.* **235**, 1575-1588 (2022).
 4. Sanow, S. et al. Molecular Mechanisms of *Pseudomonas*-Assisted Plant Nitrogen Uptake: Opportunities for Modern Agriculture. *Mol. Plant. Microbe. Interact.* **36**, 536-548 (2023).

6. Reviewer #1: In *PopGL3*-overexpressing poplar, the authors detected more recruitment of *Pseudomonas* based on fluorescence microscopy. However, the provided pictures do not clearly illustrate this increase.

Our response: We thank Reviewer 1 for this kind comment. We have provided a new figure to the revised manuscript (Fig. 6A).

To validate the colonization amount of pseudomonads (Pto1) in the roots of wild-type and *PopGL3*-overexpressing poplar again, we conducted plate counting experiments, and the results again proved that the colonization amount of pseudomonads (Pto1) in the roots of *PopGL3*-overexpressing poplar was higher. We have added these new results to the revised manuscript and provided a new figure (Fig. 6C). Moreover, we provided more details in the method section.

For instance:

Additionally, the colony forming units (CFUs) statistics further confirm this conclusion (*P*-values < 0.01; Fig. 6C).

Fig. 6 *PopGL3* recruits pseudomonads to colonize poplar roots. (A) Confocal fluorescence imaging of RFP-tagged Pto1 colonizing poplar roots of different genotypes. (B) The fluorescence intensity of RFP in poplar roots of different genotypes. The fluorescence intensity of the samples was measured by the ImageJ. (C) Colonization amount of different genotypes of poplar roots with Pto1. Colony forming units (CFUs) were quantified per fresh weight of roots. Asterisks indicate significant differences between different groups (Student's t-test, *** P -values < 0.001).

Major comments

• Controls for the soil transplant experiment

7. Reviewer #1: (Supplementary Method 12) could you clarify what is meant by “sterilized soil”? Was this soil previously used for *Peu-H* or *LM50* cultivation, or was it the mixed natural soil without prior cultivation. Was the nutrient content in the plant-conditioned soil taken into account? Additionally, was such sterilized soil intended as a control for host-conditioned microbiome or microbiome in general?

Furthermore, how were the root exudates processed and stored for the experiment to assess their impact on poplar growth and what was the CK group?

Our response: We thank Reviewer 1 for the constructive suggestions. Sterilized soil is prepared by autoclaving natural mixed soil at 121 °C for 60 minutes, followed by cooling at room temperature for at least 24 hours. The sterilized soil is a uniform mixture of soil that was used for Peu-H and LM50 cultivation. We have added more details in the Methods section.

For instance:

For soil sterilization, a uniform mixture of soil that was used for Peu-H and LM50 cultivation was subjected to high-pressure sterilization at 121 °C for 60 min, followed by cooling at room temperature for at least 24 hours.

--We have examined the total nitrogen, available phosphorus, available potassium, and organic matter contents of Peu-H and LM50 bulk soils and rhizosphere soils, respectively. Compared with bulk soil, the total nitrogen and available potassium in rhizosphere soil increased significantly, while the available phosphorus decreased significantly. However, there is no significant difference between the bulk soil and bulk soil, rhizosphere soil and rhizosphere soil between the two poplar species. This excluded the effect of soil nutrients on the results of soil transplantation. We have added the relevant descriptions to the revised manuscript.

For instance:

Notably, there were no significant differences in nutrients in soils (bulk soils or rhizosphere soils) previously grown with LM50 and Peu-H (ANOVA; Supplementary Table 1),

Supplementary Table 1. Basic chemical properties of soils.

	SOM (g/kg)	TN (%)	AP (mg/kg)	AK (mg/kg)
LM50 bulk soil	23.6 b	0.070 b	11.9 a	139.1 b

Peu-H bulk soil	23.9 ab	0.072 b	12.2 a	142.7 b
LM50 rhizosphere soil	26.2 a	0.084 a	10.6 b	153.2 a
Peu-H rhizosphere soil	25.7 a	0.087 a	10.1 b	157.4 a

Note: TN: total nitrogen, SOM: soil organic matter, AP: available phosphorus, AK: available potassium. $n = 3$ biologically independent samples. Different letters indicate significantly different groups (One-way ANOVA, P -values < 0.05).

--The sterilized soil is a uniform mixture of soil that was used for Peu-H and LM50 cultivation, thus serving as a control for the microbiome of host-conditioned. We have added the relevant descriptions in Methods section.

For instance:

For soil sterilization, a uniform mixture of soil that was used for Peu-H and LM50 cultivation was subjected to high-pressure sterilization at 121 °C for 60 min, followed by cooling at room temperature for at least 24 hours.

--The root exudates were collected by aseptic hydroponics. Peu-H and LM50 were grown in a sterile, equal-volume sucrose-free 1/2 MS solution, with ten replicates for each genotype. After 20 days, root exudates were collected, passed through a 0.2- μ m filter, and stored at -80 °C. A 1/2 MS sterile solution of uncultured plants was used as a control. We have added the relevant descriptions in Methods section.

For instance:

All root exudates of each genotype were mixed separately and stored at -80 °C. Subsequently, sterile-cultured seedlings of 84K and LM50 were transplanted into sterilized soil for pot cultivation, and 20 ml of various types of root secretions were applied to the soil pots every two weeks. 1/2 MS sterile (no sucrose) solution of uncultured plants was used as a control.

• ***Microbiome analysis***

8. Reviewer #1: How was the 16S amplicon data normalized? Why OTU clustering was used? Is there a specific reason to choose this method over ASV-based approaches? The evidence is mounting that the latter has several advantages (e.g., <https://doi.org/10.1371/journal.pone.0227434>, <https://doi.org/10.1186/s12864-020-07126-4>).

Our response: We thank Reviewer 1 for the constructive suggestions. It's no doubt that high taxonomic resolution based on Amplicon Sequence Variants (ASV) analyses will facilitate the detection of fine-scale variations in microbiome composition¹. According to the suggestions of Reviewer 1, we reanalyzed the microbiome compositional profile and performed the subsequent analyses based on ASV methods. As a result, we identified more bacterial taxa based on the ASV analyses, which may be due to the improved resolution of the ASV methods. Compared with the results based on the OTU methods, a similar pattern of results was obtained because our study mainly focuses on the dominant taxa. We redraw Fig. 1C, D, G, Fig. 2, Fig. 3 C-F, Fig. 5 H, I, Supplementary Fig. 3, Supplementary Fig. 5C, Supplementary Fig. 6A, C, Supplementary Fig. 8, and Supplementary Fig. 12 D, E (revised manuscript) according to the new analysis results.

For instance:

Results (A) in the previous manuscript based on OTU analyses and results (B) in the current manuscript based on ASV analyses (Relative abundance for the most abundant bacterial phyla).

Results in the previous manuscript based on OTU analyses and results in the current manuscript based on ASV analyses (Mean Shannon's diversity across the bulk soil and rhizosphere soil of different poplar species; B: bulk soil, R: rhizosphere soil).

References

- Callahan, B. J., McMurdie, P. J. & Holmes, S. P. Exact sequence variants should replace operational taxonomic units in marker-gene data analysis. *Isme J.* **11**, 2639-2643 (2017).

9. Reviewer #1: The differential abundance analysis was performed using LEfSe and/or STAMP. It is unclear to this reviewer when and why these methods were applied. Additionally, it is not documented in the method whether any corrections were applied to the obtained P-values. Several tools for differential abundance analysis are available, such as ANCOMBC2.

Our response: We thank Reviewer 1 for the constructive suggestions. In previous studies, Linear discrimination analysis effect size (LEfSe) and STAMP were used more frequently in the difference analysis of different groups of microorganisms¹⁻⁴. In this study, STAMP could not well display the results of microbial difference analysis between more than two groups in figures. LEfSe was used to analyze the differentially taxonomical features among microbial communities from different sections. The statistical analyses were based on the FDR-corrected Kruskal-Wallis test. The significant taxonomical biomarkers were selected with a FDR adjusted *P*-value < 0.05

and logarithmic LDA score > 2 . However, compared with LEfSe, STAMP could clearly show the differences in analysis results between the two groups of microorganisms. STAMP was used to analyze the differences between overexpression plants and wild-type poplar rhizosphere microbial communities. Statistical analyses were based on Welch's test. There's no correction here. We have added the relevant description to the revised manuscript.

For instance:

The compositional variation of the rhizosphere microbiome among the four sections is driven by significant shifts in the relative abundance of 22 specific bacterial phyla (Linear discriminate analysis effect size 'LEfSe', LDA score > 2 , Kruskal-Wallis test, FDR adjusted P -values < 0.05 ; Fig. 1C, Supplementary Fig. 3B, and Supplementary Data 2).

The results indicated that transgenic (*PopGL3-OE* and *PopCHS4-OE*) plants reshaped the rhizosphere microbial composition and significantly enriched *Pseudomonas* (P -values < 0.01 ; Fig. 5H, I, Supplementary Fig. 12D, E, by STAMP; Supplementary Fig. 13, Supplementary Data 11, by ANCOM-BC2).

Supporting Information

Linear discriminate analysis effect size (LEfSe) was used to analyze the differentially taxonomical features among microbial communities from different sections⁴. Statistical analyses were based on the FDR-corrected Kruskal-Wallis test. The significant taxonomical biomarkers were selected with FDR adjusted P -value < 0.05 and logarithmic LDA score > 2 . STAMP (v.2.1.3) software was used to analyze the differences between overexpression plants (*PopCHS4-OE* and *PopGL3-OE*) and wild-type poplar rhizosphere microbial communities. Rhizosphere microbial differences among overexpression plants (*PopCHS4-OE* and *PopGL3-OE*) and wild-type were re-analyzed using ANCOM-BC2 to increase the reliability of the results.

Additionally, we re-analyzed the abundance differences of rhizosphere microbiomes between transgenic poplars and wild types by using ANCOM-BC2⁵. The results showed that the rhizosphere of *PopGL3-OE* poplar lines was significantly enriched with *Pseudomonas* (P -values < 0.05), while *PopCHS4-OE* poplar lines were not. We have added the new results to the revised manuscript (Supplementary Fig. 13).

For instance:

The results indicated that transgenic (*PopGL3-OE* and *PopCHS4-OE*) plants reshaped the rhizosphere microbial composition and significantly enriched *Pseudomonas* (P -values < 0.01 ; Fig. 5H, I, Supplementary Fig. 12D, E, by STAMP; Supplementary Fig. 13, Supplementary Data 11, by ANCOM-BC2).

Supplementary Fig. 13 Abundance differences of rhizosphere microbiome between transgenic poplars and wild type. Compared with the wild types, the rhizosphere of *PopGL3-OE* poplar lines was significantly enriched with *Pseudomonas* (P -values < 0.05), while *PopCHS4-OE* poplar lines were not. Abundance differences (P -values < 0.05 ; ANCOM-BC2) between *PopCHS4-OE* poplar lines and WT

rhizosphere microbiomes at the phylum level (A) and genera level (C). Abundance differences (P -values < 0.05 ; ANCOM-BC2) between *PopGL3-OE* poplar lines and WT rhizosphere microbiomes at the phylum level (B) and genera level (D).

References

1. Yang, K. et al. RIN enhances plant disease resistance via root exudate-mediated assembly of disease-suppressive rhizosphere microbiota. *Mol. Plant.* **16**, 1379-1395 (2023).
2. Guo, Q., Liu, L., Liu, J., Korpelainen, H. & Li, C. Plant sex affects plant-microbiome assemblies of dioecious *Populus cathayana* trees under different soil nitrogen conditions. *Microbiome.* **10**, (2022).
3. Yin, J. et al. Heritability of tomato rhizobacteria resistant to *Ralstonia solanacearum*. *Microbiome.* **10**, 227 (2022).
4. Yu, P. et al. Plant flavones enrich rhizosphere Oxalobacteraceae to improve maize performance under nitrogen deprivation. *Nat. Plants.* **7**, 481-499 (2021).
5. Lin, H. & Peddada, S. D. Multigroup analysis of compositions of microbiomes with covariate adjustments and repeated measures. *Nat. Methods.* **21**, 83-91 (2024).

10. Reviewer #1: The differential abundance analysis showed that *Pseudomonas*, *Cellvibrio* and *Actinobacteriota* were the most abundant taxa, but the authors only focus on *Pseudomonas*, what about the other taxa?

Our response: We thank Reviewer 1 for the kind comment. In differential abundance analysis of the rhizosphere microbiomes of the four poplar sections, we identified *Pseudomonas*, *Cellvibrio*, and *Actinobacteriota* as specific microbial taxa for *Leuce* or *Turanga*, respectively. Subsequently, to further elucidate the relationships between gene expression, metabolite accumulation, and microbial enrichment, we constructed a gene-metabolite-microbe co-expression network comprising six clusters. We specifically focused on Cluster IV, which was abundant in the *Leuce*, as the sections demonstrated the best growth performances. This does not imply that other clusters

lack scientific research value; rather, this manuscript specifically focuses on Cluster IV. The other clusters present good subjects for future research. However, most ASVs of microorganisms, such as Actinobacteriota, are present in other clusters, which provide good subjects for future research.

In Cluster IV, Pseudomonadaceae exhibits the highest number of ASVs (all ASVs belong to the *Pseudomonas*), totaling 58, and Cellvibrionaceae has only 5 ASVs (all ASVs belong to the *Cellvibrio*; Fig. 3 C). This suggests that Pseudomonadaceae has more potential associations with flavones, and based on this observation, we began to focus on Pseudomonadaceae. During bacterial isolation, we utilized both general and selective media to separate the target microbes. However, we were unable to isolate any *Cellvibrio* strains. Instead, eleven *Pseudomonas* strains were isolated. We then chose *Pseudomonas* for future experiments. We have added the relevant description to the revised manuscript.

For instance:

The accumulation of flavones was associated with ASVs from Pseudomonadaceae, Burkholderiaceae, Cellvibrionaceae, and Xanthomonadaceae, among which Pseudomonadaceae had the highest number of ASVs (58; all ASVs belong to the *Pseudomonas*). Notably, Pseudomonadaceae was among the top families showing the highest correlation with flavone modules and flavonoid-related gene modules (P -values < 0.01 ; Fig. 3D). Pseudomonadaceae, taxa enriched from 0.86% in the bulk soil to the highest abundance in the *Leuce* rhizosphere (13.78%), was specifically enriched in *Leuce* and correlated with poplar growth (ANOVA, P -values < 0.01 ; Fig. 3E and Supplementary Fig. 8A, B).

Fig. 3 (C) Correlation network of flavonoid-related genes, flavones, and ASVs in Cluster IV. Red color indicates genes detected in the network. Highly correlated associations ($r \geq 0.7$, P -values < 0.01) were present.

11. Reviewer #1: L149-158: *The PGPRs were inferred at taxonomic level, but how close were those poplar-associated Bacillus, Pseudomonas and Cellvibrio related to the reported beneficial strains? Both plant growth-promoting and pathogenic bacteria could come from the same genus. More evidence is needed to show the functional potential of the enriched taxa, otherwise the authors should be careful with these statements.*

Our response: We thank Reviewer 1 for the constructive suggestions. As the reviewer stated, we need more evidence on the relationship between the enriched taxa and the reported beneficial strains. According to your suggestion, we have deleted the definition of PGPRs and modified the relevant content.

• ***Pseudomonas* colonization**

12. Reviewer #1: *Based on the pictures shown in Figure 6, it is very difficult to determine whether Pseudomonas colonize more in the overexpression lines. If and how the fluorescence density was normalized? Was the root area/biomass taken into account? Could the authors add extra proves? For example, colonization assays can be performed by counting CFUs on selective media.*

Our response: We thank Reviewer 1 for the constructive suggestions. We performed local magnification on the original image, and the enlarged image clearly showed the difference in pseudomonad (Pto1) colonization in poplar roots. We have added this new figure to the revised manuscript (Fig. 6A).

--The fluorescence intensity of the samples was measured by ImageJ (v.1.53q). We have added the measure method to Methods section in the revised manuscript.

For instance:

The fluorescence intensity of the samples was measured by the ImageJ (v.1.53q)

--According to your suggestion, in order to validate the colonization amount of pseudomonads (Pto1) in the roots of wild-type, *PopCHS4-OE* and *PopGL3-OE* poplars again, we conducted plate counting experiments. **Bacteria colonies were counted and expressed as CFU/mg root fresh weight.** The results again proved that the colonization amount of pseudomonads (Pto1) in the roots of *PopCHS4-OE* and *PopGL3-OE* poplars were higher. We have added these new results and details to the revised manuscript and provided a new figure (Fig. 6C).

For instance:

Additionally, the colony forming units (CFUs) statistics further confirm this conclusion (P -values < 0.01; Fig. 6C).

Fig. 6 *PopGL3* recruits pseudomonads to colonize poplar roots. (A) Confocal fluorescence imaging of RFP-tagged Pto1 colonizing poplar roots of different genotypes. (B) The fluorescence intensity of RFP in poplar roots of different genotypes. The fluorescence intensity of the samples was measured by the ImageJ. (C) Colonization amount of different genotypes of poplar roots with Pto1. Colony forming units (CFUs) were quantified per fresh weight of roots. Asterisks indicate significant differences between different groups (Student's t-test, *** P -values < 0.001).

Materials and methods

Following the hydroponic cultivation method described above, both transgenic and wild-type poplars were grown in a sterile, equal-volume 1/2 MS solution (10 g/L sucrose), with three replicates for each genotype. After 20 days, 200 μ l of Pto1 suspensions (resuspended in 10 mM $MgSO_4$; OD600 = 0.5–0.6) were added for co-cultivation. After two days, the roots soaked in the culture solution were collected, and the surface moisture was drained with sterile paper. The roots were collected in

pre-weighed tubes and fresh weight was recorded. Samples were machine-homogenized by TissueLyser (Retsch) at a frequency of 25 Hz. A continuous diluent was prepared with a PBS solution and spread on KB agar plates. After overnight culturing at 28 °C, bacteria colonies were counted and expressed as CFU/mg root fresh weight.

• *Transcriptome*

13. Reviewer #1: *The authors determined from the transcriptome results that flavonoid biosynthesis was highly expressed compared to what condition exactly? It's unclear which comparisons were made to reach this conclusion. Additionally, besides flavonoid pathways, other pathways were also overexpressed. Why did the authors choose flavonoids for further analysis and validation? Many other genes, such as the ones involved in terpenoids and phenylpropanoids biosynthesis, were also enriched (Figure S5). What is the significance and relevance of these compounds? Were they also exuded by different poplar species at varying abundance levels?*

Our response: We thank Reviewer 1 for the constructive suggestions. We are sorry for the unclear use of “overrepresented”. We modified “overrepresented” to “significantly enriched”.

Original manuscript:

Functional enrichment analyses revealed that differentially expressed genes (DEGs; $|\log_2FC| > 1$, P -values < 0.05 ; Supplementary Fig. 4 and Supplementary Data 4) were overrepresented in functions related to flavonoid metabolism (P -values < 0.05 ; Supplementary Fig. 5 and Supplementary Data 5).

Current manuscript:

Moreover, functional enrichment analyses revealed that differentially expressed genes (DEGs; $|\log_2FC| \geq 1$, FDR adjusted P -values < 0.05 ; Supplementary Data 5) among

the four sections were **significantly enriched** in functions related to flavonoid metabolism (P -values < 0.05; Supplementary Fig. 4 and Supplementary Data 6).

--As the reviewer stated, different genotypes of poplars exude different compounds^{1,2}. In previous studies, flavonoids, terpenoids, strigolactones, and coumarins in root plant secretions have been shown to affect plant fitness and regulate the assembly of specific microbial taxa in the rhizosphere³⁻⁶. In the study, the differentially expressed genes of the four sections significantly enriched flavonoid synthesis and terpenoid synthesis pathways (Figure S4, original manuscript is Figure S5). Flavonoids are one of the most studied classes of such metabolites, regulating both plant development and the interaction with commensal microbes³. In addition, the importance of flavonoids in the process of *Rhizobium* nodulation in plants belonging to the Fabaceae family is best known⁷, and the role of flavonoids in the recruitment of beneficial microorganisms under biological stress has been extensively studied^{8,9}. However, other mechanisms by which flavonoids determine interaction with microorganisms involved in plant abiotic stress responses have remained partially elusive. We have highlighted these descriptions in the Introduction section of the revised manuscript.

For instance:

Flavonoids are one of the most studied classes of such metabolites, regulating both plant development and the interaction with commensal microbes. Root secretion of flavonoids occurs frequently under biotic stress and is involved in promoting microbial colonization during stress generation. Infection of part of the tomato (*Solanum lycopersicum*) root system with *Ralstonia solanacearum* changes numerous root exudates and involves disease suppression via the recruitment of disease-suppressing *Streptomyces* for colonization, which was associated with increased exudation of 3-hydroxyflavone. Under abiotic stress, flavonoid production is often elevated in plants. However, our knowledge of how flavonoid-mediated plant-microbe interactions may improve plant resistance to abiotic stresses remains elusive.

As mentioned earlier, our laboratory has also carried out long-term research on the genetic mechanisms of poplar stress resistance and wood formation. The secretion of flavonoids occurs frequently under biotic or abiotic stress, which is the main defense substance of plants external factors such as light and temperature. Lu et al. (2021) analyzed 300 *Populus tomentosa* individuals using multi-omics analysis and revealed the genetic architecture of flavonoid metabolites in poplars¹⁰. Lignin is one of the important components of xylem, and lignins and flavonoids are synthesized through the same metabolic pathway (ko00940: Phenylpropanoid biosynthesis) and interconnected with the synthesis of flavonoids. We have resolved the genetic mechanism of flavonoid synthesis associated with the lignin traits in the poplar population (unpublished). Based on the research of the poplar population, we carried out research on the genetic mechanism of flavonoid synthesis in poplar roots and further extended it to the influence of the interaction between flavonoids and microbes on poplar growth and stress resistance. In conclusion, based on the above reasons, we chose flavonoids as the research object.

--In this study, the differentially expressed genes of the four poplar sections were significantly enriched in various metabolic pathways, such as the pathways of flavonoid synthesis, phenylpropanoid biosynthesis, phenylalanine metabolism, and terpenoid synthesis. Phenylpropane biosynthesis and phenylalanine metabolism pathways are important pathways which associated with the synthesis of plant secondary metabolites, lignin and flavonoids can be synthesized, **and the flavonoid biosynthesis pathway belongs to the phenylpropane biosynthesis and phenylalanine metabolism pathways.** Terpenoids play a role in plant-microbial interactions, and we focus on flavonoids, terpenoids and other compounds that have not been extensively studied. The composition of these compounds in the root exudates across different poplar species and their interactions with microbes may provide new insights for future studies. We hope that these concerns have been adequately addressed.

References

1. Veach, A. M. et al. Rhizosphere microbiomes diverge among *Populus trichocarpa* plant-host genotypes and chemotypes, but it depends on soil origin. *Microbiome*. **7**, (2019).
2. Li, Z., Rubert-Nason, K. F., Jamieson, M. A., Raffa, K. F. & Lindroth, R. L. Root Secondary Metabolites in *Populus tremuloides*: Effects of Simulated Climate Warming, Defoliation, and Genotype. *J. Chem. Ecol.* **47**, 313-321 (2021).
3. Wang, L. et al. Multifaceted roles of flavonoids mediating plant-microbe interactions. *Microbiome*. **10**, (2022).
4. Stringlis, I. A. et al. MYB72-dependent coumarin exudation shapes root microbiome assembly to promote plant health. *Proc. Natl. Acad. Sci. U. S. A.* **115**, E5213-E5222 (2018).
5. Yu, P. et al. Plant flavones enrich rhizosphere Oxalobacteraceae to improve maize performance under nitrogen deprivation. *Nat. Plants*. **7**, 481-499 (2021).
6. Zhong, Y. et al. Root-secreted bitter triterpene modulates the rhizosphere microbiota to improve plant fitness. *Nat. Plants*. **8**, 887-896 (2022).
7. Mathesius, U. The role of the flavonoid pathway in *Medicago truncatula* in root nodule formation. A review. *The Model Legume Medicago truncatula*. 2019. pp. 434-438.
8. Yang, K. et al. RIN enhances plant disease resistance via root exudate-mediated assembly of disease-suppressive rhizosphere microbiota. *Mol. Plant*. **16**, 1379-1395 (2023).
9. Murata, K. et al. Natural variation in the expression and catalytic activity of a naringenin 7-O-methyltransferase influences antifungal defenses in diverse rice cultivars. *The Plant Journal*. **101**, 1103-1117 (2020).
10. Lu, W. et al. Multi-omics analysis provides insights into genetic architecture of flavonoid metabolites in *Populus*. *Ind. Crop. Prod.* **168**, 113612 (2021).

• *Metabolome*

14. Reviewer #1: *As mentioned earlier, it is unclear to the reviewer why the authors*

chose targeted metabolomics when many other pathways were enriched.

Our response: We thank Reviewer 1 for this kind comment. We explained the reasons for choosing flavonoids as subjects of study above and hope we have adequately addressed the concerns.

Minor comments

15. Reviewer #1: L142: volatility?

Our response: Thank you for your careful review. We modified “compositional volatility” to “compositional variation”.

Original manuscript:

The compositional volatility of the rhizosphere microbiome among the four sections is driven by significant shifts in the relative abundance of nineteen specific bacterial phyla (LDA score > 2, Kruskal-Wallis test, FDR adjusted *P*-values < 0.05; Fig. 1C, Supplementary Fig. 3B, and Supplementary Data 3).

Current manuscript:

The **compositional variation** of the rhizosphere microbiome among the four sections is driven by significant shifts in the relative abundance of 22 specific bacterial phyla (Linear discriminate analysis effect size ‘LEfSe’, LDA score > 2, Kruskal-Wallis test, FDR adjusted *P*-values < 0.05; Fig. 1C, Supplementary Fig. 3B, and Supplementary Data 2).

16. Reviewer #1: L196: which are these 159 candidate genes? Data S7 gave an overview of the co-response clusters, but flavonoid biosynthetic genes/pathways are not highlighted.

Our response: We thank Reviewer 1 for this constructive comment. **We've taken these genes to Supplementary Data 8 and described them.** As a result of our correction of the differential expression gene analysis, the number of genes strongly associated with flavonoids changed, among which flavonoid biosynthesis pathway enzyme genes became 147.

17. Reviewer #1: L292: why switch to Arabidopsis? Why the *plt3plt5plt7* triple mutant and *wox11wox12* double mutant were selected?

Our response: We thank Reviewer 1 for the kind comments. In the process of verifying the effects of pseudomonads on poplar growth, we found that the Pto1 strain could significantly promote secondary roots (SRs) development in poplar. Dicotyledons secondary roots can be further divided into lateral roots (LRs) and adventitious lateral roots (adLRs)¹. Secondary roots are morphologically undistinguishable from the primary root, but the genetic regulation mechanisms of these two kinds of secondary roots are different. We want to further investigate what kind of secondary root is induced by the Pto1 strain and which genetic pathways are involved, and obtaining the mutant with adventitious lateral root deletion and the mutant with lateral root deletion is the key to carrying out this study.

Auxin can activate the plant-specific AP2 transcription factors PLETHORA (PLT) 3, PLT 5, and PLT 7²⁻⁴. These PLTs are essential for subsequent lateral root initiation, emergence, and spacing of lateral root primordia. In response to wounding or other environmental signals, plants activate the adventitious rooting pathway to produce another class of secondary roots, adventitious lateral roots or adventitious roots (AR)^{5,6}. Adventitious lateral roots are not primed in the root meristem, but are initiated *de novo* between lateral roots in a non-acropetal sequence. In adventitious lateral root, auxin promotes the fate transition of the *de novo* root founder pericycle cells by upregulating the expression of *WUSCHEL-RELATED HOMEODOMAIN 11* (WOX11) and the partially redundant WOX12 to initiate adventitious root primordia⁶⁻⁸.

We conducted homologous gene comparisons for PLT 3, PLT 5, PLT 7 and WOX11, WOX12 in 84K, respectively. Because 84K is a hybrid of *Populus alba* and *P. glandulosa*, it has two sets of chromosomes sequences. A total of 20 homologous genes of PLT 3 or PLT 5 or PLT 7 and 7 homologous genes of WOX11 or WOX12 were detected. Due to the large number of homologous genes, the function of genes is redundant, and it is difficult to obtain the ideal 84K poplar mutant by knocking out a

few genes. Additionally, our laboratory has academic exchanges with Professor Ton Bisseling, Professor Viola Willemsen, and Professor Huchen Li. During their discussion on this study, we knew that their lab possess the *plt3plt5plt7* Arabidopsis triple mutant, which exhibits compromised lateral root formation, and the *wox11wox12* Arabidopsis double mutant, displaying deficiencies in adventitious lateral roots and adventitious root formation, and they generously shared the mutant seeds with us⁹. Therefore, we used Arabidopsis mutants to investigate the genetic pathway of PTO1-induced secondary rooting growth. We have already thanked Professors Viola Willemsen and Huchen Li in the Acknowledgments section and we ignored it in the previous manuscript.

For instance:

We thank Professors Ton Bisseling and Huchen Li for their careful guidance and constructive ideas on this study. We thank Professors Viola Willemsen and Huchen Li for sharing the *plt3plt5plt7* and *wox11wox12* mutants.

Phylogenetic analysis of PLT 3, PLT 5, and PLT 7 homologous genes in the 84K

genome.

Phylogenetic analysis of WOX11 and WOX12 homologous genes in the 84K genome.

Arabidopsis and poplar differ significantly in their physiology, ecology, and genetic makeup, and bacterial strains may not have a high degree of conservation in across plant species. The results obtained in Arabidopsis may not apply to poplars. Therefore, we revised the relevant description to emphasize the results obtained in Arabidopsis and modified the title. We have carefully reviewed the revised manuscript and hope we have addressed the concerns sufficiently.

For instance:

Original manuscript:

Flavones enrich rhizosphere *Pseudomonas* to enhance nitrogen utilization and lateral root growth in *Populus*

Current manuscript:

Flavones enrich rhizosphere *Pseudomonas* to enhance nitrogen utilization and **secondary root** growth in *Populus*

References

1. Li, Q. et al. Plant growth-promoting rhizobacterium *Pseudomonas* sp. CM11 specifically induces lateral roots. *New Phytol.* **235**, 1575-1588 (2022).
2. Du, Y. & Scheres, B. Lateral root formation and the multiple roles of auxin. *J. Exp. Bot.* **69**, 155-167 (2018).
3. Du, Y. & Scheres, B. PLETHORA transcription factors orchestrate de novo organ patterning during Arabidopsis lateral root outgrowth. *Proceedings of the National Academy of Sciences.* **114**, 11709-11714 (2017).
4. Hofhuis, H. et al. Phyllotaxis and Rhizotaxis in Arabidopsis Are Modified by Three PLETHORA Transcription Factors. *Curr. Biol.* **23**, 956-962 (2013).
5. Karlova, R., Boer, D., Hayes, S. & Testerink, C. Root plasticity under abiotic stress. *Plant Physiology.* **187**, 1057-1070 (2021).
6. Sheng, L. et al. Non-canonical *WOXII*-mediated root branching contributes to plasticity in Arabidopsis root system architecture. *Development.*, (2017).
7. Liu, J. et al. *WOXII* and *I2* Are Involved in the First-Step Cell Fate Transition during de Novo Root Organogenesis in Arabidopsis. *The Plant Cell.* **26**, 1081-1093 (2014).
8. Xu, L. *De novo* root regeneration from leaf explants: wounding, auxin, and cell fate transition. *Curr. Opin. Plant Biol.*, 125-126 (2017).
9. Li, Q. et al. Plant growth-promoting rhizobacterium *Pseudomonas* sp. CM11 specifically induces lateral roots. *New Phytol.* **235**, 1575-1588 (2022).

18. Reviewer #1: L321: (constitutive) expression level of PopGL3 in the PopCHS4-OE line was not shown.

Our response: We thank Reviewer 1 for the constructive suggestion. We have conducted additional experiments to detect the expression levels of *PopGL3* in the *POPCHS4-OE* lines and *PopCHS4* in the *POPGL3-OE* lines. The results showed that the constitutive expression of *PopCHS4* had no significant effect on the transcription of *PopGL3*, but the constitutive expression of *PopGL3* could activate the transcription

of *PopCHS4*. We have added new results to the revised manuscript.

For instance:

The constitutive expression of *PopGL3* or *PopCHS4* could activate the transcription of *PopF3'H* and *flavone synthase (PopFNS)* and release more tricetin in the rhizosphere of the *PopCHS4-OE* (chalcone synthase catalyzes the first committed step of the multi-branched flavonoid pathway) and *PopGL3-OE* lines (P -values < 0.01; Fig. 5 E, F).

Fig. 5 (E) Gene relative expression levels of WT, *PopCHS4-OE*, and *PopGL3-OE* poplar lines in unsterilized nitrogen-poor soil. $n = 3$ biologically independent samples.

19. Reviewer #1: L377: genotype-cried microbiota >> genotype-specific?

Our response: We thank Reviewer 1 for the constructive suggestion. We modified “genotype-cried” to “genotype-specific”.

Original manuscript:

Consistent with these observations, the biomass of LM50 (*Leuce*) was greater than that of Peu-H (*Turanga*) in sterile and nitrogen-poor soil, while the biomass of both increased in unsterilized soil, but the difference between them expanded further,

indicating that genotype-cried microbiota exerts varying degrees of positive feedback on poplar growth.

Current manuscript:

We found that LM50 (*Leuce*) produced 7.37 g more stem biomass than Peu-H (*Turanga*) in sterile and nitrogen-poor soil. In the unsterilized, nitrogen-deficient soil, this difference widened to 8.29 g (Peu-H grown soil) and 10.26 g (LM50 grown soil), respectively. Notably, there were no significant differences in nutrients between soils (bulk soils or rhizosphere soils) previously grown with LM50 and Peu-H; root exudates had no significant effect on poplar growth; and bulk soil microbial diversity showed no significant variation. These results indicate that **genotype-specific** microbiota exerts varying degrees of positive feedback on poplar growth.

20. Reviewer #1: L388: *PopGL3* and *PopCHS* transgenic plants >> *PopGL3-OE* and *PopCHS4-OE* plants?

Our response: We thank Reviewer 1 for the constructive suggestions. We modified “*PopGL3* and *PopCHS* transgenic plants” to “*PopGL3-OE* and *PopCHS4-OE* plants”.

Original manuscript:

Rhizosphere microbiome analyses of *PopGL3* and *PopCHS* transgenic plants, integrating the metabolite profiles of root extracts and secretions, demonstrate the causal role of *PopGL3* in tricetin secretion and recruitment of *Pseudomonas*.

Current manuscript:

Rhizosphere microbiome analyses of *PopGL3-OE* and *PopCHS4-OE* plants, integrating the metabolite profiles of root extracts and secretions, demonstrate the causal role of *PopGL3* in tricetin secretion and recruitment of *Pseudomonas*.

21. Reviewer #1: L395: in nutrient-deficient conditions >> under nutrient-deficient conditions

Our response: We thank Reviewer 1 for the constructive suggestion. We modified “in nutrient-deficient conditions” to “under nutrient-deficient conditions”.

Original manuscript:

These findings suggest that the poplar *GL3* gene regulates tricetin synthesis and secretion to call for pseudomonad colonization to help it grow and nitrogen absorption in nutrition-deficient conditions.

Current manuscript:

In summary, our findings suggest that the poplar *GL3* gene regulates tricetin synthesis and secretion to call for pseudomonad colonization to help it grow and nitrogen absorption **under** nutrition-deficient conditions (Fig. 7).

22. Reviewer #1: L398: core microbiome was discussed here, but no related results were shown in this manuscript. Does this statement stand true also for poplar and is *Pseudomonas* core in this context?

Our response: We thank Reviewer 1 for this kind comment. The concept of the core microbiome has been removed.

23. Reviewer #1: L437: typo? *Pse-Z* >> *Psz-Z*, *Ptr-M*>>*Pto-M*?

Our response: We thank Reviewer 1 for these constructive suggestions. *Pse-Z* is *Psz-Z*, and *Ptr-M* is *Pot-M*. We have modified them.

Original manuscript:

Tissue culture plantlets of poplar clones (84K, *Pse-Z*, *Ptr-M*, and *Peu-H*) were maintained in our laboratory, while the remaining five species were collected from the GuanXian state-owned *P. tomentosa* forest farm in Shandong Province, China (E: 115°22'8", N: 36°30'54") to acquire sterile monoclonal tissue culture seedlings.

Current manuscript:

Tissue culture plantlets of poplar clones (84K, ***Psz-Z***, ***Pot-M***, and *Peu-H*) were maintained in our laboratory, while the remaining five species were collected from the GuanXian state-owned *P. tomentosa* forest farm in Shandong Province, China (E: 115°22'8", N: 36°30'54") to acquire sterile monoclonal tissue culture seedlings.

24. Reviewer #1: L445-L447: this sentence needs to be rephrased.

Our response: We thank Reviewer 1 for the constructive suggestion. These two sentences have been modified.

Original manuscript:

Notably, these forests, aged over 15 years, have never received fertilization. Removed the surface 10 cm of soil, and at the surface 10–40 cm of soil around the poplars was collected, with each soil taken from at least five poplars.

Current manuscript:

Notably, the poplars in these forests were more than 15 years old and had never been fertilized. At a distance of 2-3 m around the poplars, we collected 10-40 cm of soil after removing the surface 10 cm of soil. Each forest soil is taken from at least five poplars, and all soil samples collected from each site are mixed.

25. Reviewer #1: L452: which fertilizer? At what concentration/composition?

Our response: We thank Reviewer 1 for the constructive suggestions. The fertilizer we used was Huaduo 1 (Huaduo, China). Its composition is N: 20%, P: 20%, and K: 20%. The concentration we used was 1 g of fertilizer to 1 L of sterile water, and each pot of poplar was watered with 300 ml. In order to ensure the normal growth of the seedlings, this fertilization was only watered once when they were transplanted. We added relevant descriptions to the Methods.

For instance:

In order to ensure the normal growth of the seedlings, we used 1 g fertilizer (Huaduo 1, China; N: 20%, P: 20%, and K: 20%) in 1 L sterile water to fertilize after transplanting and watered 300 ml per pot of poplar, not again in the later period.

26. Reviewer #1: L477: why only fine roots were utilized and what is meant by “responses”?

Our response: We thank Reviewer 1 for the constructive suggestion. In this study, we pay more attention to the fine roots. In several previous studies, fine roots play important roles in resource acquisition and nutrient cycling, are an essential component of plant species^{1,2}. Relatively high abundance of bacterial taxa was detected in fine root may due to a high level of metabolic activity³. In *Populus*, a fine root is conventionally defined as less than 2 mm in diameter⁴. In addition, in previous studies on the interaction between poplar and microorganisms, only fine roots were collected as research objects^{5,6}. Therefore, we only collected the fine roots (< 2 mm in diameter) here for further analysis. We have added relevant descriptions to the Methods section.

For instance:

In order to focus analyses on the most active roots, only fine roots (< 2 mm in diameter) were utilized for these analyses.

-- We modified “responses” to “analyses”. These analyses refer to “root metabolomics for flavonoids metabolite analysis, root transcriptome analysis, and rhizosphere soils for 16S rRNA amplicon-based sequencing” as described in the previous sentence.

Original manuscript:

Each plant’s rooting system was subsampled for assessment of multiple response variables: root metabolomics for flavonoids metabolite analysis, root transcriptome analysis, and rhizosphere soils for 16S rRNA amplicon-based sequencing. Only fine roots (< 2 mm in diameter) were utilized for these responses.

Current manuscript:

Each plant’s rooting system was subsampled for assessment of multiple response variables: root metabolomics for flavonoids metabolite analysis, root transcriptome analysis, and rhizosphere soils for 16S rRNA amplicon-based sequencing. In order to focus analyses on the most active roots, only fine roots (< 2 mm in diameter) were utilized for these **analyses**.

References

1. Picon-Cochard, C. et al. Effect of species, root branching order and season on the root traits of 13 perennial grass species. *Plant Soil*. **353**, 47-57 (2012).
2. M. Begon, A. H. F. The Ecology of Root Lifespan. *Adv. Ecol. Res.* **27**, 1-60 (1997).
3. Pervaiz, Z. H. et al. Root microbiome changes with root branching order and root chemistry in peach rhizosphere soil. *Rhizosphere*. **16**, 100249 (2020).
4. Berhongaray, G., Janssens, I. A., King, J. S. & Ceulemans, R. Fine root biomass and turnover of two fast-growing poplar genotypes in a short-rotation coppice culture. *Plant Soil*. **373**, 269-283 (2013).
5. Veach, A. M. et al. Rhizosphere microbiomes diverge among *Populus trichocarpa* plant-host genotypes and chemotypes, but it depends on soil origin. *Microbiome*. **7**, (2019).
6. Schadt, C. et al. An integrated metagenomic, metabolomic and transcriptomic survey of *Populus* across genotypes and environments. *Sci. Data*. **11**, 339 (2024).

27. Reviewer #1: L482: by the laboratory >> in the laboratory; in each plant >> for each plant

Our response: We thank Reviewer 1 for the constructive suggestions. We modified “by the laboratory” to “in the laboratory” and “in each plant” to “for each plant”.

Original manuscript:

For rhizosphere soil, the qualified roots were collected in a 50-ml centrifuge tube containing 30 ml of sterile Phosphate Buffer Saline (PBS) buffer (pH 7.0, per liter: 6.33 g NaH₂PO₄·H₂O, 16.5 g Na₂HPO₄·7H₂O, and 200 µl Silwet L-77) and stored on ice for further processing by the laboratory. In each plant, bulk soil samples were collected from around the root system and frozen in liquid nitrogen immediately.

Current manuscript:

For rhizosphere soil, the qualified roots were collected in a 50-ml centrifuge tube containing 30 ml of sterile Phosphate Buffer Saline (PBS) buffer (pH 7.0, per liter:

6.33 g $\text{NaH}_2\text{PO}_4 \cdot \text{H}_2\text{O}$, 16.5 g $\text{Na}_2\text{HPO}_4 \cdot 7\text{H}_2\text{O}$, and 200 μl Silwet L-77) and stored on ice for further processing **in** the laboratory. **For** each plant, bulk soil samples were collected from around the root system and frozen in liquid nitrogen immediately.

#####Reviewer #2:#####

Reviewer #2 (Remarks to the Author):

The study by Wu et al. provides some interesting insights into how rhizosphere microbiomes influence Poplar growth via various mechanisms. The authors have conducted an extensive series of experiments and analyses at both the microbial and molecular levels, utilizing datasets from transcriptomes, metabolomes, and microbiomes across various Poplar genotypes to establish a gene-metabolite-microbe network. This method is particularly compelling and underscores the significance of plant-microbe interactions in determining plant performance. However, there are several critical issues that could substantially compromise the quality and integrity of the manuscript.

Our response: We sincerely appreciate Reviewer 2 for the time and effort in reviewing the manuscript. Your insightful feedback has been instrumental in enhancing the quality and rigor of our work. We have made point-by-point responses to these comments and made corresponding revisions to the manuscript as below. We hope that these concerns have been adequately addressed.

Major concerns:

1. Reviewer #2: Bioinformatic analysis of amplicon sequencing: I am not familiar with Mothur. Based on my experience with QIIME 2 for processing raw sequencing data, the authors should use paired-end data for profiling bacterial communities due to the 500 bp base pair coverage in the quality control. Typically, forward-end reads yield better quality than reverse-end reads. I am concerned about the decision to not trim reads separately and would appreciate clarification on the quality score thresholds used to filter out low-quality reads. Specifically, how were sequences under 300 bp or over 500 bp treated? Additionally, I noticed that the authors used the QIIME 2 pipeline for taxonomic assignment. Why wasn't QIIME 2 used to process the raw sequencing data as well? Furthermore, I would like to know the identity threshold the authors set for taxonomic classification.

Our response: We thank Reviewer 2 for the constructive suggestions. According to

the suggestions of Reviewer 2, we have changed the bioinformatics analysis method of amplicon sequencing and modified the Methods section.

For instance:

Supporting Information

The 16S rRNA gene sequences were processed using QIIME (v.1.9.1), USEARCH (v.10.0) and in-house scripts. The quality of the paired-end Illumina reads was checked by FastQC (v.0.11.5) and processed in the following steps by USEARCH: joining of paired-end reads and relabeling of sequencing names (-fastq_mergepairs); removal of barcodes and primers (-fastx_truncate); filtering of low-quality reads (-fastq_filter); and finding non-redundancy reads (-fastx_uniques). Reads were clustered at 100% sequence similarity using Unoise3 with default parameters. Taxonomic assignment utilized the SILVA reference database (v132) for bacteria, one table was created for each taxonomic level (Domain, Phylum, Class, Order, Family, and Genus). All absolute sequence variants (ASVs) identified as chloroplast and mitochondria were discarded from the dataset, and ASVs represented by fewer than two sequences were filtered to avoid biases. The QIIME (v.1.9.1) was selected to evaluate the alpha diversity index (Shannon index, Chao1 index) and beta diversity index (weighted/unweighted Unifrac distance and Bray-Curtis dissimilarity) of microbial communities among all soil samples.

2. Reviewer #2: Enrichment analysis: I believe there may be a critical oversight concerning the genes related to flavonoid metabolism reported as overrepresented in this manuscript. According to the data presented in Supplementary Data 5, the adjusted p-value (even p value) for flavonoid metabolism in the enrichment analysis is over 0.05, with the enrichment factor of 0.62, which these genes related to flavonoid metabolism, I believe, are not overrepresented. This could represent a fatal flaw, as much of the subsequent experiments and analyses rely on this assumption.

Our response: We thank Reviewer 2 for this kind comment. We mistakenly pasted the KEGG enrichment analysis results of Cluster I into the location of the KEGG

enrichment analysis results of differentially expressed genes (all). Due to our negligence, the KEGG enrichment analysis results of differentially expressed genes (all) in the original manuscript and Cluster I were exactly the same. We have added the new results of the correct differential expression gene KEGG enrichment analysis to Supplementary Data 6 (revised manuscript) and checked the revised manuscript carefully. Moreover, according to the suggestions of Reviewer 2, we corrected the p-values of the differentially expressed genes and re-performed the KEGG enrichment analysis. The p-value of the flavonoid metabolism pathway was 0.00173 and the enrichment factor was 1.37 before differential expression gene correction, and the p-value was 0.00289 and the enrichment factor was 1.36 after correction.

3. Reviewer #2: Speculative conclusion: *In some sections, the conclusions cannot be fully supported by the results. For examples, the conclusion (Overall, these results showed that flavonoid pathway genes in Leuce roots regulate the secretion of flavones to enrich specific beneficial bacteria, thereby promoting poplar nitrogen metabolism and growth.), is primarily based on correlation analyses. This appears overly definitive and speculative without stronger causal evidence.*

Our response: We thank Reviewer 2 for these constructive comments. We have modified these speculative conclusions and hope we have addressed the concerns sufficiently.

For instance:

We modified “This suggests that flavonoid metabolism may vary in poplar roots and mediate the composition of the rhizosphere microbial community.” to “Thus, we hypothesize that changes in the composition and diversity of the poplar rhizosphere microbiome may be related to flavonoid metabolites.”.

We modified “Together, these results indicate that genotype properties establish root-inhabiting bacterial communities by selecting specific microbial taxa that are associated with plant nutrition and growth performance.” to “Together, these results indicate that genotype properties establish root-inhabiting bacterial communities by selecting specific microbial taxa.”.

We modified “Overall, these results showed that flavonoid pathway genes in Leuce roots regulate the secretion of flavones to enrich specific beneficial bacteria, thereby promoting poplar nitrogen metabolism and growth.” to “Overall, these results showed that specific bacteria taxa become enriched as a consequence of Leuce-specific properties and are associated with gene expression, flavonoid accumulation, and plant growth.”.

We modified “These results suggest that Pto1 inoculation induces the LR pathway mediated by *PLT3PLT5PLT7* through the secretion of IAA.” to “These results suggest that Pto1-secreted IAA plays a critical role in inducing *PLT3PLT5PLT7*-mediated LR pathways in Arabidopsis.”.

4. Reviewer #2: Language and structure: The manuscript is challenging to read in places, likely due to the structure of placing the Results and Discussion sections before the Methods section. This leads to a lot of confusions in the Results and Methods sections. I recommend reorganizing and refining the text to improve readability and coherence. Besides, the discussions should be improved to closely link to results. I noticed that there are a lot of discussion in the Results section, which should be addressed to maintain a clear distinction between these sections. Otherwise, the authors can try to combine Results and Discussion sections.

Our response: We thank Reviewer 2 for the kind suggestions. Following your suggestions, we have reorganized and refined the text to improve readability and coherence. In addition, we have modified the results section and the discussion section. We divided the discussion section into three parts, and each established the title. We hope that all concerns raised by the reviewer are now well addressed.

(1) We divided the discussion section into three parts, and each established the title.

“Comprehensive gene expression, flavonoid metabolism, and rhizosphere microbial community co-expression network construction;

Strategies of poplar ‘cry for help’ rhizosphere microorganisms under low nitrogen conditions;

Matthew effect' of interaction between host and microorganisms.”

(2) We have removed the discussion in the results section and added a link to the results in the discussion section.

For instance:

Discussion

Within this network, we identified 147 enzyme genes that encode enzymes catalyzing the twelve enzymatic reaction steps of the flavonoid biosynthesis pathway. The MBW ternary complexes containing R2R3-MYB and bHLH transcription factors along with WD-repeat proteins have been reported to regulate the biosynthesis of flavonoids. A total of 97 MYB and 72 bHLH transcription factors were identified, including MYC2 (bHLH), which shifted root microbiota composition to enhance Arabidopsis growth and immunity under shade.

We found that LM50 (*Leuce*) produced 7.37 g more stem biomass than Peu-H (*Turanga*) in sterile and nitrogen-poor soil. In the unsterilized, nitrogen-deficient soil, this difference widened to 8.29 g (Peu-H grown soil) and 10.26 g (LM50 grown soil), respectively. Notably, there were no significant differences in nutrients between soils (bulk soils or rhizosphere soils) previously grown with LM50 and Peu-H; root exudates had no significant effect on poplar growth; and bulk soil microbial diversity showed no significant variation. These results indicate that genotype-specific microbiota exerts varying degrees of positive feedback on poplar growth. Each section recruits specific taxa to shape its own rhizosphere microbial community. For instance, the *Leuce* enriches *Pseudomonas* to aid in nitrogen uptake and SR growth. The abundance of *Pseudomonas* in the *Leuce*, *Aigeiros*, *Tacamahaca*, and *Turanga* is 13.77%, 3.47%, 2.78%, and 2.76%, respectively.

Minor concerns:

5. Reviewer #2: L. 31 What are these four sections?

Our response: We thank Reviewer 2 for the constructive suggestion. We provided details to Supplementary Table 4. We also added relevant description.

For instance:

Here, we collected roots and rhizosphere soils from nine *Populus* species belonging to four sections (*Leuce*, *Aigeiros*, *Tacamahaca*, and *Turanga*), generated metabolite and transcription data for roots and microbiota data for rhizospheres, and conducted comprehensive multi-omics analyses.

Supplementary Table 4. Details of nine poplar species in four sections.

Sections	Species	Abbreviations	Sources
Leuce	P. alba × P. glandulosa Y	Pal-Y	Shandong, China
	P. tomentosa × P. bolleana M	Pto-M	Beijing, China
	P. alba × P. glandulosa 84K	84K	Korea
	P. tomentosa Lumao50	LM50	Shandong, China
Aigeiros	P. euramericana 74/76	107	Shandong, China
	P. euramericana H3-1	H3-1	Shandong, China
Tacamahaca	P. szechuanica Z	Psz-Z	Sichuan, China
	P. trichocarpa M	Pot-M	America
Turanga	P. euphratica H	Peu-H	Xinjiang, China

6. Reviewer #2: L. 34 What does “robust” mean here? Do you mean more relative abundance or number of taxa?

Our response: We thank Reviewer 2 for the constructive suggestions. Here "robust" means poplar with good growth performance. In order to ensure easy understanding, we modified “robust” to “vigorous” to maintain the consistency of the full manuscript and double-checked the relevant descriptions in the revised manuscript.

For instance:

Original manuscript:

We demonstrated that the roots of robust *Leuce* poplar enriched more plant growth-promoting rhizobacteria, which compared with the poorly performing poplar, agreeing with the 'Matthew effect' on poplar-microbe interaction.

Current manuscript:

We demonstrated that the roots of **vigorous** *Leuce* poplar enriched more *Pseudomonas*, compared with the poorly performing poplar.

7. Reviewer #2: L. 35-36 Suggest deleting "agreeing with the 'Matthew effect' on poplar-microbe interaction ". The concept of the 'Matthew effect' may not be familiar to many readers. It would be more appropriate to discuss this concept in the Discussion section, where you can provide a detailed explanation to help readers understand its relevance and implications in the context of your study.

Our response: We thank Reviewer 2 for the constructive suggestions. We have deleted the sentence "agreeing with the 'Matthew effect' on poplar-microbe interaction." and added a detailed description of the "Matthew effect" in the Discussion section.

For instance:

Discussion

In social psychology, the "Matthew effect" describes the phenomenon that the strong become stronger and the weak become weaker. We introduce the concept of the "Matthew effect" in plant-microbial interactions for the first time. That is, vigorous or resistant plant genotypes can recruit specific microbes to give them more growth advantages or better resistance.

8. Reviewer #2: L. 38 Please spell out the full name of the gene the first time it is mentioned in the text.

Our response: We thank Reviewer 2 for this constructive comment. We have spelled out the full names of all the genes first mentioned in the article. The revised manuscript has been checked carefully.

For instance:

Moreover, we confirmed that *Pseudomonas* was strongly associated with tricetin and apigenin biosynthesis and identified that gene ***GLABRA3 (GL3)*** was critical for tricetin secretion. The elevated tricetin secretion via constitutive transcription of *PopGL3* and ***Chalcone synthase (PopCHS4)*** could drive *Pseudomonas* colonization in the rhizosphere and further enhance poplar growth, nitrogen acquisition, and lateral root development in nitrogen-poor soil.

For example, flavonoids (luteolin, apigenin, etc.) could interact with rhizobial **nodulation (NodD)** proteins activating the transcription of nodulation genes responsible for the deformation of plant root hairs and assisting rhizobial entry via infection threads;

Four out of thirteen *CHS* genes and all three ***flavonoid 3'-hydroxylase (F3'H)*** genes were present in Cluster IV,

Notably, our findings revealed the presence of 28 **basic helix-loop-helix (bHLH)** and 39 MYB transcription factors in Cluster IV.

Moreover, two ***flavonol synthase (FLS)*** genes and seven (7/16) flavonols were highly correlated in Cluster V.

Further characterization showed that seven isolates possessed the capacity for nitrogen fixation and carried the ***nitrogen fixation (nifH)*** gene,

we conducted a structural analysis of the root systems of the **PLETHORA (*plt3plt5plt7*)** Arabidopsis triple mutant, which exhibits compromised LR formation, and the **WUSCHEL-related homeobox (*wox11wox12*)** Arabidopsis double mutant, displaying deficiencies in adLR and AR formation.

DAP-seq experiment revealed that *PopGL3* could regulate the transcription of ***peroxidase 2 (PopPA2)***, *PopF3'H*, ***Phospho-2-oxo-3-deoxyheptonate aldolase (PopDAHP)***, ***cinnamoyl coa reductase 1 (PopCCR1)***, and ***tetrahydroberberine oxidase (PopTHB)***, which are involved in flavonoid synthesis (Fig. 5A, B).

The constitutive expression of *PopGL3* or *PopCHS4* could activate the transcription of *PopF3'H*, and ***flavone synthase (PopFNS)*** and release more tricetin in the rhizosphere of the *PopCHS4-OE* (chalcone synthase catalyzes the first committed step

of the multi-branched flavonoid pathway) and *PopGL3-OE* lines (P -values < 0.01 ; Fig. 5 E, F).

we tagged *Pto1* with **red fluorescent protein (RFP)** gene and used confocal microscopy to image the colonization of root tissue across various genotypes.

9. Reviewer #2: L. 41-44 Please ensure consistent tense.

Our response: We thank Reviewer 2 for the kind suggestion. We modified “reveals” to “revealed” to ensure consistent tense.

Original manuscript:

This study reveals plant-metabolite-microbe regulation patterns contribute to the poplar fitness and thoroughly decoded the key regulatory mechanisms of triclin, and provided new insights into the interactions of the plant’s key metabolites with its transcriptome, rhizosphere microbes.

Current manuscript:

This study **revealed** plant-metabolite-microbe regulation patterns contribute to the poplar fitness and thoroughly decoded the key regulatory mechanisms of triclin, and provided new insights into the interactions of the plant’s key metabolites with its transcriptome, rhizosphere microbes.

10. Reviewer #2: L. 85-87. Please clarify the reason why the correlation analysis between genes (are these microbial or plant genes?) and microbial factors (are these at the community level or functional level?) is insufficient to elucidate regulatory pathways using meta-omics perspectives.

Our response: We thank Reviewer 2 for the constructive suggestions. Here "genes" means the genes of the plant host, and "microbial factors" refers to at the community level. We have modified this sentence for ease of understanding.

Original manuscript:

Previous studies have effectively integrated plant transcriptomics with microbiome community data using methods such as Weighted Gene Co-Expression Network Analysis (WGCNA) and Microbiome-Wide Association Studies (MWAS) analyses, demonstrating that host gene expression indeed influences the composition of the microbial community. However, analyzing the correlation between the gene-microbe factors cannot truly elucidate the regulating pathways from a multi-omics perspective.

Current manuscript:

Previous studies have effectively integrated plant transcriptomics or genomics with microbiome community data using methods such as Weighted Gene Co-Expression Network Analysis (WGCNA) and Microbiome-Wide Association Studies (MWAS) analyses, demonstrating the significance of host gene expression (CYP72A154 and Nucleotide-Binding-Leucine-Rich-Repeat) in shaping the composition of microbial communities. However, analyzing the correlation between host genes and the microbial community cannot elucidate the role of metabolites in plant-regulated microbial structure.

11. Reviewer #2: L.74 Overall assembly and functional changes of microbial communities

Our response: We appreciated your valuable suggestion. We have modified this sentence.

Original manuscript:

However, the mechanism by which poplar genotypes regulate microbial communities overall assembly and functional changes remains largely unclear, as are the effects of host genotype-selected rhizosphere microbiomes on poplar growth and fitness.

Current manuscript:

However, the genetic mechanism by which poplar genotypes regulate the **overall assembly and functional changes of microbial communities** in nitrogen-poor conditions remains largely unclear, as do the effects of host genotype-selected rhizosphere microbiomes on poplar growth and fitness.

12. Reviewer #2: L.75 as do ...

Our response: We appreciated Reviewer 2 for the valuable suggestion. We have modified this sentence.

Original manuscript:

However, the mechanism by which poplar genotypes regulate microbial communities overall assembly and functional changes remains largely unclear, as are the effects of host genotype-selected rhizosphere microbiomes on poplar growth and fitness.

Current manuscript:

However, the genetic mechanism by which poplar genotypes regulate the overall assembly and functional changes of microbial communities in nitrogen-poor conditions remains largely unclear, **as do** the effects of host genotype-selected rhizosphere microbiomes on poplar growth and fitness.

13. Reviewer #2: L.92-94 This sentence is unclear to me.

Our response: We thank Reviewer 2 for this kind comment. We have modified this sentence.

Original manuscript:

By using this multidimensional dataset, the regulating chains that how poplar recruiting target beneficial microbes and how the microbes affect the host's fitness were constructed.

Current manuscript:

By using this multidimensional dataset, the mechanisms of how poplar recruits target beneficial microbes and how these microbes affect host fitness were revealed.

14. Reviewer #2: L. 95 What are the molecular analysis?

Our response: We thank Reviewer 2 for this kind comment. We modified “molecular analysis” to “molecular experiments”. The molecular experiments included DAP-seq,

84K genetic transformation, metabolite quantification, and qPCR.

Original manuscript:

The function of the key genes was further investigated by using molecular analyses. We highlight how root exudates, the rhizobiome, and their mutual interactions affect host fitness and how these explicitly processes occurred in our investigations of belowground plant-rhizobiome interactions.

Current manuscript:

The function of the key genes was further investigated by using **molecular experiments**. We highlight how host genes, root exudates, and the rhizobiome have mutual interactions and how these explicitly processes affect plant fitness.

15. Reviewer #2: L.96 explicit

Our response: We thank Reviewer 2 for this kind comment. We have modified this sentence.

Original manuscript:

We highlight how root exudates, the rhizobiome, and their mutual interactions affect host fitness and how these explicitly processes occurred in our investigations of belowground plant-rhizobiome interactions.

Current manuscript:

We highlight how host genes, root exudates, and the rhizobiome have mutual interactions and how these explicitly processes affect plant fitness.

16. Reviewer #2: L.97 *It would benefit readers to understand the scope of this study if the authors could provide their hypotheses*

Our response: We thank Reviewer 2 for this valuable suggestion. We have added the hypotheses to the revised manuscript.

For instance:

In this study, we hypothesize that (1) the comprehensive gene-metabolite-microbe

co-expression network provides an effective research tool to elucidate the genetic mechanisms by which poplar regulates metabolite-mediated specific microbial recruitment; (2) poplar regulates the secretion of flavonoids by functional genes to reshape the rhizosphere microbial community, and attract specific microbes that, through the ‘cry for help’, improve plant adaptability to low-nitrogen stress.

17. Reviewer #2: L.99 This subtitle is unclear to me. It could benefit from some modifications to improve clarity.

Our response: We thank Reviewer 2 for this valuable suggestion. This subtitle has been modified to "Growth promotion of poplar by the soil microbial community".

Original manuscript:

Robust species become more robust by shaping the root microbial community

Current manuscript:

Growth promotion of poplar by the soil microbial community

18. Reviewer #2: L. 100-104 I understand the intent behind the authors' statements, but these sentences could benefit from further refinement for enhanced clarity and coherence. Besides, could you specify what these eleven phenotypes are?

Our response: We thank Reviewer 2 for this kind comment. We have modified these sentences in the revised manuscript and added the phenotype data to Supplementary Data 1.

For instance:

To assess the importance of the root microbiome in plant fitness, we performed a pot experiment on nine representative poplar species derived from four sections (*Leuce*, *Aigeiros*, *Tacamahaca*, and *Turanga*; grown in low nitrogen, unsterilized natural mixed soil; total nitrogen: 0.089%). After three months of growth, eleven phenotypes of poplar were detected (Supplementary Fig. 1; Supplementary Data 1).

--Sorry for this being unclear. Due to the word limit of the main text, we have

included eleven phenotypes in the Methods section, where they are described in detail.

For instance:

we examined eleven representative characteristics encompassing morphological and structural (growth traits: plant height, ground diameter, shoot biomass, root biomass, and root length; leaf traits: leaf length, leaf width, leaf area, and leaf number; physiological functional: chlorophyll content; and component content: leaf nitrogen content) aspects of nine poplar species (Supplementary Data 1).

19. Reviewer #2: L.108-109 *Could you please indicate where these results are displayed?*

Our response: We thank Reviewer 2 for this kind comment. We added the phenotypic data to Supplementary Data 1.

For instance:

The growth parameters varied among species. For example, mean root biomass ranged from 0.56 to 15.76 g, plant height ranged from 27.02 to 129.91 cm, and leaf area ranged from 1.06–76.30 cm² (Supplementary Data 1).

20. Reviewer #2: L. 115-119 *In the Methods section (L.448), it is stated that soil mixed from four parts was used for the pot study. However, this description is somewhat confusing as it suggests that different soil sources might have been used to investigate plant-microbial interactions.*

Our response: We thank Reviewer 2 for this kind comment. Plants and environmental microorganisms have established a co-evolution relationship¹. Although genetic variation within plant hosts is influential for root microbial community assembly, variation in soil conditions is associated with differential microbial colonization, such as soil microbial community composition^{2,3}. Because mixing soils from different sources can provide more abundant microorganisms, thus forming a microbial bank for poplars. We have modified the Methods section.

Original manuscript:

Four parts of soil were mixed in equal volumes and thoroughly stirred.

Current manuscript:

To provide more abundant microorganisms for different genotypes of poplar, four parts of soil were mixed in equal volumes and thoroughly stirred.

References

1. Mesny, F., Hacquard, S. & Thomma, B. P. Co-evolution within the plant holobiont drives host performance. *Embo Rep.* **24**, e57455 (2023).
2. Veach, A. M. et al. Rhizosphere microbiomes diverge among *Populus trichocarpa* plant-host genotypes and chemotypes, but it depends on soil origin. *Microbiome.* **7**, (2019).
3. Guo, Q., Liu, L., Liu, J., Korpelainen, H. & Li, C. Plant sex affects plant-microbiome assemblies of dioecious *Populus cathayana* trees under different soil nitrogen conditions. *Microbiome.* **10**, (2022).

21. Reviewer #2: L. 119 "growth inhibition" to what (plant roots? shoots? total biomass?). Please clarify it.

Our response: We appreciated your valuable suggestions. This refers to shoot biomass, and we have modified this sentence.

Original manuscript:

In contrast, LM50 showed significant growth inhibition when transplanted into Peu-H-grown soil compared with LM50-grown soil (a reduction of 19.58%; ANOVA, P -values < 0.01 ; Fig. 1A, B).

Current manuscript:

In contrast, LM50 showed significant growth inhibition when transplanted into Peu-H-grown soil compared with LM50-grown soil (**19.58% decrease in shoot biomass**; ANOVA, P -values < 0.01 ; Fig. 1A, B).

22. Reviewer #2: L.125-126 *Did you apply root exudates to different genotypes? It appears that the root exudates were only applied to the 84K genotype. Could you please clarify this?*

Our response: We thank Reviewer 2 for this kind comment. In order to examine the effects of secretions on the growth of different poplar genotypes, we inoculated root secretions with Peu-H. The results showed that secretions from different poplar genotypes had no significant effects on the growth of Peu-H. We have added these new results and methods to the revised manuscript and prepared a new figure (Supplementary Fig. 2G-I).

For instance:

Notably, there were no significant differences in nutrients in soils (bulk soils or rhizosphere soils) previously grown with LM50 and Peu-H (ANOVA; Supplementary Table 1), and root exudates from all genotypes had no significant effect on poplar (84K and Peu-H) biomass in sterilized soil (Supplementary Fig. 2D-I).

G

H

I

Supplementary Fig. 2 (G) Morphological differences of Peu-H after applying different poplar root exudates (LM50 exudate or Peu-H exudate). Plant height (**H**) and fresh shoot biomass (**I**) of Peu-H after applying different poplar root exudates (LM50 exudate or Peu-H exudate). $n = 3$ biologically independent samples. Asterisks indicate significant differences between different groups (Student's t-test, ns: not significant). FW, fresh weight. Scale bars: (G) 5 cm.

23. Reviewer #2: L.127-129 *I don't believe this is due to the root microbial community. Rather, the 'rhizosphere microbiome' would presumably be more*

accurate.

Our response: We thank Reviewer 2 for this kind suggestion. Soil microbes play critical roles for plants's growth and health, while rhizosphere microbes are critical determinants of plant growth and health. In our soil transplantation experiment, bulk soil and rhizosphere soil were transplanted. Although the effects of soil nutrients and root exudates on poplar growth are excluded, it cannot be proved that bulk soil microbes or rhizosphere microbes affect poplar differential growth. Therefore, it is not suitable to use "rhizosphere microbiome" here. We have modified "root microbial community" to "soil microbial community", and hope that this concern has been adequately addressed.

Original manuscript:

Specifically, the root microbial community shaped by the vigorous genotype was more conducive to plant growth, whereas the promoting effect of the root microbial community recruited by the less robust genotype was weaker.

Current manuscript:

Specifically, the **soil microbial community** shaped by the vigorous genotype was more conducive to plant growth, whereas the promoting effect of the **soil microbial community** recruited by the less robust genotype was weaker.

To investigate whether rhizosphere microbes or bulk soil microbes affect the differential growth of poplar, we performed amplicon analysis. The results showed that there was no significant difference in bulk soil microbial diversity among different poplar sections, but there was a significant difference in rhizosphere microbial diversity (Supplementary Fig. 3A). Four poplar sections established root-dwelling microbial communities by selecting specific microbial taxa (Fig. 1C, Supplementary Fig. 3B). These results suggest that rhizosphere-specific microbes may be involved in the growth of poplar.

24. Reviewer #2: L. 129-131 In addition to microbial communities, soil nutrients

play a crucial role in influencing plant growth. I would suggest including a comparison of soil nutrient levels in this study as well.

Our response: We appreciated your valuable suggestions. To determine whether there are differences in nutrients in soils previously grown with the two poplar genotypes (Peu-H and LM50), we have examined the total nitrogen, available phosphorus, available potassium, and organic matter contents of Peu-H and LM50 bulk soils and rhizosphere soils, respectively. Compared with bulk soil, the total nitrogen and available potassium in rhizosphere soil increased significantly, while the available phosphorus decreased significantly. However, there is no significant difference between the bulk soil and bulk soil, rhizosphere soil and rhizosphere soil between the two poplar species. This excluded the effect of soil nutrients on the results of soil transplantation. We have added the relevant descriptions to the revised manuscript.

For instance:

Notably, there were no significant differences in nutrients in soils (bulk soils or rhizosphere soils) previously grown with LM50 and Peu-H (ANOVA; Supplementary Table 1).

Supplementary Table 1. Basic chemical properties of soils.

	SOM (g/kg)	TN (%)	AP (mg/kg)	AK (mg/kg)
LM50 bulk soil	23.6 b	0.070 b	11.9 a	139.1 b
Peu-H bulk soil	23.9 ab	0.072 b	12.2 a	142.7 b
LM50 rhizosphere soil	26.2 a	0.084 a	10.6 b	153.2 a
Peu-H rhizosphere soil	25.7 a	0.087 a	10.1 b	157.4 a

Note: TN: total nitrogen, SOM: soil organic matter, AP: available phosphorus, AK: available potassium. $n = 3$ biologically independent samples. Different letters indicate significantly different groups (One-way ANOVA, P -values < 0.05).

25. Reviewer #2: L. 132 poplar genotypes?

Our response: We thank Reviewer 2 for this constructive suggestion. We have modified this subtitle.

Original manuscript:

Taxonomic features of the rhizosphere microbial composition between poplar

Current manuscript:

Taxonomic features of the rhizosphere microbial composition between **poplar genotypes**

26. Reviewer #2: L.138 Please specify the type of diversity. I think here is the alpha diversity.

Our response: We thank Reviewer 2 for this constructive comment. Diversity here refers to alpha diversity (i.e. Shannon diversity). We have modified this sentence.

Original manuscript:

Across sections, *Turanga* showed the highest diversity in rhizosphere microbiota, followed by *Aigeiros*, *Tacamahaca*, and *Leuce* (ANOVA, P -values < 0.05; Supplementary Fig. 3A).

Current manuscript:

Across sections, *Turanga* showed the highest **shannon diversity** in rhizosphere microbiota, followed by *Aigeiros*, *Tacamahaca*, and *Leuce* (ANOVA, P -values < 0.05; Supplementary Fig. 3A).

27. Reviewer #2: L. 140-141 This sentence seem discussion.

Our response: We thank Reviewer 2 for the constructive suggestions. Following your suggestion, we have removed this sentence.

28. Reviewer #2: L. 142 The term 'compositional volatility' used here is somewhat unusual. I assume the authors intended to refer to 'compositional variation' instead.

Our response: We thank Reviewer 2 for this constructive comment. We modified “compositional volatility” to “compositional variation”.

Original manuscript:

The compositional volatility of the rhizosphere microbiome among the four sections is driven by significant shifts in the relative abundance of nineteen specific bacterial phyla (LDA score > 2, Kruskal-Wallis test, FDR adjusted *P*-values < 0.05; Fig. 1C, Supplementary Fig. 3B, and Supplementary Data 3).

Current manuscript:

The **compositional variation** of the rhizosphere microbiome among the four sections is driven by significant shifts in the relative abundance of 22 specific bacterial phyla (Linear discriminate analysis effect size ‘LEfSe’, LDA score > 2, Kruskal-Wallis test, FDR adjusted *P*-values < 0.05; Fig. 1C, Supplementary Fig. 3B, and Supplementary Data 2).

29. Reviewer #2: L. 145-148: What percentage of taxa were identified at the genus level? Is the 16S primer set sufficient for genus-level taxonomy identification (if so, provide the evidence by citing previous references in the method section and briefly mention here)?

Our response: We thank Reviewer 2 for this kind comment. Following your suggestion, we have added the relative abundance of taxa identified at the genus level to the revised manuscript (Supplementary Data3). Moreover, we have submitted the research data to the figshare database ([10.6084/m9.figshare.26426578](https://doi.org/10.6084/m9.figshare.26426578)). which contains the percentage of all taxa at the genus level.

For instance:

At the genera level, we identified 109 specific markers in *Turanga*, 90 in *Aigeiros*, 45 in *Tacamahaca*, and 37 in *Leuce* (LDA score > 2, Kruskal-Wallis test, FDR adjusted *P*-values < 0.05; Fig. 1C), respectively. We noticed that *Nitrosospira* (1.07%), *Actinomadura* (0.03%), and *Tumebacillus* (0.23%) were highly abundant in the *Turanga* (Supplementary Data3). *Bacillus* (1.61%) and *Enterobacter* (2.80%) were

found to be enriched in *Aigeiros*. Notably, the 37 marker genera detected in *Leuce* accounted for 41.15% of the relative abundance, with *Pseudomonas* having the highest abundance at 13.77%.

--Previous studies have shown that 16S primers can be used for genus-level classification identification^{1,2}, and we have cited previous references in the methods section and briefly described them here.

For instance:

The annotated taxonomic levels were (Domain, Phylum, Class, Order, Family, and Genus).

Supporting Information

Taxonomic assignment utilized the SILVA reference database (v132) for bacteria, one table was created for each taxonomic level (Domain, Phylum, Class, Order, Family and Genus).

References

1. Lu, J. & Salzberg, S. L. Ultrafast and accurate 16S rRNA microbial community analysis using Kraken 2. *Microbiome*. **8**, (2020).
2. Lima, J. et al. Taxonomic annotation of 16S rRNA sequences of pig intestinal samples using MG-RAST and QIIME2 generated different microbiota compositions. *J. Microbiol. Methods*. **186**, 106235 (2021).

30. Reviewer #2: L. 146: "exclusively found". In contrast to the taxa exclusively linked with each plant section, what percentage of microbial taxa are shared across these sections? If a significant portion of taxa are shared and their relative abundance remains consistent across plant sections, how can we ascertain that alterations in plant performance are attributed to the unique taxa rather than the shared ones, which might change their functions in response to varying plant genotypes (or soil sources, if not used mixed soil)?

Our response: We thank Reviewer 2 for this kind comment. We made a mistake here.

What we want to say is that "we identified 109, 90, 45, and 37 specific markers in *Turanga*, *Aigeiros*, *Tacamahaca*, and *Leuce*, respectively". We have modified this sentence in the revised manuscript, and we are sorry for causing the you wrong understanding.

Original manuscript:

At the genera level, we identified 82 specific markers in *Turanga*, compared to those exclusively found in *Aigeiros* (28 genera), *Tacamahaca* (23 genera), and *Leuce* (11 genera; LDA score > 2, Kruskal-Wallis test, FDR adjusted *P*-values < 0.05; Fig. 1C).

Current manuscript:

At the genera level, we identified 109 specific markers in *Turanga*, 90 in *Aigeiros*, 45 in *Tacamahaca*, and 37 in *Leuce* (LDA score > 2, Kruskal-Wallis test, FDR adjusted *P*-values < 0.05; Fig. 1C), respectively.

--Here, we cannot be ascertain that alterations in plant performance are attributed to the unique taxa rather than the shared ones. Therefore, we have revised the conclusions of this paragraph to remove "that are associated with plant nutrition and growth performance".

Original manuscript:

Together, these results indicate that genotype properties establish root-inhabiting bacterial communities by selecting specific microbial taxa that are associated with plant nutrition and growth performance.

Current manuscript:

Together, these results indicate that genotype properties establish root-inhabiting bacterial communities by selecting specific microbial taxa.

31. Reviewer #2: L: 150: "highly abundant" clarify whether it is relative abundance or absolute abundance based here.

Our response: We thank Reviewer 2 for this kind comment. "highly abundance"

means relative abundance. Here we added the relative abundance of specific genera and modified the legend in Figure 1C.

For instance:

(1) We added the relative abundance of specific genera

At the genera level, we identified 109 specific markers in *Turanga*, 90 in *Aigeiros*, 45 in *Tacamahaca*, and 37 in *Leuce* (LDA score > 2, Kruskal-Wallis test, FDR adjusted *P*-values < 0.05; Fig. 1C), respectively. We noticed that *Nitrosospira* (1.07%), *Actinomadura* (0.03%), and *Tumebacillus* (0.23%) were highly abundant in the *Turanga* (Supplementary Data3). *Bacillus* (1.61%) and *Enterobacter* (2.80%) were found to be enriched in *Aigeiros*. Notably, the 37 marker genera detected in *Leuce* accounted for 41.15% of the relative abundance, with *Pseudomonas* having the highest abundance at 13.77%.

(2) We modified the legend in Figure 1C

Original manuscript:

Each node represents a species, and the larger the node, the higher the abundance.

Current manuscript:

Each node represents a species, and the larger the node, the higher the **relative abundance**.

32. Reviewer #2: L.155-163 *This part looks more like discussion rather than results.*

Our response: We thank Reviewer 2 for this kind comment. We have removed this part from the revised manuscript.

33. Reviewer #2: L.166- 167 and 171 – 173: *Similar to the comments above, it is not known if the changes in plant gene expressions respond to microbial mediated activities or soil chemistry or both. Clarify if the mixed soil was used for this section.*

Our response: We thank Reviewer 2 for this kind comment. Plant functional genes have been found to play an important role in shaping the rhizomicrobiome¹⁻³. Plant

functional genes could regulate root phenotypic traits and the secretion of root exudates, such as organic acids and hormones^{4,6}, which have been found to drive microbial community assembly in the rhizosphere⁶. In contrast, rhizosphere microorganisms can also confer health advantages to plants by inducing the expression of plant functional genes^{7,8}. It has been proven that the expression of genes related to nutrient uptake and stress resistance is affected by structural and functional changes in the rhizosphere microbiota. In addition, soil chemistry also influenced changes in plant gene expression⁹. Following your suggestion, we have modified “To guide our efforts to investigate molecular mechanisms during rhizosphere microbial recruitment in poplar,” to “To explore the correlation between poplar gene expression and rhizosphere microbial recruitment,”.

We performed PERMANOVA analyses and showed that poplar genotypes significantly influence rhizosphere bacterial community composition, phenotype, and root transcriptomics. We describe these results in detail below, and modified “This indicates a strong impact of poplar genetic regulation on the composition of the rhizosphere microbiome.” to “This indicates that microbial composition, growth characteristics, and gene expression were significantly affected by genotype.”

--In order to eliminate the influence of soil chemical factors on the experimental results, we planted nine poplar species from four sections in the same mixed soil. Here, all poplar genotypes are grown in the same unsterilized mixed soil. We have clarified this condition in the revised manuscript.

Original manuscript:

To guide our efforts to investigate molecular mechanisms during rhizosphere microbial recruitment in poplar, we generated 73.7 Gb of root transcriptomic data across nine poplar species.

Current manuscript:

To explore the correlation between poplar gene expression and rhizosphere microbial recruitment, we generated 73.7 Gb of root transcriptomic data across nine poplar

species (**grown in low nitrogen, unsterilized natural mixed soil**).

References

1. Song, Y. et al. FERONIA restricts *Pseudomonas* in the rhizosphere microbiome via regulation of reactive oxygen species. *Nat. Plants*. **7**, 644-654 (2021).
2. Zhang, J. et al. NRT1.1B is associated with root microbiota composition and nitrogen use in field-grown rice. *Nat. Biotechnol.* **37**, 676-684 (2019).
3. Stringlis, I. A. et al. MYB72-dependent coumarin exudation shapes root microbiome assembly to promote plant health. *Proceedings of the National Academy of Sciences*. **115**, E5213-E5222 (2018).
4. Kaushal, R. et al. Dicer-like proteins influence Arabidopsis root microbiota independent of RNA- directed DNA methylation. *Microbiome*. **9**, 57 (2021).
5. Lei Wang, B. et al. Strigolactone and Karrikin Signaling Pathways Elicit Ubiquitination and Proteolysis of SMXL2 to Regulate Hypocotyl Elongation in *Arabidopsis thaliana*. *The Plant Cell*. **32**, 2251-2270 (2020).
6. Yu, P. et al. Plant flavones enrich rhizosphere Oxalobacteraceae to improve maize performance under nitrogen deprivation. *Nat. Plants*. **7**, 481-499 (2021).
7. Liu, S. et al. Transcriptome profiling of genes involved in induced systemic salt tolerance conferred by *Bacillus amyloliquefaciens* FZB42 in *Arabidopsis thaliana*. *Sci. Rep.* **7**, 10713-10795 (2017).
8. Berendsen, R. L. et al. Disease-induced assemblage of a plant-beneficial bacterial consortium. *Isme J.* **12**, 1496-1507 (2018).
9. Zhu, F. et al. Species-specific plant-soil feedbacks alter herbivore-induced gene expression and defense chemistry in *Plantago lanceolata*. *Oecologia*. **188**, 801-811 (2018).

34. Reviewer #2: L.168-171 I strongly recommend conducting a statistical analysis, such as PERMANOVA, to determine whether genotypes significantly affect the rhizosphere bacterial community composition, phenotype, and root transcriptomics.

Our response: We thank Reviewer 2 for this constructive comment. Following your

suggestion, we performed PERMANOVA analyses to the rhizosphere bacterial community composition, phenotype, and root transcriptomics. Differences between poplar genotypes explained the variation for rhizosphere bacterial community composition (PERMANOVA, $R^2=0.52$, P -values < 0.01 ; Fig. 1D, G and Supplementary Data 4), phenotype (PERMANOVA, $R^2=0.92$, P -values < 0.01 ; Fig. 1E, H and Supplementary Data 4), and root transcriptomics (PERMANOVA, $R^2=0.69$, P -values < 0.01 ; Fig. 1F, I and Supplementary Data 4). Moreover, the rhizosphere bacterial community composition, phenotype, and root transcriptomics showed significant differences (P adjusted < 0.05 ; Supplementary Data 4) among any two poplar sections. We have added detailed descriptions in the Result and Method sections and added the PERMANOVA analysis results to Fig. 1D-F.

For instance:

The Principal Component Analysis (PCA) and Hierarchical Clustering Analysis (HCA) based on all microbial (amplicon sequence variant, ASV > 2 ; PERMANOVA, $R^2=0.52$, P -values < 0.01 ; Fig. 1D, G and Supplementary Data 4), phenotypic (PERMANOVA, $R^2=0.92$, P -values < 0.01 ; Fig. 1E, H and Supplementary Data 4), and transcriptomic (PERMANOVA, $R^2=0.69$, P -values < 0.01 ; Fig. 1F, I and Supplementary Data 4) data clearly classified the nine poplar species into four distinct subgroups, each associated with a specific section.

Materials and methods

The statistical significance of rhizosphere bacterial community composition, phenotype, and root transcriptomics differences between poplar genotypes was assessed by PERMANOVA using the *vegan* (v.2.6-2) package.

Fig. 1 Principal Component Analysis (PCA) of the microbiome (**D**; ASV > 2), phenotype (**E**), and transcriptome (**F**; TPM > 0) datasets from the nine poplar species.

35. Reviewer #2: L. 174 *Why haven't adjusted p-values been used here? Relying solely on p-values to filter over-represented genes could lead to false positives.*

Our response: We appreciated your valuable suggestions. We agree with the reviewer's perspective. Following your suggestion, we have adjusted the p-values for the differentially expressed genes (DEGs). When screening the DEGs ($|\log_2FC| > 1$, FDR adjusted P -values < 0.05), we found that the number of significantly differentially expressed genes changed only slightly. For instance, the DEGs between the *Leuce* and *Aigeiros* sections decreased by 3.90% (474/12,144), between the *Leuce* and *Tacamahaca* sections by 4.24% (552/13,009), and the total number of DEGs decreased by only 2.50% (597/23,927). We have used adjusted p-values in the revised manuscript, re-performed KEGG functional enrichment analysis for DEGs, and reconstructed the co-expression network. The relevant results and figures have been added to the revised manuscript.

We redraw Fig. 1F, I, Fig. 2, Fig. 3C, D, Supplementary Fig. 4, Supplementary Fig. 5A, Supplementary Fig. 6B, D, and Supplementary Fig. 7 (revised manuscript) according to the new analysis results.

36. Reviewer #2: L. 177-180 and 203-204 *Again, these are discussion.*

Our response: We thank Reviewer 2 for this kind comment. The necessary explanation of flavonoid and microbial interactions is given here (L. 177-180) to make it more readable. We have modified this section and hope that this concern has been adequately addressed.

Original manuscript:

Flavonoids play vital roles in the assembly of plant root microbiome communities, such as the roots of *Arabidopsis* (*Arabidopsis thaliana* L.) and maize (*Zea mays* L.).

This suggests that flavonoid metabolism may vary in poplar roots and mediate the composition of the rhizosphere microbial community.

Current manuscript:

Flavonoids play vital roles in the assembly of plant root microbiome communities, such as the roots of *Arabidopsis* (*Arabidopsis thaliana* L.) and maize (*Zea mays* L.). Thus, we hypothesized that poplar functional genes regulate flavonoid synthesis to mediate changes in rhizosphere microbiome composition and diversity.

--We have removed this sentence “L. 203-204: Among them, apigenin (methylApigenin C-pentoside) could recruit beneficial bacteria such as *Rhizobium*, Oxalobacteraceae, and *Pseudomonas*, enhancing the plant’s nitrogen uptake capacity.”

37. Reviewer #2: L.183 Please explain the rationale behind setting this particular cutoff value of $|\log_2FC|$?

Our response: We thank Reviewer 2 for this kind comment. $|\log_2FC| > 1.585$, which corresponds to a fold change ≥ 3 or ≤ 0.333 . In the screening of differential metabolites, the usual standard is $|\log_2FC| > 1$ (i.e., fold change ≥ 2 or ≤ 0.5). To enhance the stringency of our selection, we have elevated the differential fold change threshold to fold change ≥ 3 or ≤ 0.333 . We modified “ $|\log_2FC| > 1.585$ ” to “fold change ≥ 3 or ≤ 0.333 ” for easier understanding.

Original manuscript:

Next, we quantified 129 flavonoids from the root samples of nine poplar species, of which 110 (85.27%) were differentially accumulated in at least two species ($|\log_2FC| > 1.585$, P -values < 0.05 ; Supplementary Data 6).

Current manuscript:

Next, we quantified 129 flavonoids from the root samples of nine poplar species, of

which 110 (85.27%) were differentially accumulated in at least two species (**fold change ≥ 3 or ≤ 0.333 , P -values < 0.05 ; Supplementary Data 7).**

38. Reviewer #2: L.188 *What do you mean by 'a unified and distinct abundance pattern'? How can these two conditions coexist simultaneously?*

Our response: We appreciated your valuable suggestions. We modified “a unified and distinct abundance pattern” to “a distinct abundance pattern”.

Original manuscript:

These clusters demonstrated a unified and distinct abundance pattern related to specific sections, such as *Turanga* (Cluster I), *Leuce* (Cluster IV), and *Aigeiros* (Cluster V).

Current manuscript:

These clusters demonstrated a distinct abundance pattern related to specific sections, such as *Turanga* (Cluster I), *Leuce* (Cluster IV), and *Aigeiros* (Cluster V).

39. Reviewer #2: L. 208 *the elucidation of ...*

Our response: We thank Reviewer 2 for this kind comment. We have modified this sentence to make it easier to understand.

Original manuscript:

Therefore, the co-expression network facilitates elucidating the genetic mechanisms of microbial recruitment and identifying candidate genes.

Current manuscript:

In summary, our network data contains a substantial number of genes related to flavonoid synthesis, indicating that the co-expression network facilitates elucidating the genetic mechanisms of microbial recruitment and identifying candidate genes.

40. Reviewer #2: L.226: *Briefly indicate here for what tools were used to identify/predict the function of these OTUs?*

Our response: We thank Reviewer 2 for this kind comment. Considering the suggestions of Reviewer#5, this part has been deleted.

41. Reviewer #2: L. 254 Could you please display the magnitude of relative abundance in Figure 3E?

Our response: We appreciated your valuable suggestions. Following your suggestion, we have redrawn Figure 3E (Supplementary Fig. 8A in the revision) and the new figure shows the relative abundance of the top 10 dominant families of rhizosphere microorganisms in bulk soils and rhizosphere soils.

Supplementary Fig. 8 (A) Relative abundance of the top 10 rhizosphere microbial families in bulk soil and rhizosphere soil. The numbers are the relative abundances of Pseudomonadaceae.

42. Reviewer #2: L. 264: Were these rhizosphere soil collected from the pots with the mixed soil samples cultivated with Leuce, or from the site where Leuce was initially grown?

Our response: We thank Reviewer 2 for this kind comment. These rhizosphere soils were collected from the pots with the mixed soil samples cultivated with *Leuce*, including Pto-M, 84K, Pal-Y, and LM50. We have added relevant descriptions to this

sentence and Supplementary Methods 4.

Original manuscript:

We further isolated eleven *Pseudomonas* strains from rhizosphere soil samples of *Leuce*, and the 16S rRNA genes of nine isolates exhibited highly homologous (> 98%) to OTUs of Cluster IV (Supplementary Data 9).

Current manuscript:

We further isolated eleven *Pseudomonas* strains from rhizosphere soil samples of *Leuce* (**Pto-M, 84K, Pal-Y, and LM50**), and the 16S rRNA genes of ten isolates exhibited highly homologous (> 99%) to ASVs of Cluster IV (Supplementary Data 10).

Supplementary Information

Original manuscript:

The poplar rhizosphere soils *Leuce* (Pto-M, 84K, Pal-Y, and LM50) were diluted using a 10-fold dilution method to a dilution of 10^{-5} .

Current manuscript:

The poplar rhizosphere soils collected from **the pots with the mixed soil samples** cultivated with *Leuce* (Pto-M, 84K, Pal-Y, and LM50) were diluted using a 10-fold dilution method to a dilution of 10^{-5} .

43. Reviewer #2: L. 265 Do you mean sanger sequencing of each strain?

Our response: We thank Reviewer 2 for this kind suggestion. We sequenced the 16S rRNA genes of these strains by Sanger sequencing, and added relevant descriptions in Supplementary Method 4.

For instance:

Supplementary Information

The 16S rRNA genes (V3-V6) were cloned from all single strains using the specific primer (341F-1046R; Supplementary Data 12) and sequenced using the Sanger

method.

44. Reviewer #2: L.272 may be

Our response: We thank Reviewer 2 for this constructive comment. We modified “may” to “may be”.

Original manuscript:

qRT-PCR analysis suggests that flagellar-related genes (*motA*, *fliG*, and *bifA*) and biofilm formation-related gene *algU* were activated, which may critical for successful bacterial root colonization (P -values < 0.01 ; Fig. 4B).

Current manuscript:

qRT-PCR analysis suggests that flagellar-related genes (*motA*, *fliG*, and *bifA*) and biofilm formation-related gene *algU* were activated, which **may be** critical for successful bacterial root colonization (ANOVA, P -values < 0.01 ; Fig. 4B).

Reviewer #2: L. 279 What is this poplar genotype?

Our response: We thank Reviewer 2 for this kind comment. This poplar genotype is *P. alba* × *P. glandulosa* 84K ‘84K’. We have added relevant descriptions to the revised manuscript.

For instance:

Inoculated isolates significantly increased the shoot biomass (26.04%–48.03%), root biomass (57.51%–81.46%), and leaf nitrogen content (7.98%–10.15%) of poplar (84K; ANOVA, P -values < 0.01 ; Fig. 4C, D and Supplementary Fig. 10A).

45. Reviewer #2: L.291 Why did not you use poplar to directly test the effect of Pto1 on lateral root growth. Arabidopsis and poplar differ significantly in their physiology, ecology, and genetic makeup. What works in Arabidopsis might not directly translate to poplar due to these differences.

Our response: We thank Reviewer 2 for this constructive comment. In the process of verifying the effects of pseudomonads on poplar growth, we found that the Pto1 strain

could significantly promote secondary roots (SRs) development in poplar. Dicotyledons secondary roots can be further subdivided into lateral roots (LRs) and adventitious lateral roots (adLRs)¹. Secondary roots are morphologically undistinguishable from the primary root, but the genetic regulation mechanisms of these two kinds of secondary roots are different. We want to further investigate what kind of secondary root is induced by the Pto1 strain and which genetic pathways are involved. Therefore, obtaining the mutant with adventitious lateral root deletion is the key to carrying out this study.

Auxin can activate the plant-specific AP2 transcription factors PLETHORA (PLT) 3, PLT 5, and PLT 7²⁻⁴. These PLTs are essential for subsequent lateral root initiation, emergence, and spacing of lateral root primordia. In response to wounding or other environmental signals, plants activate the adventitious rooting pathway to produce another class of secondary roots, adventitious lateral roots or adventitious roots (AR)^{5,6}. Adventitious lateral roots are not primed in the root meristem, but are initiated *de novo* between lateral roots in a non-acropetal sequence. In adventitious lateral root, auxin promotes the fate transition of the *de novo* root founder pericycle cells by upregulating the expression of *WUSCHEL-RELATED HOMEODOMAIN 11* (WOX11) and the partially redundant WOX12 to initiate adventitious root primordia⁶⁻⁸.

We conducted homologous gene comparisons for PLT 3, PLT 5, PLT 7 and WOX11, WOX12 in 84K, respectively. Because 84K is a hybrid of *Populus alba* and *P. glandulosa*, it has two sets of chromosomes sequences. A total of 20 homologous genes of PLT 3 or PLT 5 or PLT 7 and 7 homologous genes of WOX11 or WOX12 were detected. Due to the large number of homologous genes, the function of genes is redundant, and it is difficult to obtain the ideal 84K poplar mutant by knocking out a few genes. Additionally, our laboratory has academic exchanges with Professor Ton Bisseling, Professor Viola Willemsen, and Professor Huchen Li. During their discussion on this study, we learned that their lab possess the *plt3plt5plt7* Arabidopsis triple mutant, which exhibits compromised lateral root formation, and the *wox11wox12* Arabidopsis double mutant, displaying deficiencies in adventitious

lateral roots and adventitious root formation, and they generously shared the mutant seeds with us⁹. Therefore, we used Arabidopsis mutants to investigate the genetic pathway of PTO1-induced secondary rooting growth. We have already thanked Professors Viola Willemsen and Huchen Li in the Acknowledgments section.

For instance:

We thank Professors Ton Bisseling and Huchen Li for their careful guidance and constructive ideas on this study. We thank Professors Viola Willemsen and Huchen Li for sharing the *plt3plt5plt7* and *wox11wox12* mutants.

Phylogenetic analysis of PLT 3, PLT 5, and PLT 7 homologous genes in the 84K genome.

Phylogenetic analysis of WOX11 and WOX12 homologous genes in the 84K genome.

Arabidopsis and poplar differ significantly in their physiology, ecology, and genetic makeup, and that bacterial strains may not have a high degree of conservation across plant species. The results obtained in Arabidopsis may not apply to poplars. Therefore, we revised the relevant description to emphasize the results obtained in Arabidopsis and modified the title. We have carefully reviewed the revised manuscript and hope we have addressed the concerns sufficiently.

For instance:

Original manuscript:

Flavones enrich rhizosphere *Pseudomonas* to enhance nitrogen utilization and lateral root growth in *Populus*

Current manuscript:

Flavones enrich rhizosphere *Pseudomonas* to enhance nitrogen utilization and **secondary root** growth in *Populus*

References

1. Li, Q. et al. Plant growth-promoting rhizobacterium *Pseudomonas* sp. CM11 specifically induces lateral roots. *New Phytol.* **235**, 1575-1588 (2022).
2. Du, Y. & Scheres, B. Lateral root formation and the multiple roles of auxin. *J. Exp. Bot.* **69**, 155-167 (2018).
3. Du, Y. & Scheres, B. PLETHORA transcription factors orchestrate de novo organ patterning during Arabidopsis lateral root outgrowth. *Proceedings of the National Academy of Sciences.* **114**, 11709-11714 (2017).
4. Hofhuis, H. et al. Phyllotaxis and Rhizotaxis in Arabidopsis Are Modified by Three PLETHORA Transcription Factors. *Curr. Biol.* **23**, 956-962 (2013).
5. Karlova, R., Boer, D., Hayes, S. & Testerink, C. Root plasticity under abiotic stress. *Plant Physiology.* **187**, 1057-1070 (2021).
6. Sheng, L. et al. Non-canonical *WOXII*-mediated root branching contributes to plasticity in Arabidopsis root system architecture. *Development.*, (2017).
7. Liu, J. et al. *WOXII* and *I2* Are Involved in the First-Step Cell Fate Transition during de Novo Root Organogenesis in Arabidopsis. *The Plant Cell.* **26**, 1081-1093 (2014).
8. Xu, L. *De novo* root regeneration from leaf explants: wounding, auxin, and cell fate transition. *Curr. Opin. Plant Biol.*, 125-126 (2017).
9. Li, Q. et al. Plant growth-promoting rhizobacterium *Pseudomonas* sp. CM11 specifically induces lateral roots. *New Phytol.* **235**, 1575-1588 (2022).

46. Reviewer #2: L. 306: This does not exclude the possibility that the effect on the IAA pathways could be indirect.

Our response: We thank Reviewer 2 for this constructive comment. We have modified this conclusion.

Original manuscript:

These results suggest that Pto1 inoculation induces the LR pathway mediated by *PLT3PLT5PLT7* through the secretion of IAA.

Current manuscript:

These results suggest that Pto1-secreted IAA plays a critical role in inducing *PLT3PLT5PLT7*-mediated LR pathways in Arabidopsis.

47. Reviewer #2: L.307 recruit

Our response: We thank Reviewer 2 for this valuable comment. We modified “recruited” to “recruit”.

Original manuscript:

PopGL3 regulated the synthesis of tricetin to recruited *Pseudomonas*

Current manuscript:

PopGL3 regulates the synthesis of tricetin to **recruit** *Pseudomonas*

48. Reviewer #2: L. 313 to 324: Too many assumption here without the direct results to support these assumptions. Suggest only keeping the direct result in the result section and move the assumptions to the discussion section.

Our response: We thank Reviewer 2 for this kind comment. We have modified this results section and removed these assumptions.

For instance:

We removed “*GL3* was reported to interact with MYB transcription factors and WD40 repeat proteins to form the MYB-bHLH-WD40 (MBW) transcriptional complex, regulating anthocyanin synthesis. However, the potential roles of *GL3* in flavone synthesis and interactions with the rhizosphere microbiome remain unclear.” and “Notably, *F3'H* catalyzes the conversion of flavone precursors into flavones (Fig. 5C).”

49. Reviewer #2: Line 325: what was the soil source?

Our response: We thank Reviewer 2 for this kind comment. The soil used here is a

natural soil mixture. We have added relevant descriptions to the revised manuscript.

Original manuscript:

Following two months of growth in unsterilized nitrogen-poor soil (with a small amount of ^{15}N -labeled ammonium nitrate), *PopGL3-OE* and *PopCHS4-OE* plants displayed increased biomass and leaf nitrogen accumulation (P -values < 0.01 ; Fig. 5D, G and Supplementary Fig. 13B, C).

Current manuscript:

Following two months of growth in an **unsterilized natural soil mixture** (low nitrogen; with a small amount of ^{15}N -labeled ammonium nitrate), *PopGL3-OE* and *PopCHS4-OE* plants displayed increased biomass and leaf nitrogen accumulation (P -values < 0.01 ; Fig. 5D, G and Supplementary Fig. 12B, C).

50. Reviewer #2: Line 325-347: Briefly explain the exp design for L. 325 to 347 will help us to understand the scope of this section.

Our response: We appreciated your valuable suggestions. Based on your suggestions, we have added an explanation of the exp design to the revised manuscript and hope that this concern has been adequately addressed.

For instance:

The contribution of BNF to nitrogen nutrition in different poplar genotypes was determined by the ^{15}N isotope dilution method.

Amplicon sequencing was used to elucidate the reasons for the differences in BNF among different poplar genotypes.

51. Reviewer #2: L.413-419 Again, concluding that the *Pto1* inoculation induces the LR pathway mediated by *PLT3PLT5PLT7* through the secretion of IAA based solely on results from *Arabidopsis* is a substantial extrapolation. This leap assumes a high degree of conservation in the interaction between the bacterial strain and these very different plant species.

Our response: We appreciated your valuable suggestions. We have explained above

the reasons for choosing Arabidopsis as the subject of study and revised the relevant conclusions, and we hope we have addressed the concerns sufficiently.

52. Reviewer #2: L. 429-430: Kindly suggest moving this part to the end of Methods section.

Our response: We thank Reviewer 2 for this kind suggestion. This part has been moved to the end of the Methods section.

53. Reviewer #2: L. 448 What do you mean by four parts of soil? Did you mix all soil samples that collected from each site?

Our response: We thank Reviewer 2 for these constructive comments. These four parts of soil refer to the soil collected earlier from the four poplar forests described above. All soil samples collected from each site were mixed, and the soil collected from each poplar forest was considered one part of the soil sample. We have added relevant descriptions in the Methods section.

For instance:

Notably, the poplars in these forests were more than 15 years old and had never been fertilized. At a distance of 2-3 m around the poplars, we collected 10-40 cm of soil after removing the surface 10 cm of soil. Each forest soil is taken from at least five poplars, and all soil samples collected from each site are mixed.

54. Reviewer #2: L. 452 Please add the detailed plant incubation conditions and fertilization in the supplementary information and cite it here.

Our response: We thank Reviewer 2 for this kind comment. We have added a detailed description of plant incubation conditions and fertilization here.

For instance:

They were grown in the same environment (phytochamber conditions: 25 °C; 16 h day/8 h night light cycle) for three months, and water poured every two days. In order to ensure the normal growth of the seedlings, we used 1 g fertilizer (Huaduo 1, China;

N: 20%, P: 20%, and K: 20%) in 1 L sterile water to fertilize after transplanting and watered 300 ml per pot of poplar, not again in the later period.

55. Reviewer #2: L. 458: No chlorophyll data were presented in the result section?

Our response: We thank Reviewer 2 for this kind comment. We did not specifically show the chlorophyll data in the results section, but it was shown in Fig. 3G (Fig. 3F in the revision). In addition, we added all phenotypic data from the nine poplar species to Supplementary Data 1.

Fig. 3 (F) Pearson's correlation between dominant genera and 11 characteristics of poplars. The color of the heat map represents the size of the correlation coefficient.

56. Reviewer #2: Figure 3F: what is the unit of X-axis?

Our response: We thank Reviewer 2 for this constructive comment. The X-axis in Figure 3F (Fig. 3E in the revision) refers to the relative abundance of Pseudomonadaceae, which has been modified in the revised manuscript.

Fig. 3 (E) Pearson's correlation between Pseudomonadaceae and the shoot and root biomass of poplars.

57. Reviewer #2: *Fig. 4 I am quite confused about the statistical analysis in Fig 4B and F. What do you mean by different groups? I also noticed the absence of asterisks above the bars in the figures. What does that statistically mean?*

Our response: We thank Reviewer 2 for these constructive comments. To investigate the effects of flavones (tricin, apigenin) on motility-related genes in pseudomonads (Pto1). We used the DMSO-treated strain as a negative control, and detected the expression of flagellar-related genes (*motA*, *fliG*, *flhA*, and *bifA*) and biofilm formation-forming genes (*algU* and *rmlD*) in the triclin and apigenin-treated strains by qRT-PCR. Therefore, the treatment in Fig. 4B was divided into three groups: DMSO, Tricin, and Apigenin. The DMSO-treated strain was used as a template for the negative control. We have added relevant descriptions to the legend of Fig. 4B.

Original manuscript:

qRT-PCR assays revealed that flagellar-related genes (*motA*, *fliG*, *flhA*, and *bifA*) and biofilm formation-related genes (*algU* and *rmlD*) expression were induced by triclin and apigenin.

Current manuscript:

qRT-PCR assays revealed that triclin and apigenin induce the expression of

flagellar-related genes (*motA*, *fliG*, *flhA*, and *bifA*) and biofilm formation-related genes (*algU* and *rmlD*) in pseudomonad (Pto1). **The DMSO-treated strain was used as a template for the negative control.**

--To investigate the effects of pseudomonads (Pto1) on lateral root development of poplars (84K) in sterile nitrogen-poor medium and sterile nitrogen-rich medium. We used 10 mM MgSO₄ solution as a negative control, and Pto1 suspended in 10 mM MgSO₄ solution was dropped on the roots of poplars. After 20 days of growth, the effects of Pto1 on the biomass and lateral roots of poplar were observed. Therefore, the 'CK' in Fig. 4F represents the 10 mM MgSO₄ solution. We have added relevant descriptions to the legend of Fig. 4F.

Original manuscript:

The secondary root number, secondary root length, and total fresh biomass of poplars inoculated with Pto1 in sterile nitrogen-poor medium and sterile nitrogen-rich medium. $n = 3$ biologically independent samples.

Current manuscript:

The secondary root number, secondary root length, and total fresh biomass of poplars (84K) inoculated with Pto1 in sterile nitrogen-poor medium and sterile nitrogen-rich medium. $n = 3$ biologically independent samples. **10 mM MgSO₄ solution was used as a negative control.**

--We modified the asterisks in **Figures 4B and F**. In Fig. 4B, the differences between different groups were re-analyzed using One-way ANOVA, and the differences between different groups were represented by letters. The asterisks in Fig. 4F were moved to the middle of the two bars, representing the difference between the treatment group (Pto1) and the control group (CK). We have carefully reviewed the revised manuscript and hope we have addressed the concerns sufficiently.

Fig. 4 (B) qRT-PCR assays revealed that triclin and apigenin induce the expression of flagellar-related genes (*motA*, *fliG*, *flhA*, and *bifA*) and biofilm formation-related genes (*algU* and *rmlD*) in pseudomonad (*Pto1*). The DMSO-treated strain was used as a negative control. Different letters indicate significantly different groups (One-way ANOVA, P -values < 0.05).

Fig. 4 (F) The secondary root number, secondary root length, and total fresh biomass of poplars (84K) inoculated with *Pto1* in sterile nitrogen-poor medium and sterile nitrogen-rich medium. $n = 3$ biologically independent samples. 10 mM $MgSO_4$ solution was used as a negative control. Asterisks indicate significant differences between different groups (Student's t-test, *** P -values < 0.001, ** P -values < 0.01, * P -values < 0.05, ns: not significant). FW, fresh weight.

#####Reviewer #5:#####

Reviewer #5 (Remarks to the Author):

The authors state that the overall goal of this manuscript is to generate a comprehensive gene-metabolite-microbe network. After reading this manuscript many times, it is unclear to me if that is the actual goal. I am impressed by the large dataset the authors are presenting and see the utility in such a dataset. There have been significant resources put into this manuscript in terms of both experiments and data collected. Unfortunately, as written, it is unclear what the major questions are within this project, what hypotheses the authors are testing, and how this dataset is being leveraged to answer them. Thorough revisions are needed before publication. I recommend the authors develop a conceptual framework or diagram that highlights the questions they are asking and demonstrates the utility of each individual experiment and dataset. It was very challenging to follow the flow and rationale of each individual experiment within the methods and results. A solid conceptual framework can be an organizing theme throughout the whole manuscript. Additionally, there are numerous grammatical errors throughout that need addressing. Detailed comments/suggestions/questions from each section are below.

Our response: We sincerely appreciate Reviewer 5 for the time and effort in reviewing the manuscript. Your insightful feedback has been instrumental in enhancing the quality and rigor of our work. We have made point-by-point responses to these comments and made corresponding revisions to the manuscript as below. We hope that these concerns have been adequately addressed.

Abstract:

1. Reviewer #5: Within the abstract, the authors mention the 'Matthew Effect' on poplar - microbe interactions. This concept is not picked up within the introduction and was confusing.

Our response: We thank Reviewer 5 for this constructive comment. This concept of the 'Matthew Effect' has been removed.

Introduction:

2. Reviewer #5: Line 90 - This paragraph is confusing and needs clarification.

I believe the authors mean 'In this study', then on line 92 the statement 'the regulating chains that how' is confusing. It is unclear what the authors mean by this. What are the key questions and hypotheses driving this research? The authors do not lay the groundwork to set up the story they are telling throughout the rest of the paper.

Our response: We thank Reviewer 5 for these constructive comments. We have modified this sentence in the revised manuscript to make it easier to understand.

Original manuscript:

By using this multidimensional dataset, the regulating chains that how poplar recruiting target beneficial microbes and how the microbes affect the host's fitness were constructed.

Current manuscript:

By using this multidimensional dataset, the mechanisms of poplar recruits target beneficial microbes and these microbes affect host fitness were revealed.

--We have added hypotheses to the introduction section of the revised manuscript.

For instance:

In this study, we hypothesize that (1) the comprehensive gene-metabolite-microbe co-expression network provides an effective research tool to elucidate the genetic mechanisms by which poplar regulates metabolite-mediated specific microbial recruitment; (2) poplar regulates the secretion of flavonoids by functional genes to reshape the rhizosphere microbial community, and attract specific microbes that, through the 'cry for help', improve plant adaptability to low-nitrogen stress.

3. Reviewer #5: Methods - There are numerous areas of the methods that require clarification and restructuring. It is unclear how all the methods/analyses fit

together and inform a coherent story. Looking through the supplementary material, it appears there are significantly more experiments/measurements than were included in the main document. It is unclear to me how the data chosen inform a coherent story. I suggest major restructuring throughout. A targeted question or set of questions should be established within the introduction and followed throughout the manuscript.

Our response: We thank Reviewer 5 for these constructive suggestions. Due to the word limit of the main text, we had to put some analytical and experimental methods into the supplementary information. Based on your suggestions, we have clarified and restructured the unclear areas of the methods. In addition, we moved some of the key methods to the main document and reorganized the Materials and methods sections in order of results. Hypotheses are added in the introduction and followed throughout the manuscript. We hope that all concerns raised by the reviewer are now well addressed.

For instance:

(1) We have clarified and restructured the unclear areas of the methods.

Notably, the poplars in these forests were more than 15 years old and had never been fertilized. At a distance of 2-3 m around the poplars, we collected 10-40 cm of soil after removing the surface 10 cm of soil. Each forest soil is taken from at least five poplars, and all soil samples collected from each site are mixed.

They were grown in the same environment (phytochamber conditions: 25 °C; 16 h day/8 h night light cycle) for three months, and water poured every two days. In order to ensure the normal growth of the seedlings, we used 1 g fertilizer (Huaduo 1, China; N: 20%, P: 20%, and K: 20%) in 1 L sterile water to fertilize after transplanting and watered 300 ml per pot of poplar, not again in the later period.

Since the root and rhizosphere soil of one Peu-H plant were insufficient for subsequent sequencing experiments, the two individual plants were treated as one biological replicate, and the one biological replicate of all phenotypic data for Peu-H was the average of the two biological plants. For other poplar species, three plants of

comparable growth were selected as biological replicates.

(2) The order of the reorganized supplementary methods is as follows:

Main text

Plant and soil materials, growth conditions;

Plant measurements and sample collection;

Soil transplantation and root secretion inoculation assays;

Establishment of the co-expression network;

Correlation analysis between modules and bacterial families;

Pseudomonads swarming motility assay;

Pseudomonads total RNA extraction, cDNA preparation, and quantitative real-time PCR assay;

Growth promotion experiment with pseudomonad inoculation;

Construction of poplar transgenic lines;

RNA extraction and qPCR from transgenic poplar lines;

RFP-labeled Pseudomonas strain Pto1;

Colonization of poplar roots by RFP-tagged isolate;

Pseudomonads root colonization measurements.

Supplementary information

Supplementary Methods 1. 16S rRNA gene sequencing and microbiome analysis;

Supplementary Methods 2. Root total RNA extraction, transcriptome sequencing and analysis;

Supplementary Methods 3. Metabolites measurement and metabolome analysis;

Supplementary Methods 4. Isolation and functional detection of pseudomonads;

Supplementary Methods 5. ¹⁵N isotope dilution assay;

Supplementary Methods 6. DNA affinity purification sequencing (DAP-seq) and data analysis;

Supplementary Methods 7. Flavone quantification of root extract and root exudates.

(3) We have added hypotheses to the introduction section of the revised manuscript.

In this study, we hypothesize that (1) the comprehensive gene-metabolite-microbe co-expression network provides an effective research tool to elucidate the genetic mechanisms by which poplar regulates metabolite-mediated specific microbial recruitment; (2) poplar regulates the secretion of flavonoids by functional genes to reshape the rhizosphere microbial community, and attract specific microbes that, through the ‘cry for help’, improve plant adaptability to low-nitrogen stress.

4. Reviewer #5: Within the methods, it is confusing how and why the soils were collected. I assume the authors wished to provide diverse natural inoculum for the plants to select from. Did the authors add inoculum to a standard mix to control for variation in nutrients as opposed to microbes?

Our response: We thank Reviewer 5 for these kind comments. As the reviewer stated, we collect a variety of natural soils to provide a variety of natural inoculants for plants to choose from. Plants and environmental microorganisms have established a co-evolution relationship¹. Although genetic variation within plant hosts is influential for root microbial community assembly, variation in soil conditions is associated with differential microbial colonization, such as soil microbial community composition^{2,3}. Because mixing soils from different sources can provide more abundant microorganisms, thus forming a microbial bank for poplars to choose from.

--We used 1 g fertilizer (Huaduo 1, China; N: 20%, P: 20%, and K: 20%) to 1 L sterile water to fertilize after transplanting, and watered 300 ml per pot of poplar, not again in the later period. We have added relevant descriptions to the Methods section.

For instance:

In order to ensure the normal growth of the seedlings, we used 1 g fertilizer (Huaduo 1, China; N: 20%, P: 20%, and K: 20%) in 1 L sterile water to fertilize after transplanting and watered 300 ml per pot of poplar, not again in the later period.

References

1. Mesny, F., Hacquard, S. & Thomma, B. P. Co-evolution within the plant holobiont drives host performance. *Embo Rep.* **24**, e57455 (2023).
2. Veach, A. M. et al. Rhizosphere microbiomes diverge among *Populus trichocarpa* plant-host genotypes and chemotypes, but it depends on soil origin. *Microbiome.* **7**, (2019).
3. Guo, Q., Liu, L., Liu, J., Korpelainen, H. & Li, C. Plant sex affects plant-microbiome assemblies of dioecious *Populus cathayana* trees under different soil nitrogen conditions. *Microbiome.* **10**, (2022).

5. Reviewer #5: It is also unclear from the methods how the experiment flowed. Reading the results, it appears numerous experiments were conducted sequentially.

Our response: We thank Reviewer 5 for this kind comment. Due to the word limit of the main text, we only chose some key methods to put in the main text, and the rest of the methods were put in supplementary information. We have reorganized the method sections in the order of experiments to make them more clear. The new order of the reorganized supplementary methods has been shown above.

6. Reviewer #5: A schematic highlighting the plants selected including the origin, and the soils collected would be helpful.

Our response: We thank Reviewer 5 for this kind suggestion. We have added a detailed description of the collected poplars and soils in Supplementary Table 4 and 5.

Supplementary Table 4. Details of nine poplar species in four sections.

Sections	Locations	Abbreviations	Sources
Leuce soil	E: 115°22'8", N: 36°30'54"	Leuce	Guanxian state-owned P. tomentosa forest farm
Aigeiros soil	E: 115°22'8", N: 36°30'54"	Aigeiros	Guanxian state-owned P. tomentosa forest farm
Tacamahaca soil	E: 115°22'8", N: 36°30'54"	Tacamahaca	Guanxian state-owned P. tomentosa forest farm
Turanga soil	E: 80°15'18", N: 40°45'39"	Turanga	Akesu Danglang tribe

Supplementary Table 5. Details of four poplar forest soils.

Sections	Species	Abbreviations	Sources
Leuce	P. alba × P. glandulosa Y	Pal-Y	Shandong, China
	P. tomentosa × P. bolleana M	Pto-M	Beijing, China
	P. alba × P. glandulosa 84K	84K	Korea
	P. tomentosa Lumao50	LM50	Shandong, China
Aigeiros	P. euramericana 74/76	107	Shandong, China
	P. euramericana H3-1	H3-1	Shandong, China
Tacamahaca	P. szechuanica Z	Psz-Z	Sichuan, China
	P. trichocarpa M	Pot-M	America
Turanga	P. euphratica H	Peu-H	Xinjiang, China

7. Reviewer #5: Was there a 'no inoculum' control? From the figure, it looks like there was a sterilized soil control, but there is no mention of that in the methods.

Our response: We thank Reviewer 5 for this kind comment. We did not set a 'no inoculum' control in the nine species of poplar pot experiment, so we did not mention that in the methods. This can be seen from Supplementary Fig. 1 and has been added to the legend of Supplementary Fig. 1 the type of soil in which poplars grow.

For instance:

Growth differences of the whole plant (**A**) and root (**B**) between nine poplar species growing in natural soil mixture. a: Pto-M; b: 84K; c: Pal-Y; d: LM50; e: H3-1; f: 107; g: Pot-M; h: Psz-Z; i: Peu-H. Fresh shoot (**C**) and root (**D**) biomass of nine poplar species growing **in natural soil mixture**.

--To investigate whether genotype-mediated soil microbiota was involved in shaping disparities in poplar growth, a follow-up soil transplant experiment was conducted on the vigorous LM50 and the poorly performing Peu-H. We added a 'no inoculum' control to the soil transplantation experiment, i.e., sterilized soil control in Fig. 1 and Supplementary Fig. 2, which was described in detail in Methods section.

For instance:

As a control, sterile PBS buffer was used to reintroduce sterilized soil. For soil sterilization, a uniform mixture of soil that was used for Peu-H and LM50 cultivation was subjected to high-pressure sterilization at 121 °C for 60 min, followed by cooling at room temperature for at least 24 hours.

8. Reviewer #5: Why weren't all the biological reps used for the plant measurement data? Why was Peu-H treated differently and averaged?

Our response: We thank Reviewer 5 for these kind comments. Due to the fact that the biomass of Peu-H is relatively small, especially the roots, the fresh root biomass of a Peu-H plant was 0.53-0.61 g (Supplementary Data 1). However, transcriptome sequencing and metabolome sequencing required more than 0.5 g of fresh root

samples, respectively, and the root and rhizosphere soil were insufficient for subsequent sequencing experiments. Therefore, we treated the two individual Peu-H were pooled as one biological replicate. One biological replicate of all phenotypic data for Peu-H was the average of two individual plants. We have added descriptions to the Methods section.

For instance:

Since the root and rhizosphere soil of one Peu-H plant were insufficient for subsequent sequencing experiments, the two individual plants were treated as one biological replicate, and the one biological replicate of all phenotypic data for Peu-H was the average of the two biological plants. For other poplar species, three plants of comparable growth were selected as biological replicates.

9. Reviewer #5: Why the focus on pseudomonads and swarming?

Our response: We thank Reviewer 5 for this kind comment. In bulk soil, the relative abundances of Pseudomonadaceae and *Pseudomonas* in the *Leuce* were 0.86% and 0.86%, respectively, while the relative abundances of them were enriched to 13.78% and 13.78%, respectively, in the rhizosphere soil of the *Leuce*. **Notably, Pseudomonadaceae is the most abundant specific family in the *Leuce* rhizosphere, and *Pseudomonas* is the most abundant specific genus.** We have highlighted these descriptions in the revised manuscript and changed the figure (Supplementary Fig. 8).

For instance:

Pseudomonadaceae, taxa enriched from 0.86% in the bulk soil to the highest abundance in the *Leuce* rhizosphere (13.78%), was specifically enriched in *Leuce* and correlated with poplar growth (ANOVA, *P*-values < 0.01; Fig. 3E and Supplementary Fig. 8A, B).

Supplementary Fig. 8 Relative abundance of Pseudomonadaceae and Pseudomonas. (A) Relative abundance of the top 10 rhizosphere microbial families in bulk soil and rhizosphere soil. The numbers are the relative abundances of Pseudomonadaceae. The abundance of Pseudomonadaceae (B) and *Pseudomonas* (C) in the rhizosphere of four sections. Different letters indicate significantly different groups (One-way ANOVA, P -values < 0.05).

To further enrich putative regulating networks, we specifically focused on Cluster IV, which was abundant in the *Leuce*, as the sections demonstrated the best growth performances. **In cluster IV, Pseudomonadaceae showed the highest number of flavonoid-associated ASVs, totaling 58 (all ASVs belong to the *Pseudomonas*),** while the second most numerous family, Xanthomonadaceae, had only 15 ASVs. We have added these new descriptions to the revised manuscript and prepared a new figure (Fig. 3C).

For instance:

The accumulation of flavones was associated with ASVs from Pseudomonadaceae, Burkholderiaceae, Cellvibrionaceae, and Xanthomonadaceae, among which Pseudomonadaceae had the highest number of ASVs (58; all ASVs belong to the *Pseudomonas*).

Fig. 3 (C) Correlation network of flavonoid-related genes, flavones, and ASVs in Cluster IV. Red color indicates genes detected in the network. Highly correlated associations ($r \geq 0.7$, P -values < 0.01) were present.

Moreover, our established gene-metabolite-microbe co-expression network serves as an effective tool for identifying beneficial microbes. In addition to Pseudomonadaceae, we also found that among the ASVs associated with flavonoid accumulation, the most abundant families were Burkholderiaceae, Cellvibrionaceae, and Xanthomonadaceae. During bacterial isolation, we utilized both general and selective media to separate the target microbes. However, we were unable to isolate any Xanthomonadaceae or Cellvibrionaceae strains, and eleven *Pseudomonas* strains and two Burkholderiaceae strains were isolated. **These two Burkholderiaceae strains failed in the transformation of red fluorescent protein (RFP), while *Pseudomonas* strains succeeded.** Consequently, we could only select *Pseudomonas* for further experiments.

--Chemotaxis is the motility-based ability of microbes to sense chemical gradients and direct their movement either up the gradient toward the source (attraction) or down the gradient away from the source (repulsion)¹. Motility and chemotaxis of vegetative bacterial cells are essential for rhizosphere colonization, as well as for establishing primary bacteria-root interactions. Root exudates activate chemosensory pathways

and cause motile bacteria to move toward the root². Chemotaxis and motility then drive the selection of the initial contact site on the root. The success of these processes determines the root colonization efficiency. We have added a brief explanation in the revised manuscript to make it easier to understand.

For instance:

Flagellated bacteria, such as pseudomonads, could achieve movement towards plant roots through swarming motility, with the success of this process determining the efficiency of root colonization.

References

1. Liu, Y. et al. Root colonization by beneficial rhizobacteria. *Fems Microbiol. Rev.* **48**, (2024).
2. Kearns, D. B. A field guide to bacterial swarming motility. *Nat. Rev. Microbiol.* **8**, 634-644 (2010).

Results -

10. Reviewer #5: 'robust species become more robust'. How do you define robustness? This section title is confusing.

Our response: We thank Reviewer 5 for this kind comment. Here "robust" means poplar with good growth performance. This subtitle has been modified to "Growth promotion of poplar by the soil microbial community".

Original manuscript:

Robust species become more robust by shaping the root microbial community

Current manuscript:

Growth promotion of poplar by the soil microbial community

11. Reviewer #5: It is not surprising that different poplar species exhibit different traits.

Our response: We thank Reviewer 5 for this kind comment. The different poplar

species exhibit different traits was not a major concern in this study. One of the main contents of this paper is to explain the mechanisms of different traits of poplars from the perspective of poplar-microbial interaction, and to describe the trait differences of different poplar species is to lead to our research purpose. To avoid misunderstanding, we have removed the conclusion "These findings indicate substantial variations in growth performance among distinct sections of poplar under controlled conditions."

12. Reviewer #5: LM50 showed growth inhibition when transplanted into Peu-H grown soil, and Peu-H showed increased growth in LM50 soil. Is this driven by variation in soil nutrients? How were inoculations done? Was it whole soil or a subset?

Our response: We thank Reviewer 5 for these kind comments. To determine whether there are differences in nutrients in soils previously grown with different poplar genotypes (Peu-H and LM50), we have examined the total nitrogen, available phosphorus, available potassium, and organic matter contents of Peu-H and LM50 bulk soils and rhizosphere soils, respectively. Compared with bulk soil, the total nitrogen and available potassium in rhizosphere soil increased significantly, while the available phosphorus decreased significantly. However, there is no significant difference between the bulk soil and bulk soil, rhizosphere soil and rhizosphere soil between the two poplar species. This excluded the effect of soil nutrients on the results of soil transplantation. We have added the relevant descriptions to the revised manuscript.

For instance:

Notably, there were no significant differences in nutrients in soils (bulk soils or rhizosphere soils) previously grown with LM50 and Peu-H (ANOVA; Supplementary Table 1),

Supplementary Table 1. Basic chemical properties of soils.

SOM (g/kg)	TN (%)	AP (mg/kg)	AK (mg/kg)
------------	--------	------------	------------

LM50 bulk soil	23.6 b	0.070 b	11.9 a	139.1 b
Peu-H bulk soil	23.9 ab	0.072 b	12.2 a	142.7 b
LM50 rhizosphere soil	26.2 a	0.084 a	10.6 b	153.2 a
Peu-H rhizosphere soil	25.7 a	0.087 a	10.1 b	157.4 a

Note: TN: total nitrogen, SOM: soil organic matter, AP: available phosphorus, AK: available potassium. $n = 3$ biologically independent samples. Different letters indicate significantly different groups (One-way ANOVA, P -values < 0.05).

--We clarified the methods of soil transplantation in the methods section.

For instance:

The soil transplantation experiment was carried out as previously described with minor modifications, **with the whole soil (bulk soil and rhizosphere soil) transplanted**. Sterile-cultured seedlings of LM50 and Peu-H were transplanted into mixed natural soil for pot cultivation, with a normal growth of eight weeks. During transplantation, rhizosphere soils from poplar (LM50 and Peu-H) were collected according to the described method, and the rhizosphere soil suspension was reintroduced into the original pots from which the plants were uprooted. Subsequently, LM50, Peu-H, and 84K sterile-cultured seedlings were individually transplanted into pots containing soil (bulk soil and rhizosphere soil) corresponding to either the LM50 or Peu-H genotype, with three replicates for each treatment. As a control, sterile PBS buffer was used to reintroduce sterilized soil. For soil sterilization, a uniform mixture of soil that was used for Peu-H and LM50 cultivation was subjected to high-pressure sterilization at 121 °C for 60 min, followed by cooling at room temperature for at least 24 hours.

13. Reviewer #5: Line 142, what is 'compositional volatility'?

Our response: We thank Reviewer 5 for this kind comment. We modified

“compositional volatility” to “compositional variation”.

Original manuscript:

The compositional volatility of the rhizosphere microbiome among the four sections is driven by significant shifts in the relative abundance of nineteen specific bacterial phyla (LDA score > 2, Kruskal-Wallis test, FDR adjusted *P*-values < 0.05; Fig. 1C, Supplementary Fig. 3B, and Supplementary Data 3).

Current manuscript:

The **compositional variation** of the rhizosphere microbiome among the four sections is driven by significant shifts in the relative abundance of 22 specific bacterial phyla (Linear discriminate analysis effect size ‘LEfSe’, LDA score > 2, Kruskal-Wallis test, FDR adjusted *P*-values < 0.05; Fig. 1C, Supplementary Fig. 3B, and Supplementary Data 2).

14. Reviewer #5: Line 160 - the authors haven't shown specific taxa increase plant growth or performance. I would soften this statement.

Our response: We thank Reviewer 5 for this kind suggestion. Following your suggestion, we have modified this statement in the revised manuscript.

Original manuscript:

Together, these results indicate that genotype properties establish root-inhabiting bacterial communities by selecting specific microbial taxa that are associated with plant nutrition and growth performance.

Current manuscript:

Together, these results indicate that genotype properties establish root-inhabiting bacterial communities by selecting specific microbial taxa.

15. Reviewer #5: Did anything other than flavonoids show up as significant in your results?

Our response: We thank Reviewer 5 for this kind comment. Only flavonoids were detected in our results. Some compounds, such as hormones and terpenoids, were not quantified in our samples due to the limited scope of detection in this study. However, flavonoids are one of the most studied classes of such metabolites, regulating both plant development and the interaction with commensal microbes¹. The gene-metabolite-microbial co-expression network we constructed contained 17,698 DEGs (17,698/23,330; 75.86%) and 2,579 ASVs (2,579/27,716; 9.31%), which can provide an effective research tool for elucidating the mechanism of poplar metabolites mediating rhizosphere microbial composition. We have highlighted these descriptions in the Introduction section of the revised manuscript.

For instance:

Flavonoids are one of the most studied classes of such metabolites, regulating both plant development and the interaction with commensal microbes. Root secretion of flavonoids occurs frequently under biotic stress and is involved in promoting microbial colonization during stress generation. Infection of part of the tomato (*Solanum lycopersicum*) root system with *Ralstonia solanacearum* changes numerous root exudates and involves disease suppression via the recruitment of disease-suppressing *Streptomyces* for colonization, which was associated with increased exudation of 3-hydroxyflavone. Under abiotic stress, flavonoid production is often elevated in plants. However, our knowledge of how flavonoid-mediated plant-microbe interactions may improve plant resistance to abiotic stresses remains elusive.

Our laboratory has also carried out long-term research on the genetic mechanisms of poplar stress resistance and wood formation. The secretion of flavonoids occurs frequently under biotic or abiotic stress, which is the main defense substance of plants external factors such as light and temperature. Lu et al. (2021) conducted a multi-omics analysis of 300 *Populus tomentosa* individuals and revealed the genetic architecture of flavonoid metabolites in poplar leaves². Lignin is one of the important components of xylem, and lignins and flavonoids are synthesized through

the same metabolic pathway (ko00940: Phenylpropanoid biosynthesis) and interconnected with the synthesis of flavonoids. Other laboratory members have resolved the genetic mechanism of flavonoid synthesis in the poplar population's xylem (unpublished). Based on the research of the poplar population, we carried out research on the genetic mechanism of flavonoid synthesis in poplar roots, and further extended it to the influence of the interaction between flavonoids and microbes on poplar growth and stress resistance. In conclusion, based on the above reasons, flavonoids were chosen to perform further analyses. We hope that this concern has been adequately addressed.

References

1. Wang, L. et al. Multifaceted roles of flavonoids mediating plant-microbe interactions. *Microbiome*. **10**, 233 (2022).
2. Lu, W. et al. Multi-omics analysis provides insights into genetic architecture of flavonoid metabolites in *Populus*. *Ind. Crop. Prod.* **168**, 113612 (2021).

16. Reviewer #5: *Functional profiles inferred from 16S data are not trustworthy. I do not recommend their use.*

Our response: We thank Reviewer 5 for this kind suggestion. Following your suggestion, we have removed this part.

17. Reviewer #5: *Discussion - As with the intro and methods, the discussion is quite scattered and unfocused. A good conceptual framework or model could really help streamline this manuscript.*

Our response: We thank Reviewer 5 for these constructive suggestions. Based on your suggestions, we have reorganized the Introduction, Discussion, and Methods sections. We divided the discussion into three parts in the Discussion section, and each established the title. We hope that all concerns raised by the reviewer are now well addressed.

For instance:

Comprehensive gene expression, flavonoid metabolism, and rhizosphere microbial community co-expression network construction;

Strategies of poplar “cry for help” rhizosphere microorganisms under low nitrogen conditions;

Matthew effect’ of interaction between host and microorganisms.

18. Reviewer #5: The 'cry for help' theory is first introduced in the discussion. This theory could be introduced in the introduction and help unify the sections of the paper.

Our response: We thank Reviewer 5 for this constructive suggestion. Based on your suggestions, we have reorganized the introduction and added an introduction to the “cry for help” theory in the revised manuscript.

For instance:

Under the infection of pathogens, plants employ a “cry for help” strategy, and signaling chemicals activated by the immune response change the composition of the rhizosphere microbiome, recruiting beneficial microorganisms to help them resist these stresses. Therefore, it is crucial and challenging to elucidate the causal relationship between plant metabolites and beneficial microbes.

--In the Introduction, in order to increase coherence in the whole text, we emphasize the important role of flavonoids in plant-microbial interaction and "cry for help".

For instance:

Flavonoids are one of the most studied classes of such metabolites, regulating both plant development and the interaction with commensal microbes. Root secretion of flavonoids occurs frequently under biotic stress and is involved in promoting microbial colonization during stress generation. Infection of part of the tomato (*Solanum lycopersicum*) root system with *Ralstonia solanacearum* changes numerous root exudates and involves disease suppression via the recruitment of disease-suppressing *Streptomyces* for colonization, which was associated with increased exudation of 3-hydroxyflavone. Under abiotic stress, flavonoid production

is often elevated in plants. However, our knowledge of how flavonoid-mediated plant-microbe interactions may improve plant resistance to abiotic stresses remains elusive.

19. Reviewer #5: Line 377 - what is 'genotype-cried'?

Our response: We thank Reviewer 5 for this constructive suggestion. We modified “genotype-cried” to “genotype-specific”.

Original manuscript:

Consistent with these observations, the biomass of LM50 (*Leuce*) was greater than that of Peu-H (*Turanga*) in sterile and nitrogen-poor soil, while the biomass of both increased in unsterilized soil, but the difference between them expanded further, indicating that genotype-cried microbiota exerts varying degrees of positive feedback on poplar growth.

Current manuscript:

We found that LM50 (*Leuce*) produced 7.37 g more stem biomass than Peu-H (*Turanga*) in sterile and nitrogen-poor soil. In the unsterilized, nitrogen-deficient soil, this difference widened to 8.29 g (Peu-H grown soil) and 10.26 g (LM50 grown soil), respectively. Notably, there were no significant differences in nutrients between soils (bulk soils or rhizosphere soils) previously grown with LM50 and Peu-H; root exudates had no significant effect on poplar growth; and bulk soil microbial diversity showed no significant variation. These results indicate that **genotype-specific** microbiota exerts varying degrees of positive feedback on poplar growth.

20. Reviewer #5: The 'matthew effect' is introduced in the discussion with limited explanation or follow through.

Our response: We thank Reviewer 5 for this constructive suggestion. Based on your suggestions, we have reorganized the discussion and added an explanation to the 'matthew effect' in the revised manuscript.

For instance:

In social psychology, the “Matthew effect” describes the phenomenon that the strong become stronger and the weak become weaker. We introduce the concept of the ‘Matthew effect’ in plant-microbial interactions for the first time. That is, vigorous or resistant plant genotypes can recruit specific microbes to give them more growth advantages or better resistance.

Response to the editor and reviewers (NCOMMS-24-15351A)

We sincerely thank the editor and reviewers for their constructive and insightful comments. We have revised the manuscript accordingly in response to your feedback. We sincerely appreciate the time and effort dedicated to providing such thorough reviews, and we hope that we have effectively addressed the concerns raised. Below, we present our point-by-point responses to the reviewers:

#####Reviewer #4:#####

Reviewer #4 (Response):

We thank the authors for their detailed response to our previous comments. The revised manuscript starts with clear hypotheses, which benefits the readers to understand the rationale behind each individual experiment and dataset. Most of previous major concerns were addressed satisfactorily. There are only several minor points listed as below.

Our response: We are grateful for your valuable suggestions and recognition of our work. Your insightful feedback has played a crucial role in enhancing the quality and rigor of our research. Below, we provide point-by-point responses to your comments, and we hope that we have adequately addressed your concerns.

1. Reviewer #4: Line 54: ACS417 or WCS 417, please double check the strain name.

Our response: We thank Reviewer 4 for this kind suggestion. ACS417 has been changed to WCS 417. We have modified the instances of 'ACS417' to 'WCS 417' in the manuscript.

Original manuscript:

For example, flavonoids (luteolin, apigenin, etc.) could interact with rhizobial nodulation (NodD) proteins activating the transcription of nodulation genes responsible for the deformation of plant root hairs and assisting rhizobial entry via infection threads; coumarins selectively affect the assembly of rhizosphere microbial communities, inducing the colonization of *Pseudomonas simiae* ACS417, thereby improving the niche establishment of microbial partners.

Current manuscript:

For example, flavonoids (luteolin, apigenin, etc.) could interact with rhizobial nodulation (NodD) proteins activating the transcription of nodulation genes responsible for the deformation of plant root hairs and assisting rhizobial entry via infection threads; coumarins selectively affect the assembly of rhizosphere microbial communities, inducing the colonization of *Pseudomonas simiae* WCS417, thereby improving the niche establishment of microbial partners.

2. Reviewer #4: Line 118: which phenotypes were detected? Or the authors meant some parameters were measured?

Our response: We thank Reviewer 4 for this kind suggestion. We determined eleven representative phenotypes of all poplars, encompassing plant height, ground diameter, shoot biomass, root biomass, root length, leaf length, leaf width, leaf area, leaf number, chlorophyll content, and leaf nitrogen content.

We have revised this sentence and incorporated the phenotypes into the manuscript. Additionally, these phenotypes are described in detail in the 'Plant Measurements and Sample Collection' section of the Methods, and the corresponding phenotype data have been included in Supplementary Data 1.

Original manuscript:

After three months of growth, eleven phenotypes of poplar were detected (Supplementary Fig. 1; Supplementary Data 1).

Current manuscript:

After three months of growth, eleven phenotypes of poplar (plant height, ground diameter, shoot biomass, root biomass, root length, leaf length, leaf width, leaf area, leaf number, chlorophyll content, and leaf nitrogen content) were detected (Supplementary Fig. 1; Supplementary Data 1).

3. Reviewer #4: Line 202: what are these 17,698 DEGs, and 2,840 ASVs and where are the numbers from? Please mention in previous sections to give more context.

Our response: We thank Reviewer 4 for the constructive suggestion. Our study generated a comprehensive multivariate dataset of the poplar transcriptome, metabolome, and microbiome under low-nitrogen conditions. **To gain further insights into the gene-metabolites-microbe regulatory network, differential flavonoids were initially classified into six clusters with distinct accumulation patterns using the *k*-means clustering algorithm. Subsequently, using metabolites as a medium, Pearson correlation analysis ($r \geq 0.7$, P -values < 0.01) was conducted to identify differentially expressed genes (DEGs) and amplicon sequence variants (ASVs) that exhibited strong correlations with the flavonoids in each cluster. For instance, in Cluster I, 3,672 DEGs and 865 ASVs exhibited strongly associated with at least one of the eleven flavonoids. Across all clusters, a total of 17,698 DEGs and 2,840 ASVs were co-expressed with at least one differential flavonoid.** The 110 differential flavonoids, 17,698 DEGs, and 2,840 ASVs formed a gene-metabolite-microbe co-expression network consisting of six clusters, where the genes, flavonoids, and microbes within each cluster exhibited a consistent trend in abundance. **Detailed information of the co-expression network is available in Supplementary Data 8.** We have modified these sentences and hope we have adequately addressed this concern.

Original manuscript:

Next, we quantified 129 flavonoids from the root samples of nine poplar species, of which 110 (85.27%) were differentially accumulated in at least two species (fold change ≥ 3 or ≤ 0.333 , P -values < 0.05 ; Supplementary Data 7). To gain further

insights into the gene-metabolites-microbiome regulatory network, the 110 differential flavonoids, 17,698 DEGs, and 2,840 ASVs were further classified into six co-expression clusters using the *k*-means clustering algorithm and Pearson's correlation analysis ($r \geq 0.7$, P -values < 0.01 ; Fig. 2, Supplementary Fig. 5, and Supplementary Data 8). These clusters demonstrated a distinct abundance pattern related to specific sections, such as *Turanga* (Cluster I), *Leuce* (Cluster IV), and *Aigeiros* (Cluster V).

Current manuscript:

Next, we quantified 129 flavonoids from the root samples of nine poplar species, of which 110 (85.27%) were differentially accumulated across at least two species (fold change ≥ 3 or ≤ 0.333 , P -values < 0.05 ; Supplementary Data 7). To gain further insights into the gene-metabolites-microbiome regulatory network, differential flavonoids were initially classified into six clusters based on their accumulation patterns using the *k*-means clustering algorithm (Supplementary Data 8). Subsequently, a rigorous correction (Pearson; $r \geq 0.7$, P -values < 0.01) was employed to screen for DEGs and ASVs that were significantly associated with the flavonoids in each cluster (Fig. 2, Supplementary Fig. 5, and Supplementary Data 8). Across all clusters, a total of 17,698 DEGs and 2,840 ASVs were co-expressed with at least one flavonoid. The genes, flavonoids, and microbes within these clusters demonstrated a distinct abundance pattern related to specific sections, such as *Turanga* (Cluster I), *Leuce* (Cluster IV), and *Aigeiros* (Cluster V).

4. Reviewer #4: Line 207: will it change the clustering patterns by removing the flavonoid-correlated genes and ASVs?

Our response: We thank Reviewer 4 for this constructive suggestion. We performed Principal Component Analysis (PCA) and Hierarchical Clustering Analysis (HCA) for genes and ASVs not strongly associated with flavonoids. The results indicated that these genes and ASVs also showed clustering patterns similar to the subgroups of global genes and microbes. We have removed this sentence and Supplementary Fig. 6

(Original manuscript) in the revised manuscript.

After removing ASVs and genes that were highly correlated ($r \geq 0.7$, P -values < 0.01) with any of the flavonoids, the PCA and HCA of the remaining ASVs (A, C) and genes (B, D) still showed clustering patterns similar to the clusters of global microbes and genes.

5. Reviewer #4: Line 251: what are flavones?

Our response: Thank you for your careful review. We have corrected 'flavnes' to 'flavones' in the revised manuscript and have meticulously checked the entire document for accuracy.

Original manuscript:

Within the metabolic cluster, the flavones, tricetin and apigenin (with their derivatives), were uncovered as the most enriched metabolites in *Leuce* (ANOVA, P -values < 0.01 ; Fig. 3A).

Current manuscript:

Within the metabolic cluster, the flavones, tricetin and apigenin (with their derivatives),

were uncovered as the most enriched metabolites in *Leuce* (ANOVA, P -values < 0.01 ; Fig. 3A).

6. Reviewer #4: Line 258: in order to ask... Please write it more concisely.

Our response: We thank Reviewer 4 for the valuable suggestion. We have corrected 'In order to ask' to 'To determine' in the revised manuscript.

Original manuscript:

In order to ask whether poplar rhizosphere microbiome composition is linked with its transcriptome signature, flavonoid metabolism, and growth performance, we performed detailed analyses of the subnetwork of genes, flavones, microbes, and growth traits.

Current manuscript:

To determine whether poplar rhizosphere microbiome composition is linked with its transcriptome signature, flavonoid metabolism, and growth performance, we performed detailed analyses of the subnetwork of genes, flavones, microbes, and growth traits.

#####Reviewer #5:#####

Reviewer #5 (Response):

Reviewer #5: The authors have made significant improvements to this manuscript and have addressed the majority of my concerns. I still struggle a bit with how all the datasets fit together to tell a cohesive story and recommend a bit of restructuring and a useful conceptual diagram to solidify how everything flows together.

Our response: We sincerely appreciate the time and effort Reviewer 5 dedicated to reviewing our manuscript. Your insightful feedback has been instrumental in enhancing the quality and rigor of our work.

We thank Reviewer 5 for this constructive suggestion. Our study generated a comprehensive multivariate dataset of the poplar transcriptome, metabolome, and microbiome under low-nitrogen conditions. To fit together the datasets of genes, metabolites, and microbes, differential flavonoids were initially classified into six clusters based on distinct accumulation patterns using the *k*-means clustering algorithm. Subsequently, using metabolites as a medium, Pearson correlation analysis ($r \geq 0.7$, P-values < 0.01) was conducted to identify differentially expressed genes (DEGs) and amplicon sequence variants (ASVs) that exhibited strong correlations with the flavonoids in each cluster. For instance, in Cluster I, 3,672 DEGs and 865 ASVs were strongly associated with at least one of the eleven flavonoids. Across all clusters, a total of 17,698 DEGs and 2,840 ASVs were co-expressed with at least one differential flavonoid. The 110 differential flavonoids, 17,698 DEGs, and 2,840 ASVs formed a gene-metabolite-microbe co-expression network consisting of six clusters, where the genes, flavonoids, and microbes within each cluster exhibited a consistent trend in abundance.

We specifically focused on Cluster IV, which showed the highest peak at the *Leuce* that demonstrated the best growth performances. Genes, flavonoids, and microbes within Cluster IV were selected for detailed subnetwork analysis with poplar phenotypes. We found that genes in Cluster IV were significantly enriched in pathways associated with flavonoid metabolism, such as phenylalanine metabolism,

phenylpropanoid biosynthesis, and flavonoid biosynthesis. Within the metabolic cluster, flavones like tricetin and apigenin (with their derivatives) were particularly abundant and significantly enriched in *Leuce*. Notably, Pseudomonadaceae represented the most numerous taxa in Cluster IV, with 62 ASVs (all ASVs belong to the *Pseudomonas*). The Pseudomonadaceae have been demonstrated to enhance plant growth through the processes of biological nitrogen fixation or phosphorus solubilization¹. Correlation analysis revealed that genes related to flavonoid biosynthesis and flavones exhibited the strongest association with Pseudomonadaceae and *Pseudomonas*, while the increased abundance of Pseudomonadaceae and *Pseudomonas* was highly correlated with the growth characteristics of poplar. Additionally, *PopGL3*, a transcription factor known to be involved in anthocyanin synthesis², was co-expressed with flavones and genes related to flavonoid biosynthesis in Cluster IV; however, its potential roles of *GL3* in flavone synthesis and interactions with the rhizosphere microbiome remain unclear. Hence, *PopGL3*, flavones (apigenin, tricetin), and *Pseudomonas* were selected for further experimental study.

We restructured the descriptive part of the constructed gene-metabolite-microbe co-expression network to make it easier to understand.

Original manuscript:

Next, we quantified 129 flavonoids from the root samples of nine poplar species, of which 110 (85.27%) were differentially accumulated in at least two species (fold change ≥ 3 or ≤ 0.333 , P -values < 0.05 ; Supplementary Data 7). To gain further insights into the gene-metabolites-microbiome regulatory network, the 110 differential flavonoids, 17,698 DEGs, and 2,840 ASVs were further classified into six co-expression clusters using the k -means clustering algorithm and Pearson's correlation analysis ($r \geq 0.7$, P -values < 0.01 ; Fig. 2, Supplementary Fig. 5, and Supplementary Data 8). These clusters demonstrated a distinct abundance pattern related to specific sections, such as *Turanga* (Cluster I), *Leuce* (Cluster IV), and *Aigeiros* (Cluster V).

Current manuscript:

Next, we quantified 129 flavonoids from the root samples of nine poplar species, of which 110 (85.27%) were differentially accumulated across at least two species (fold change ≥ 3 or ≤ 0.333 , P -values < 0.05 ; Supplementary Data 7). To gain further insights into the gene-metabolites-microbiome regulatory network, differential flavonoids were initially classified into six clusters based on their accumulation patterns using the k -means clustering algorithm (Supplementary Data 8). Subsequently, a rigorous correction (Pearson; $r \geq 0.7$, P -values < 0.01) was employed to screen for DEGs and ASVs that were significantly associated with the flavonoids in each cluster (Fig. 2, Supplementary Fig. 5, and Supplementary Data 8). Across all clusters, a total of 17,698 DEGs and 2,840 ASVs were co-expressed with at least one flavonoid. The genes, flavonoids, and microbes within these clusters demonstrated a distinct abundance pattern related to specific sections, such as *Turanga* (Cluster I), *Leuce* (Cluster IV), and *Aigeiros* (Cluster V).

Original manuscript:

To further enrich putative regulating networks, we specifically focused on Cluster IV and Cluster I, which abundant in the *Leuce* and *Turanga*, respectively (Supplementary Fig. 5A–C), as the two sections demonstrated the most contrasting growth performances (Supplementary Fig. 1C, D).

Current manuscript:

To further enrich putative regulating networks, we specifically focused on Cluster IV and Cluster I, which **peaked at** the *Leuce* and *Turanga*, respectively (Supplementary Fig. 5A–C), as the two sections demonstrated the most contrasting growth performances (Supplementary Fig. 1C, D).

--According to your suggestion, we drew a conceptual diagram of the experimental design and hope we have adequately addressed this concern.

1. Sample collection and sequencing

2. Multi-omics analyses and Experimental processing

Supplementary Fig. 13 Conceptual diagram of experimental design of this study.

1. Sample collection and sequencing: Fine-root (< 2 mm), rhizosphere soil, and bulk soil samples were collected from nine poplar species across four sections (*Leuce*,

Aigeiros, *Tacamahaca*, and *Turanga*) cultivated in low-nitrogen natural soil for three months. Three biological replicates per species. Transcriptome and flavonoid metabolite sequencing were conducted on root samples; 16S rRNA amplicon sequencing was performed on rhizosphere soil and bulk soil samples. 2. Multi-omics analyses and Experimental processing: Multi-omics analyses were performed on all datasets to elucidate the intricate mechanisms of gene-metabolite-microbe interactions. Differential flavonoids were classified into six clusters based on their accumulation patterns using the *k*-means clustering algorithm, and a rigorous correction (Pearson; $r \geq 0.7$, P -values < 0.01) was employed to screen for DEGs and ASVs that were significantly associated with the flavonoids in each cluster. Since Cluster IV peaked at the *Leuce*, which exhibited the best growth performance, the genes, metabolites, and microbes in Cluster IV were selected for detailed subnetwork analysis with poplar phenotypes. Within the metabolic cluster, flavones such as tricetin and apigenin (with their derivatives) were the most abundant and significantly enriched in *Leuce*. Notably, Pseudomonadaceae represented the most numerous taxa in Cluster IV, with 62 ASVs (all ASVs belong to the *Pseudomonas*). Correlation analysis revealed that genes related to flavonoid biosynthesis and flavones exhibited the strongest association with Pseudomonadaceae and *Pseudomonas*, while the increased abundance of Pseudomonadaceae and *Pseudomonas* was highly correlated with the growth characteristics of poplar. Additionally, the transcription factor *PopGL3* was found to be co-expressed with flavones and flavonoid biosynthesis genes in Cluster IV. Consequently, *PopGL3*, flavones (apigenin, tricetin), and *Pseudomonas* were selected for further investigation. *PopCHS4-OE* (*CHS* catalyzes the first committed step of the multi-branched flavonoid pathway) and *PopGL3-OE* lines were obtained via the 84K poplar genetic transformation method, and then qPCR, LC-MS, and amplicon sequencing were carried out on the poplar transgenic lines. Swarming motility assay indicated that apigenin and tricetin significantly enhance the motility of pseudomonad isolates (Pto1, Pto5, and Pto10). Subsequently, Pto1, Pto5, and Pto10 were inoculated in poplar and other plant roots for growth promotion experiments. The methods or other conditions for each step are listed around the steps. For further details, please

see the Methods of the manuscript.

References

1. Pérez Rodríguez, M. M. et al. Halotolerant native bacteria *Enterobacter* 64S1 and *Pseudomonas* 42P4 alleviate saline stress in tomato plants. *Physiol. Plant.* **174**, (2022).
2. Qi, T. et al. The Jasmonate-ZIM-Domain Proteins Interact with the WD-Repeat/bHLH/MYB Complexes to Regulate Jasmonate-Mediated Anthocyanin Accumulation and Trichome Initiation in *Arabidopsis thaliana*. *The Plant Cell.* **23**, 1795-1814 (2011).

#####Reviewer #6:#####

Reviewer #6 (Response):

I was invited to review the authors' responses to the comments from Reviewer #2. There are 4 major comments (comments 1-4) and 53 minor comments (comments 5-57). The authors have sufficiently responded to the minor comments, so I am going to focus on the major comments. The authors' responses to comments 2-4 are also sufficient but I have one additional concern regarding comment 1 as follows.

Our response: We sincerely appreciate Reviewer 6 for the time and effort in reviewing the manuscript. Your insightful feedback has been instrumental in enhancing the quality and rigor of our work.

Reviewer #6: The authors' responses to Reviewer #2's major comment 1 are not sufficient. In major comment 1, the reviewer proposed a different option for bioinformatic analysis of amplicon sequencing. The authors responded that they have changed their bioinformatics analysis method of amplicon sequencing and that they have modified the Methods section. However, it is unclear how the change of the bioinformatics analysis affects the results. The authors should clarify whether the results have been affected and if so, how the results change.

Our response: We thank Reviewer 6 for this constructive suggestion. The high taxonomic resolution based on Amplicon Sequence Variants (ASV) analyses will facilitate the detection of fine-scale variations in microbiome composition¹. We re-analyzed the microbiome compositional profile and conducted subsequent analyses using the ASV method. Consequently, we identified more bacterial taxa, including 41 phyla, 719 families, and 30,225 ASVs. In contrast, the OTU method identified 25 phyla, 170 families, and 3,775 OTUs. This difference could be attributed to the enhanced resolution provided by the ASV method. Both methods revealed that Proteobacteria and Pseudomonadaceae were the dominant taxa. A similar pattern of results was obtained because our study mainly focuses on the dominant taxa. According to the new analysis results, we redraw Fig. 1C, D, G, Fig. 2, Fig. 3 C-F, Fig. 5 H, I, Supplementary Fig. 3, Supplementary Fig. 5C, Supplementary Fig. 6A, C,

Supplementary Fig. 7, Supplementary Fig. 11 D, E, and Supplementary Fig. 12 (current manuscript), and modified the current manuscript.

For instance:

Results (A) in the previous manuscript based on OTU analyses and results (B) in the current manuscript based on ASV analyses (Relative abundance for the top 10 abundant bacterial phyla in the rhizosphere).

Results (A) in the previous manuscript based on OTU analyses and results (B) in the current manuscript based on ASV analyses (Relative abundance for the top 10 abundant bacterial families in the rhizosphere). **In both methods, the Pseudomonadaceae exhibit an abundance advantage in the rhizosphere.** The numbers are the relative abundances of Pseudomonadaceae.

Results in the previous manuscript based on OTU analyses and results in the current manuscript based on ASV analyses (Mean Shannon's diversity across the bulk soil and rhizosphere soil of different poplar species; B: bulk soil, R: rhizosphere soil).

(A) Correlation network (Original manuscript Fig. 3D; OTU method) of the top 50 microbial families with flavone modules and gene modules of Cluster IV (P -values < 0.01). The node size represents the number of microbial families significantly associated with this module. Solid edges indicate positive relationships. Edge thickness denotes the strength of correlations. (B) Correlation network (Current manuscript Fig. 3D; ASV method) of the top 20 microbial families with flavone modules and flavonoid metabolism gene modules of Cluster IV (P -values < 0.01). The node size represents the number of elements included (for example, the flavonoid biosynthesis module has 21 genes). Solid edges indicate positive relationships. Edge thickness denotes the strength of correlations. **In both methods, Pseudomonadaceae was among the top families showing the highest correlation with flavone modules**

and flavonoid-related gene modules (P -values < 0.01).

References

1. Callahan, B. J., McMurdie, P. J. & Holmes, S. P. Exact sequence variants should replace operational taxonomic units in marker-gene data analysis. *ISME J.* **11**, 2639-2643 (2017).